# Efficient robot navigation inspired by honeybee learning flights

Dequan Ou[1], Jesse J. Hagenaars[1], Maciej R. Jankowski[1], Michiel V. M. Firlefyn[1], Christophe De Wagter[1], Florian T. Muijres[2], Jacqueline Degen[3] & Guido C. H. E. de Croon[1✉]

Navigation is a crucial capability for both animals and robots. Although tiny flying insects can robustly navigate over long distances[1], state-of-the-art robot navigation methods are computationally expensive and therefore restricted to large robots[2,3]. Here we propose 'Bee-Nav', a highly efficient navigation strategy inspired by the visual learning flights of honeybees[4–6]. In equivalent robotic learning flights, a tiny neural network is trained to map omnidirectional images to a home vector based on path integration. After learning, the robot can fly far away from home, come straight back using path integration and cancel integration drift using the visual homing network. Simulations showed that, for realistic path integration accuracies, the neural network requires training on only approximately 0.25–10.00% of the total flight area. In real-world indoor and outdoor experiments, a small drone successfully returned to within 0.5 m of home for 100% of 30–110-m flights and 70% of 200–600-m flights in windy conditions, using 3.4-kB and 42-kB neural networks, respectively. The proposed navigation strategy will be vital for resource-constrained robots that perform tasks while travelling from and to a home location. Furthermore, it provides new perspectives on the neuroethology of insect navigation, from how visual learning shapes homing trajectories to the nature of cognitive maps.

Small robots are at present deprived of the autonomous navigation capabilities necessary for real-world applications. Resource-restricted robots, such as lightweight flying drones[7,8], can simply not carry or power the required computational systems for high-precision, map-based autonomous navigation[2,3]. Despite efforts towards improved computational efficiency, navigation based on detailed metric maps still requires a high-end laptop[9] or a GPU-enabled embedded computer[10]. Efficiency can be improved by sacrificing map accuracy, storing it as a topological graph with nodes as places and edges as paths[11,12]. However, the robot still needs to recognize where it is and adjust the map accordingly, leading to increased computational requirements for larger trajectories[11,13]. This limits the navigation range of the most efficient map-based robot navigation methods. The state of the art is a tiny flying robot that uses 500 kB of memory on a low-power AI chip for navigating in a 4 × 5-m area[14].

Nature shows that extremely resource-efficient, long-range navigation is possible. Small insects such as honeybees robustly navigate up to several kilometres from their hive[1]. Their impressive navigation capabilities rely on two components[15]. The first is path integration[16], which allows insects to estimate their position with respect to a starting point by integrating the directions and distances travelled. Because path integration is subject to increasing drift, insects also rely on a second component called view memory, which is the act of recalling visual landmarks and their relation to places of interest[17]. Path integration is well understood by now, even to the neuronal level[18]. By contrast,

the precise working of view memory and its interplay with path integration is less clear.

Lured by the navigational feats of insects, roboticists have proposed various insect-inspired navigation strategies. The predominant strategy is route-following, which typically relies on view memory to retrace the outbound trajectory during the return journey[19–24]. Route-following is a suitable strategy for navigating in highly cluttered environments, but in open areas, it can make the return journey unnecessarily long. Indeed, insects such as honeybees and desert ants tend to return home with a new straight path, even after long tortuous outbound journeys[25,26] (Fig. 1a). During the return journey, insects rely initially on path integration and then increasingly on view memory when nearing their home[26–28].

The idea of combining path integration and view memory in this manner formed the basis for the seminal work on 'Sahabot 2', a mobile ground robot that navigated in the desert[28]. However, separate experiments were performed for path integration and visual homing[28], and subsequent studies did not explore their combined use. For path integration, it has been shown that accuracy benefits from combining several sensory cues with a motion model, including proprioception, optical flow, airflow sensors and polarization sensors[29,30]. The accuracy reached with bioinspired mechanisms can be impressive: for example, 1.5 m drift per 100 m for a flying robot[31]. In terms of view memory, there has also been substantial progress[32], in which vision algorithms moved from landmark recognition[28,33], through the matching of

[1]Micro Air Vehicle Laboratory, Department Control & Operations, Faculty of Aerospace Engineering, Delft University of Technology, Delft, The Netherlands. [2]Experimental Zoology Group, Wageningen University, Wageningen, The Netherlands. [3]Navigation Biology Group, Institute of Biology and Environmental Sciences, Carl von Ossietzky University of Oldenburg, Oldenburg, Germany. ✉e-mail: G.C.H.E.deCroon@tudelft.nl

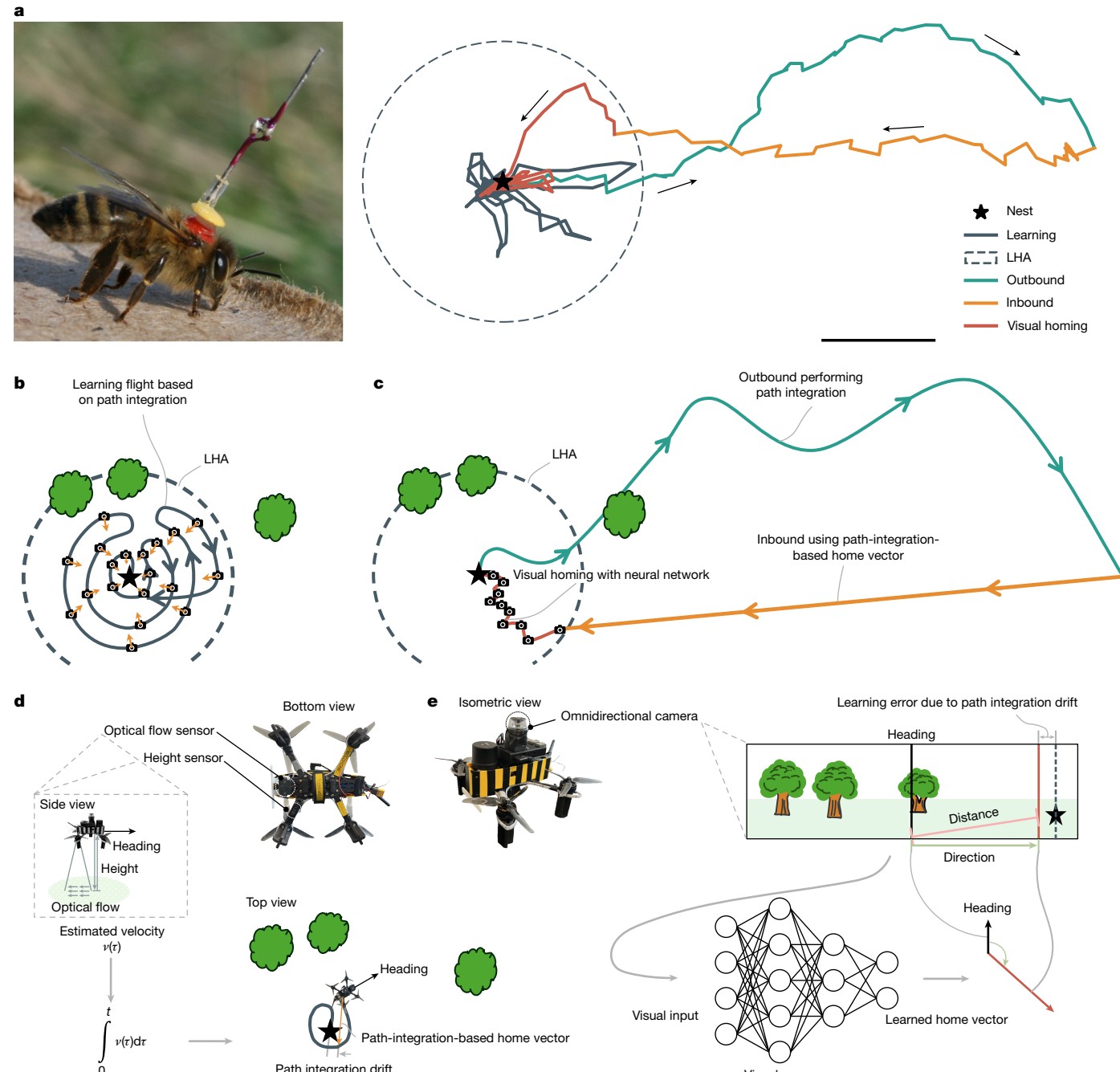

**Fig. 1 | Illustration of the proposed robot navigation strategy, Bee-Nav, inspired by honeybee learning and foraging flights. a**, Before foraging, honeybees first perform 'learning' flights (dark-grey line) close to home (star). Subsequently, they can fly out far away from home (teal line) and come back in an almost straight line (orange and red lines). Scale bar, 100 m. **b**, In Bee-Nav, the robot also first performs a learning flight, capturing omnidirectional images while using path integration for maintaining a vector (orange arrows) pointing to the home location. A neural network is trained to map the images to the home vectors. The trained network encodes an implicit view memory within the learned homing area (LHA) enclosing the learning flight trajectory (dashed circle). **c**, After learning, the robot can execute a long outbound flight to perform a task of interest (teal line), while maintaining a home vector based on path integration. It can then perform a straight inbound flight (orange line) using path integration until it supposedly arrives home. If the robot is inside the LHA, it can cancel the accumulated path integration drift by performing visual homing with its neural network (red line). It does this by making subsequent 'steps' along the homing vectors of the network. **d**, The robot in this study performs path integration based on velocity from optical flow and laser-based height measurements and a heading based on gyro integration. **e**, The view memory of the robot consists of a small feedforward convolutional neural network that maps an unwrapped omnidirectional image to a home vector. The centre of the view corresponds to the heading of the robot (solid black line) and the home direction is predicted by the neural network (vertical red line). Because the learning targets are imperfect path-integration-based home vectors, the neural network estimates are typically close but not identical to the home direction (vertical dashed black line).

snapshots[19,22,34], to neural networks for interpreting natural scenes[35–38]. Neural networks hold the potential to compress the view memory for a large visual homing area in a small parameter space. In a recent robotics study, a small 9-kB neural network enabled visual homing for distances up to 18 m in an outdoor environment[39]. The neural network transformed images directly to right or left steering commands. Learning

was performed online with a biologically plausible neural network structure and learning mechanism, inspired by the insect mushroom body[32]. The learning targets consisted of the angles from the robot to the home as obtained with the Global Positioning System (GPS)[39]. For independence of external infrastructure, the targets could be obtained from path integration.

Despite these advances, a critical gap remains: how to integrate the complementary mechanisms of path integration and view memory into a unified strategy for efficient, long-range navigation.

## Honeybee-inspired learning flights for navigation

To address this gap, we introduce a robot navigation strategy inspired by honeybee learning flights. Honeybees perform one to several such short flights before they depart on longer exploration or foraging trips[4–6] (Fig. 1a). After learning flights, honeybees can immediately forage far away, while flying back in an almost straight line. On the basis of our preliminary simulation study[40], we have developed a navigation strategy, named Bee-Nav, which is both highly efficient and suitable for long-range robot navigation. In the strategy, a robot first performs a learning flight (Fig. 1b), during which it trains a small on-board neural network to map its visual inputs directly to a home vector. This vector represents the direction and distance home from the point of view of the robot (Fig. 1d,e). The learning is self-supervised, as the target vector is determined with path integration. Self-supervised learning is a form of associative learning that is quick and requires only a few sample images and target vectors. However, because path integration drifts over time, the target vectors increasingly deviate from the true home vectors. We study the effects of this deviation on the resulting visual homing behaviour. The trained neural network represents a view memory, allowing the robot to visually estimate the home location within an area circumscribing the learning flight trajectories. We refer to this area as the learned homing area (LHA) and approximate it with a home-centred circle. After learning, the robot can fly far away from home and come back along a straight trajectory based on path integration (Fig. 1c, orange line). As long as the robot ends up in the LHA, it can cancel any accrued path integration drift with its onboard visual homing neural network (Fig. 1c, red line).

## Simulation experiments

The efficiency of Bee-Nav depends on the LHA being small compared with the total flight area. In simulation, we investigated how large the LHA needs to be for different amounts of path integration noise. The simulated robot used path integration while flying 1,000 different outbound trajectories, following a block-wave pattern that could be useful for a search mission (Fig. 2a). Each outbound flight was followed by a straight inbound trajectory back home based on noisy path integration, leading to an end point offset from the home location. We used a Gaussian noise model, which assumed the heading to drift over time. Specifically, noise was added to the heading estimate after each time step in the simulation from the normal distribution $\mathcal{N}(0, t\sigma_\psi^2)$, with $t$ in seconds. The distance estimation noise was modelled as $\mathcal{N}(0, d\sigma_d^2)$, with $d$ the distance covered during the time step in metres. We fit the parameters of the noise model to the real robot's straightforward path integration method (Supplementary Information Section 11): a tiny downward-looking optical flow sensor and laser for estimating velocity and gyro measurements for updating the heading estimate.

Figure 2b shows the percentage of the LHA with respect to the total flight area for various amounts of path integration noise. Even with the substantial path integration drift of the robot, the LHA only needs to be 3.84% of the total flight area to capture 99% of the journey end points (Fig. 2b, $\sigma_\psi = 0.63$, $\sigma_d = 0.10$). The lines represent different scaled versions of the robot's noise (both standard deviations multiplied by a factor ranging from 0.1 to 1.5). Moreover, the LHA was 0.74% when

odometry noise was similar to that of a more advanced, computationally more expensive method for visual odometry, SVO+GTSAM[41]. Finally, the noise model was fit to a bioinspired odometry method with a magnetometer heading measurement[31] for more precise path integration[42], resulting in an LHA percentage of only 0.24%. Insects also exploit absolute heading measurements to reduce integration drift, for example, through the polarization of sky light. A coarse approximation of the path integration accuracy of desert ants[27] and honeybees[43] led to LHA percentages of 7.6% and 3.4%, respectively (Supplementary Information Section 11).

A small LHA implies that the neural network can also be small, enabling inference on limited, energy-efficient computing hardware. To quantify the visual homing performance, we performed further experiments in the visually realistic NVIDIA Isaac simulator with a small 42.3-kB attention neural network (Fig. 2c). The learning flight trajectory (Supplementary Information Section 6) fits in a 10-m-radius circle around the home location. The simulated robot performed visual homing, following the vectors output by the network, starting at various distances from the home location. Figure 2d shows the results of 800 simulated visual homing runs in ten different randomly generated forests. The success rate of the proposed strategy was 100% within the LHA radius and, depending on the environment, generalized even up to 2.5 times the LHA radius. This generalization beyond the LHA is supported by our theoretical analysis (Supplementary Information Sections 1–5). Figure 2e shows 16 visual homing runs of Bee-Nav, starting at two times the LHA radius. The visual homing runs that failed typically got stuck in local minima outside the LHA, close to trees that obstructed the view. Within the LHA, the angular errors of the network were smaller than about 40° and the distance errors were smaller than 2 m (Fig. 2f). Visual homing will generally be successful if angular errors stay below 90°.

We compared the visual homing of Bee-Nav with two alternative approaches (Fig. 2d). First, we compared it with a snapshot-based approach[34] that relies on comparing views around the current position with a single snapshot taken at the home location (red lines). It started to deteriorate as soon as the view became too different from the home snapshot. Second, we devised a 'perfect memory' approach that stored all of the images and target vectors gathered during the learning flight as snapshots. During navigation, the simulated robot compared the current image with all stored snapshots. It selected the $k = 3$ best-matching snapshots and followed the average of their target vectors. The visual homing of Bee-Nav is substantially more efficient and more accurate than the alternative approaches. We expect that the better performance outside the LHA compared with perfect memory is because of the ability of the neural network to attend to relevant landmark objects, even if they are further away and hence smaller in image size. Finally, a comparison was made with a biologically plausible mushroom-body-inspired approach[39]. Although showing great promise, it does not yet achieve similar performance to Bee-Nav (Supplementary Information Section 12).

## Robot experiments

We successfully implemented Bee-Nav on a small flying drone, performing real-world experiments in different environments (Figs. 3 and 4). The drone was fitted with an omnidirectional camera and a Raspberry Pi 4 for executing the small neural networks for visual homing. In our experiments, we used two types of network: a 'compact' 3.4-kB five-layer neural network and a slightly larger, 42.3-kB eight-layer 'attention' neural network (Methods). Because our main interest was to verify the performance of the proposed strategy in various environments, we predominantly trained the neural networks after the learning flight on an Apple MacBook Air ground station laptop (offline, offboard training). For most real-world applications, offboard learning will be acceptable. However, onboard learning would improve the robot's degree of autonomy. Hence, we also tested onboard, offline learning

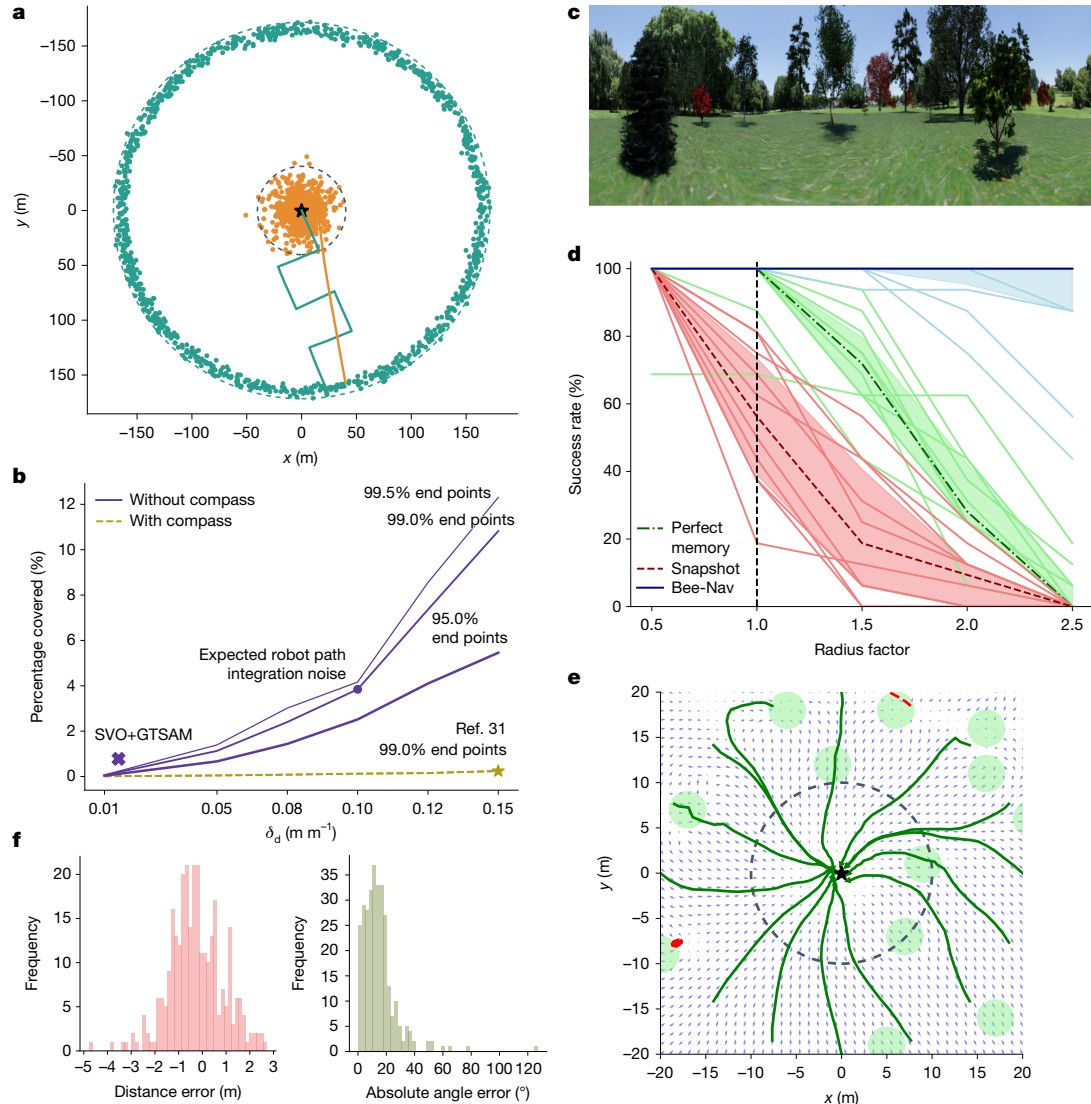

**Fig. 2 | Simulations of homing flights, showing the main characteristics of the proposed strategy in terms of efficiency and effectiveness. a**, Results of 1,000 simulated foraging flights, in which a robot used path integration while first flying a search pattern outbound trajectory and then a straight inbound trajectory home (star). Teal and orange dots represent end points of the outbound and inbound flights, respectively. The dashed teal circle contains 99% of the outbound flight end points, whereas the dashed grey circle contains 99% of the inbound end points. For the expected robot path integration noise, the area of the inner circle is 3.84% of the area of the outer circle. **b**, Effect of path integration noise level on the relative area of the LHA with respect to the total flight area for methods with and without a compass. **c**, Image of the artificially generated landscape used in the simulation learning flight and homing experiments. **d**, Visual homing results in ten different simulated forest environments. The radius factor represents the distance of initial positions from the home location, with factor one corresponding to the edge of the LHA (10 m, dashed vertical line). Light-blue lines show the success percentages of the proposed Bee-Nav strategy per environment, averaged over 16 runs per circle radius, with the dark-blue line indicating the median over all environments. The shading indicates the 25th to 75th percentile. The results in red are for a snapshot-based homing method[34] and those in green for the perfect memory method[51,52]. **e**, Example trajectories starting from a 20-m circle radius. Green and red lines show successful and failed homing trajectories, respectively, light-green circles are tree locations and the small arrows are the predicted home direction. **f**, Histograms of the distance and absolute angular errors for the home vector estimation for the test set images of the LHA of one environment.

and onboard, online learning (that is, after and during the learning flight, respectively).

First, we tested the neural-network-based visual homing in a 10 × 10-m indoor flight arena (Fig. 3a). Despite real-world path integration noise on the learning targets, the drone successfully ended up within 0.5 m of the home location in 100% of the 48 homing flights (Fig. 3b and Supplementary Video 1). Direction errors were predominantly below 40° and distance estimation errors below 1.5 m (Fig. 3c). All manners of learning led to successful homing (Fig. 3b). Notably, onboard, online learning performed slightly better than onboard, offline learning. This is probably because of the different angle resolution when performing data augmentation (Extended Data Table 1).

Still, offline, offboard learning, which uses more learning iterations and data augmentation, gave the most accurate predictions and the shortest homing paths. Hence, we used this way of learning for the remainder of the experiments.

Subsequently, we applied the full navigation scheme to larger environments, using the compact neural network. Time lapses of one of those experiments in a 30 × 40-m indoor hall, with all of the phases of Bee-Nav, show successful homing even under large drift (Fig. 4a and Supplementary Video 2). We made the robot meander during the outbound flight to cover a large distance during this flight phase. The uniformly textured floor posed challenges to visual odometry, occasionally resulting in a substantial displacement after the path-integration-based

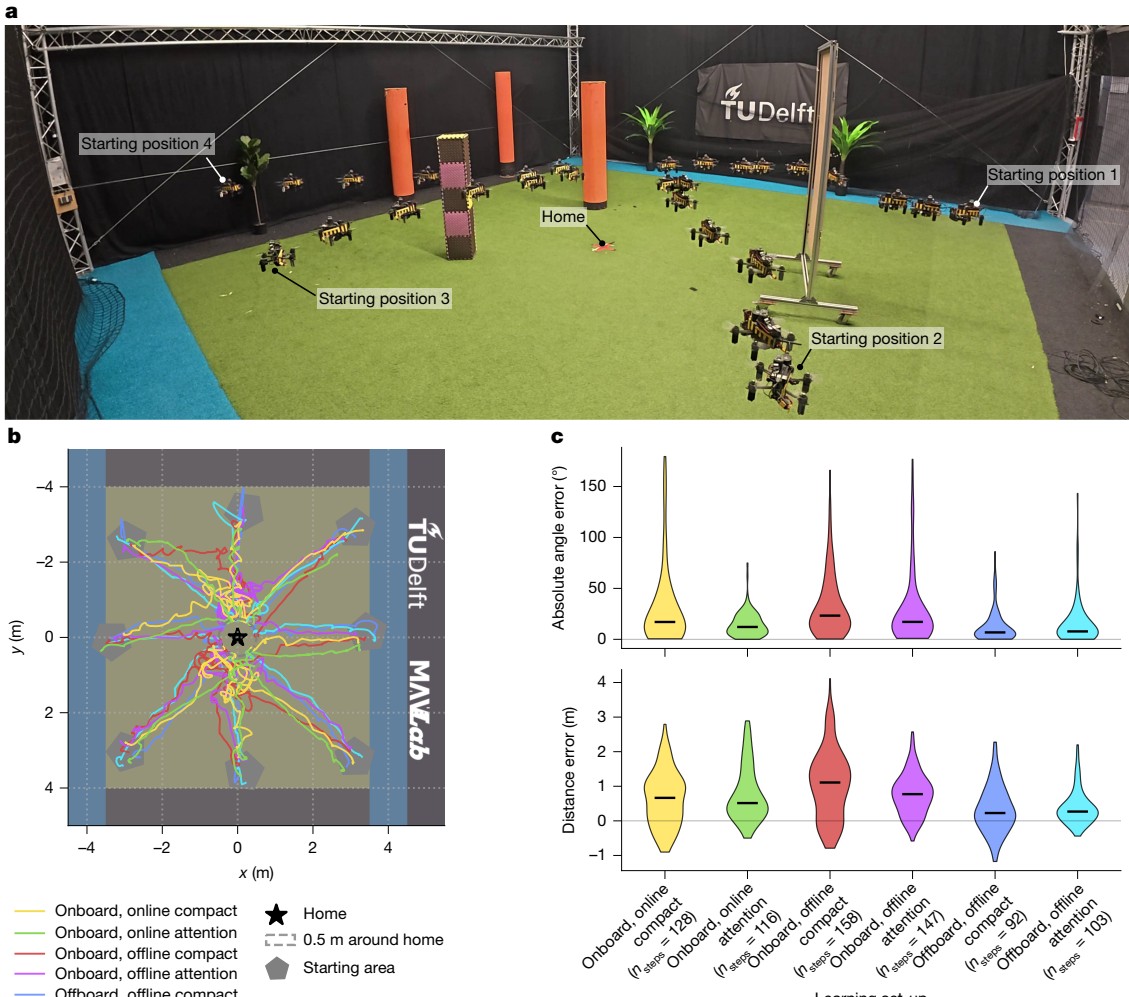

**Fig. 3 | Real-world robotic visual homing flight experiments in TU Delft's 10 × 10-m CyberZoo flight arena, quantifying the performance of the proposed visual homing networks. a**, Time-lapse image of one experimental set-up showing four visual homing flights in a scene containing obstacles, such as screens and poles, inside the LHA. Each flight begins at a different position at the edge of the arena with a varying heading, moving towards home (located at the centre, indicated by the red cross on the ground) using the output from the homing network at each step (Supplementary Video 5). **b**, In a different environment set-up, a top-view plot showing eight visual homing trajectories reaching 0.5-m areas around home (dashed grey lines) starting from different starting locations (grey pentagons). The trajectories are guided by models trained in each of the six learning set-ups: attention network and compact network trained offboard, offline (cyan and blue); attention network and compact network trained onboard, offline (purple and red); and attention and compact network trained onboard, online (green and yellow). **c**, Comparison of distributions of absolute angle prediction and distance prediction errors at each step over all considered homing trajectories ($n_{homing\text{-}flights}$ = 8 per learning set-up).

phase of the inbound flight. The subsequent visual homing effectively cancelled path integration drift.

Moreover, we performed long-range flights at a large, wide-open outdoor test terrain with a permissible flight area of 400 × 500 m (Fig. 4b). This resulted in new challenges. For example, the presence of strong wind led to large compensatory attitude angles, requiring algorithmic adaptations to compensate for the resulting camera tilt (Methods). Other challenges included quickly changing lighting conditions and the absence of close-by landmarks. Hence, at this location, we added several objects to the ground to serve as landmarks (Extended Data Fig. 6f). Because the compact network did not reach sufficient prediction accuracy in this environment, we used the attention network for homing (Supplementary Information Section 13).

The robotic experiments confirm the effectivity and efficiency of Bee-Nav. For most environments, with outbound flight distance in the range 30–110 m, the homing success was 100% (Fig. 4c). Only for the largest outdoor environment, with flight distances in the range 200–330 m, homing success reduced to 80% during days with low wind conditions and 50% in high winds (mean wind speeds $U_{wind}$ > 5 m s$^{-1}$, with wind gusts > 10 m s$^{-1}$; Supplementary Video 3). The reduced success percentage was mainly because of factors such as sun glare and the wind causing the camera to tilt, thus worsening the visual homing accuracy when further away from the learning flight trajectory. Still, in that environment, learning a 10-m-radius area with the 42.3-kB attention neural network sufficed to fly up to 600 m away and come back along an almost straight 150-m path (Supplementary Video 4 and Extended Data Fig. 8f).

## Discussion

We expect our work to have implications for both robotics and biology. For robotics, it brings long-range navigation in reach of small, resource-constrained robots. At present, robotic navigation mainly relies on creating detailed 3D maps of the environment. These maps have as advantages in that they (1) allow for optimal trajectory planning to any place in the environment and (2) may be of interest to the

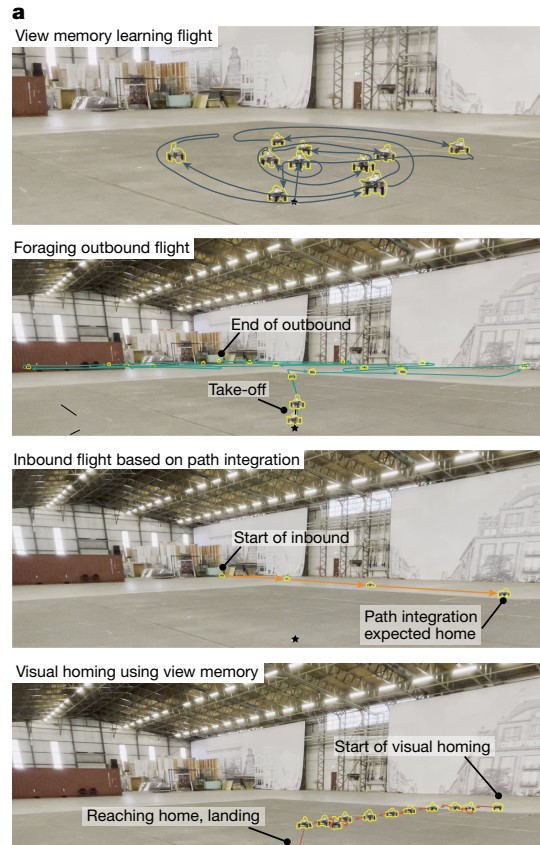

**a** View memory learning flight

Foraging outbound flight
End of outbound
Take-off

Inbound flight based on path integration
Start of inbound
Path integration expected home

Visual homing using view memory
Start of visual homing
Reaching home, landing

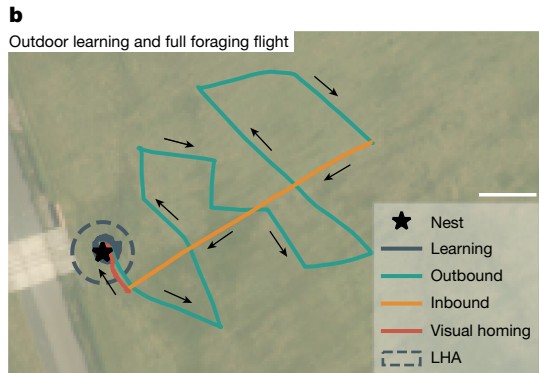

**b** Outdoor learning and full foraging flight

★ Nest
— Learning
— Outbound
— Inbound
— Visual homing
- - - LHA

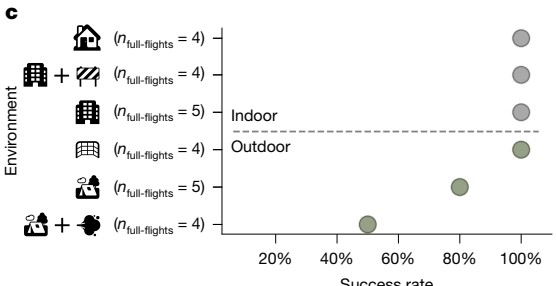

**c**

Environment

🏠 ($n_{\text{full-flights}} = 4$)
🏢 + 🚧 ($n_{\text{full-flights}} = 4$)
🏢 ($n_{\text{full-flights}} = 5$)
🏘 ($n_{\text{full-flights}} = 4$)
🏞 ($n_{\text{full-flights}} = 5$)
🏞 + 🌬 ($n_{\text{full-flights}} = 4$)

Indoor
Outdoor

20%  40%  60%  80%  100%
Success rate

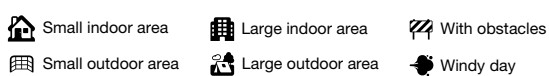

🏠 Small indoor area   🏢 Large indoor area   🚧 With obstacles
🏘 Small outdoor area   🏞 Large outdoor area   🌬 Windy day

**Fig. 4 | Real-world robotics experiments with the full proposed navigation strategy in various environments. a**, Time lapses of all four phases of the proposed navigation strategy performed by the robot flying in a 30 × 40-m indoor hall at Unmanned Valley, Valkenburg (UVV), that is, the large indoor area. **b**, GPS trajectory of an outdoor robotic navigation experiment at the 400 × 500-m UVV testing grounds (the large outdoor area), indicating the different phases of the flight. Scale bar, 50 m. **c**, Homing success percentages in the experimental environments (Extended Data Fig. 6). Satellite basemap imagery provided by Esri World Imagery; sources: Esri, i-cubed, USDA, USGS, AEX, GeoEye, Getmapping, Aerogrid, IGN, IGP, UPR-EGP and the GIS User Community. Available at https://services.arcgisonline.com/ArcGIS/rest/services/World_Imagery/MapServer.

end user. Bee-Nav purposely sacrifices these advantages to substantially reduce computational requirements. The small visual homing neural networks use up to 42.3 kB of memory, which is approximately three orders of magnitude less than the memory required by high-precision maps (hundreds of MBs for modestly sized environments[2,3]; Supplementary Information Section 9). Moreover, the networks are computationally efficient: they easily run on the robot's Raspberry Pi 4 and could even be suitable for microcontroller processors. This implies substantially reduced mass, power and economic cost of computing hardware.

Despite its present limitation to a single home location, Bee-Nav already enables robots to move in an area of interest, perform tasks there and occasionally return home for recharging or bringing back data or objects. This will enable swarms of small, lightweight robots to perform applications such as monitoring crops in greenhouses or tracking inventory in warehouses. For real-world application, the inherent passive safety brought by lightweight robots[44] will be essential, because it ensures the safety of human workers. The computational and memory efficiency of Bee-Nav can also bring benefits for larger autonomous robots, but this will not extend to all robot scales and tasks.

We identify five research avenues to further improve and characterize the approach. First, the visual homing of the robot should be followed by the recognition of the home, which could be a recharging station, and a subsequent precision landing. Second, in the case that visual homing fails, the robot might improve its chances of still arriving home by executing a structured search–just as insects do[28]. Third, path integration in outdoor environments could be improved by including a magnetometer or sun detection system for better heading estimation.

This will substantially extend the navigation range for a given LHA size (Fig. 2b). Fourth, the visual homing networks could output not only a home vector but also a measure of uncertainty. This would enable the robot to start homing immediately when it starts recognizing the LHA, combining information from path integration and view memory[27]. Moreover, the robot could use its uncertainty to trigger a new learning flight if the environment around the home changes substantially, for example, if an important visual landmark is removed.

Fifth, it would be beneficial to perform a deeper investigation of the environments and conditions in which visual homing neural networks can successfully learn to predict home vectors. Given the high capacity of deep neural networks, we expect that they will be able to exploit relevant visual landmark information, even if it is scarce. Nevertheless, it remains an open question how large a deep neural network needs to be to extract the visual homing information in a given environment and for a specific size of the LHA. Our experiments show that very small neural networks (3.4–42.0 kB) may already suffice for LHAs on the order of about 20–120 m$^2$ in different real-world environments. However, there may be environments for which these networks do not have sufficient capacity. A robot would be able to detect this by itself, because it would lead to a high training error. The robot could then automatically scale up the complexity of the network. Still, there are environments with scarce visual landmarks, such as long office corridors with many similar-looking places or large, wide-open terrains. This would also lead to a high training error, but scaling up the network would not improve performance. In real applications, either end users or the robots themselves could choose home locations to ensure the

presence of identifiable, close-by landmarks. Other conditions could have an influence on successful learning, such as opaque or dynamic objects. Dynamic objects that only occupy a small part of the image (such as the experimenter at some distance from the drone) have a negligible influence on the predictions (Supplementary Information Section 8). However, dynamic phenomena that change a large part of the omnidirectional image, such as sun glare, can affect the performance. Dealing with such situations may require modifying the sensor set-up or performing more image processing (for example, masking out dynamic objects). Opaque objects in the LHA do not affect the learning success, as the home vector targets are determined by path integration (Supplementary Video 5, Supplementary Information Section 8 and Fig. 3a).

For biology, the proposed strategy offers a new perspective on flying insect navigation. The robotic experiments have shown that Bee-Nav successfully passes the test of the real world. One of its essential elements is that path integration provides the targets for home-vector learning. In our theoretical analysis, we show that self-supervised learning on the basis of imperfect path integration estimates leads to tortuosity of the visual homing trajectory (Supplementary Information Section 4). We performed a preliminary verification of the potential biological relevance of this finding by means of reanalysing honeybee learning and foraging flight data[4] (Supplementary Information Section 7). It shows that the behaviour of honeybees is characteristic of the considered manner of learning. However, specific experiments will be necessary to uniquely identify path-integration-based learning as the cause of the tortuous trajectories. This could involve manipulating the path integration machinery[45] of the insect during learning flights and evaluating ensuing inbound flight patterns. Another notable element of the strategy is the learning of a full home vector, representing homing direction and distance. Robotic experiments demonstrated the use of the vector's magnitude, allowing speed modulation for faster visual homing at larger distances. Knowledge of home distance aligns with biological observations, both behaviourally[46–48] and neuronally[18].

To enhance the potential relevance to biology, future work could aim for biologically more plausible neural networks. A promising avenue is to enhance the neural networks inspired by the insect mushroom body[32,39] to estimate higher-resolution directions instead of just discerning between left and right. This could be achieved by mapping the Kenyon cells in this method to a larger set of output neurons, potentially representing directions in a similar manner to ring attractors[18]. It could also be worthwhile revisiting known neural data from, for example, the insect mushroom body and investigating whether the neural structure could encode more fine-grained directional data.

Finally, in our current implementation, the robot uses a home vector, expressed in its body reference frame, for direct action. Moreover, it learns only the area around the home location. Hence, the present proposed navigation strategy does not yet involve any cognitive map according to the common definition[49]. In future work, the strategy could be extended to include the learning of different places and the capability to take shortcuts between them. As suggested previously[50], if two different places are stored in memory in a global, home-centred reference frame, a simple vector subtraction would suffice for determining the vector from one place to the other along an unvisited path. Although such an extension to Bee-Nav would be straightforward to implement when done with a traditional algorithm, it will be more challenging to realize this with neural machinery alone. Robotic experiments could show the costs and benefits of such an extended strategy, helping to bring the ingredients and trade-offs of using neural cognitive maps into sharper focus.

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

# Methods

## Simulation experiments
The results in Fig. 2 are based on two simulators: (1) a simplified simulator that was used to determine the ratio of the LHA versus the total navigation area; and (2) a visually realistic simulator that was used to test the visual homing with a convolutional neural network in various generated forest environments.

**Simplified simulator.** In this simulator, we simulate a drone flying outwards from a starting position for a given time period and then trying to return straight home. Reflecting the real experiments, we model the drone as using a noisy gyroscope measurement for updating its heading and a noisy velocity measurement for estimating its travel distance. During both the outbound and the inbound flight, the drone determines its position with the help of path integration. Owing to the noisy nature of the heading update and the velocity estimation, the path-integration-estimated position will increasingly drift from the ground-truth position. We use a Gaussian noise model for path integration drift[22,27,53]. For 'compass-less' path integration, as performed by the real drone, the model adds noise to the heading estimate after each time step in the simulation. To this end, samples are drawn from the normal distribution $\varepsilon_\psi \sim \mathcal{N}(0, t\sigma_\psi^2)$, in which $t$ is the time in seconds. The noise is added to the actual turn rate and accumulates over time, leading to substantial heading drift: $\widehat{\psi} \leftarrow \widehat{\psi} + \dot{\psi} + \varepsilon_\psi$. Furthermore, the velocity estimation noise is likewise drawn from the normal distribution $\varepsilon_d \sim \mathcal{N}(0, d\sigma_d^2)$, in which $d$ is the distance travelled in the simulation time interval. It accumulates in the position estimate of the simulated drone: $\widehat{x} \leftarrow \widehat{x} + \cos(\widehat{\psi})(d + \varepsilon_d)$ and $\widehat{y} \leftarrow \widehat{y} + \sin(\widehat{\psi})(d + \varepsilon_d)$. With this "path integration noise model, the simulated drone performs a 3-min outbound flight with a speed of 2 m s⁻¹. To simulate a search task, the simulated drone flies a block-like search pattern, changing direction after every 40 m. After the outbound flight, the drone attempts to fly straight home with a higher speed of 4 m s⁻¹. Every 0.5 s, the simulated drone changes its heading to point at the presumed home location. The simulation run stops when the drone estimates that it is at (0, 0). The statistics per noise setting are based on 1,000 such simulation runs. For the investigation of different noise levels, both elements of the noise model are scaled with the factors (0.1, 0.25, 0.5, 0.75, 1.0, 1.5). The sizes of the circular LHA and total flight area are then based on the 99th percentile of the distances from the home location, for both the outbound end points and the inbound end points. We also model 'compass-based' path integration, as performed by some robots with a magnetometer[31] and by insects by using sky-light polarization[42]. In that case, at each time step, the simulated drone receives the actual heading perturbed by Gaussian noise: $\widehat{\psi} \leftarrow \psi^* + \varepsilon_\psi$. This leads to much more accurate path integration. For the simulation experiments, we selected noise settings that resulted in a similar drift to that recorded in our own experiments or in the literature (Supplementary Information Section 11). Notably, we selected $\sigma_\psi = 0.63°$ and $\sigma_d = 0.10$ m to best approximate the odometry drift of our own robotic system, which gave a mean yaw drift after 100 s of 5.10° and a mean position drift after 100 m of 0.88 m.

**Visually realistic simulator.** The main goal of this simulator was to determine whether a convolutional neural network is able to reliably estimate home vectors from visually realistic images, using these estimates to travel to the home location within the LHA. The simulator used is NVIDIA's Isaac Sim[54], as it allows quick acquisition of omnidirectional images. We use a photo of a meadow landscape as background image and generate a 'forest' by placing 40 trees from NVIDIA's vegetation library in a 50 × 50-m area around the home location. The trees are placed uniformly into this area, while ensuring that: (1) a 5-m-radius circle around the home is free so that it is accessible; (2) trees are further away from each other than 5 m to avoid intersection; and (3) there are

maximally three trees in a 10-m-radius circle around the home location to avoid too much clutter in the learning area. The experiments reported in Fig. 2 are based on ten such randomly generated forest environments.

*Proposed strategy details.* For the aggregate results in Fig. 2, we performed experiments in ten different environments. In each environment, we first perform a learning flight with the same learning flight pattern followed by the robot, with parameters $n_{loops} = 6$, $m_{points} = 150$ and $b_{spacing} = 0.35$ (see the 'Learning flight' section). As in the real robot experiments, the learning flight positions assumed by the robot are different from the actual positions owing to path integration drift. If a position risks being inside a tree, it is moved to be 0.5 m away from the landmark centre. The learning flight results in 147 images taken in a 10-m-radius area. The noisy positions and headings resulting from the path integration are used for determining the targets for the neural network learning. We train an attention network (explained later) for 150 epochs by using the Adam optimizer with a learning rate of 0.0005 and batch size of 8. We use 95% of the dataset for training and 5% for determining a validation loss. During training, rotational augmentation is used, in which each sample image and target are rotated with the same uniformly random angle from the interval [0, 2π]. After training, we simulate a homing run as follows. An omnidirectional picture is taken at the initial location. It is fed to the neural network, which determines a home vector that has both a direction and a magnitude. The vector is used in the same way as on the robot to determine a step size and direction (see the 'Use of output' section). When within 2 m of a tree, the desired motion vector is adapted with a basic artificial potential field method[55] to veer away from the obstacle. After executing the step, a new image is taken and the procedure is repeated. Each run ends when the simulated drone comes within 1.2 m of the home location or when it has taken 50 steps. For obtaining the aggregated results in Fig. 2, we perform homing runs at 16 locations for different distances, equally spread around a circle. If an initial position is closer to a tree than 1 m, it is moved away from the tree centre to a distance of 1 m. The initial distance is determined by the radius factor, for which 1.0 places the drone at the edge of the LHA. The investigated radius factors are (0.5, 1.0, 1.5, 2.0, 2.5).

*Snapshot method.* We compare the proposed strategy with a snapshot-based method[34]. The motivation for choosing this particular snapshot method is that its performance is robust if the current view is similar enough to the snapshot. For this method, a single snapshot is taken at the home location. The simulated drone performs homing as follows: it first moves to a position to the left, right, front and back of the current position. At each displaced position, an image is taken. This image is exhaustively rotated pixel by pixel. For each rotation, the image difference $D = |I_c - I_s|$ is determined with the home snapshot, in which $I_c$ is the current, rotated image and $I_s$ is the snapshot image. Per displacement, the minimal image difference min($D$) is retained. Having four values for the image difference allows the simulated robot to estimate the best direction to reduce the image difference ($D_x$, $D_y$). The robot then moves 0.5 m in that direction. Although this method requires only a single snapshot at the home location, it does require many movements on the part of the robot. Moreover, exhaustive rotations are computationally expensive.

*Perfect memory method.* We also implemented a method that uses the same learning images as the proposed strategy but now stores them for exhaustive matching during navigation. Hence, this perfect memory method stores 147 'snapshots'. At each time step, the simulated robot compares the current image with all snapshot images, using the same procedure as detailed for the snapshot method—rotating the image pixel by pixel and retaining the minimal image difference (min($D$)) per snapshot. After this exhaustive image matching, the $k = 3$ most similar snapshots are selected and the corresponding target vectors averaged to give the desired motion vector. This method is inspired by two perfect memory methods from the literature[51,52]. Both of these methods are

more efficient than exhaustive matching in different ways. In ref. 52, only four snapshots were chosen out of all snapshot images made on a grid, on the basis of an observation that more snapshots did not substantially improve homing performance. Here we included all snapshots to ensure maximal performance at the cost of more computation. In ref. 51, the navigation was purely based on left and right commands, depending on the home being visible in the left or right visual hemisphere during a real wasp's learning flight. Here we stored real-valued target vectors, that is, with a direction and a distance, for better performance and more adequate comparison with the proposed navigation method.

## Robotic platform

**Robot hardware systems.** The robotic platform is a custom-built quad-copter that operates autonomously, with all perception and computing handled by onboard systems without reliance on external positioning (Extended Data Fig. 1). The drone is equipped with a Raspberry Pi 4 as its primary onboard computer. A Pixhawk 6C Mini flight controller, running the PX4 autopilot firmware, manages low-level flight control. The perception system consists of several key sensors. A Raspberry Pi V2 camera fitted with a catadioptric omnidirectional lens (Kogeto Dot 360 panoramic lens) serves as the main vision sensor. For odometry, a PMW3901 optical flow sensor provides horizontal motion data, whereas a TFmini-S LiDAR rangefinder measures height for vertical positioning. These raw sensor data are combined with inertial measurement unit data by an extended Kalman filter, running on the Pixhawk, to generate the final odometry estimate (position relative to home and current heading). Furthermore, a RealSense D435i depth camera and TF-Nova range sensors are attached for some of the experiments and used only for obstacle avoidance (see the 'Obstacle avoidance' section). Finally, a Holybro F9P GPS module is mounted to record 'ground-truth' position data during outdoor experiments. Other available sensors, such as the magnetometer and barometer, are disabled for all experiments to ensure consistency.

**Onboard software.** The onboard software operates in a modular architecture using Docker containers. The primary vision system resides in a dedicated 'camera container', for which the omnidirectional camera is activated by an HTTP request. Within this container, captured images are first preprocessed using the methods detailed in the 'Image processing' section. Within the same container, the network then performs inference and outputs a 2D vector that represents the predicted coordinates of the home position in the current body reference frame of the drone. This vector is subsequently used to derive the desired yaw angle and distance to home, which are relayed back to the main navigation node for flight control. The 'navigation node' runs in a separate container using ROS2. It receives the output from the neural network as well as current state estimates (position and heading) from the PX4 autopilot[56]. On the basis of this information, it sends high-level control commands back to the PX4. Sensor data are routed directly to the flight controller: optical flow data are transmitted by means of UART and distance sensor data are passed by means of I2C.

## Visual homing model

**Image processing.** A catadioptric omnidirectional camera, which uses a convex mirror and a vertically oriented lens, captures a 360° panoramic view. As can be seen in Extended Data Fig. 3a, the raw output is an annular (ring-shaped) RGB image, in which the central and outer areas contain no useful visual information. A preprocessing pipeline is applied to each image. First, the annular region of interest is isolated using a predefined binary mask. Second, this region is unwrapped into a rectangular panorama through a linear–polar transformation, implemented with the linearPolar function from the OpenCV library. Finally, the resulting image is rotated to align with the body-frame orientation of the drone and resized to $3 \times 192 \times 1{,}800$ pixels, as required by the neural network.

**Data labelling.** The network training uses a self-supervised learning approach in which ground-truth labels are derived directly from the onboard odometry data of the drone. As described in Extended Data Fig. 4, for each captured image, the estimated global position of the drone to home, represented by the vector $\mathbf{p} = \begin{bmatrix} p_x \\ p_y \end{bmatrix}$, and its estimated heading (yaw), $\psi_{\mathrm{drone}}$, are logged. The objective is to calculate a target home vector label, $\mathbf{L}$, which represents the home position in the body-fixed reference frame of the drone. This vector is computed in two steps. First, the vector pointing from the current position of the drone to the home origin (0, 0) in the world frame is determined, which is simply $-\mathbf{p}$. Second, this vector is transformed into the reference frame of the drone by applying a clockwise 2D rotation matrix, $R(-\psi_{\mathrm{drone}})$, which aligns the world frame with the current heading of the drone. The final label vector $\mathbf{L}$ is calculated using the following operation:

$$\mathbf{L} = \begin{bmatrix} L_x \\ L_y \end{bmatrix} = \begin{bmatrix} \cos(\psi_{\mathrm{drone}}) & \sin(\psi_{\mathrm{drone}}) \\ -\sin(\psi_{\mathrm{drone}}) & \cos(\psi_{\mathrm{drone}}) \end{bmatrix} \begin{bmatrix} -p_x \\ -p_y \end{bmatrix}$$

The resulting components, $L_x$ and $L_y$, directly represent the coordinates of the home location relative to the forward-facing perspective of the drone. This vector $\mathbf{L}$ serves as the ground-truth label for training the neural network.

**Wind correction.** Strong wind conditions ($> 5$ m s$^{-1}$ with gust >10 m s$^{-1}$) pose a substantial challenge during outdoor flights, forcing the drone to maintain a large tilt angle—up to 30°—to counteract the aerodynamic forces. This tilt introduces two primary issues in the captured omnidirectional images, which can provide false cues to the neural network. First, if wind conditions cause the tilt of the drone to differ from what it was during the learning phase, the learned visual cues—such as the apparent distance from landmarks to the bottom of the frame—become unreliable. This can lead to a faulty estimation of the true position of the drone, even when it is at the correct spot. Second, it causes a sinusoidal distortion of the horizon line in the unwrapped panoramic image, which in turn corrupts the perceived spatial relationships between landmarks. These problems caused by camera tilt are well known in the omnidirectional vision-based homing literature[57]. To mitigate these effects, we correct the distortion by dynamically adjusting the centre point used in the linearPolar unwrapping process (also see Extended Data Fig. 3d). Two methods are used to determine the necessary offset for this unwrapping centre: (1) model-based correction: a linear model maps the current pitch and roll angles of the drone, obtained from the extended Kalman filter, to the required $x$–$y$ offset for the unwrapping centre and (2) vision-based correction: when a clear horizon is visible, it is detected in the image and used as a geometric reference to calculate the centre point that flattens the sinusoidal distortion. This vision-based method is reminiscent of the method presented in ref. 58. Both methods have their weaknesses, so the choice between these two correction methods is made dynamically on the basis of environmental conditions. The vision-based correction is given priority in environments with a consistently visible ground horizon, as it adapts in real time to the current image and is independent of other sensor measurements that may be subject to delay. In environments in which a stable horizon is not available, the system defaults to the model-based correction, which provides sufficient reliability. Please note that, if the model-based correction relies on attitude estimates based on the inertial measurement unit, it will be susceptible to drift over time. This can deteriorate the results for long trajectories.

**Neural networks.** Two neural network models have been proposed for the visual homing mode: an extremely lightweight convolutional neural network (referred to as the 'compact network') and an only slightly larger attention-based Inception network (referred to as the 'attention network'). The detailed architecture of these two models can be seen

in Extended Data Fig. 2. The compact network is designed for high efficiency, containing four convolutional layers and a final fully connected output layer with two neurons, with a total of only 868 parameters (3.4 kB). The attention network has 10,820 parameters (42.3 kB) and is built on two custom Inception modules[59]. Each module uses parallel branches with different kernel sizes, pooling and dilated convolutions to capture features at several scales. Crucially, each module also incorporates a spatial attention mechanism[60], which generates a map to reweight features and allow the network to focus on the most salient spatial regions of the image. This second, slightly deeper network is composed of the two Inception modules, two extra convolutional layers and two fully connected layers, with weights initialized using the Xavier uniform method. Both models use the tanh activation function and are designed to take panoramic colour images with a size of $3 \times 192 \times 1,800$ pixels as input.

**Use of output.** The output of the network is a 2D home vector $\mathbf{h}_{\mathrm{pred}} = \begin{bmatrix} h_x \\ h_y \end{bmatrix}$, which represents the predicted location of the home position in the body-fixed reference frame of the drone. This vector is directly translated into control commands, as described in Extended Data Fig. 5. The required change in heading, $\Delta\psi$, is determined by the angle of the vector, calculated as $\Delta\psi = \mathrm{atan2}(h_x, h_y)$, in which a positive angle corresponds to a clockwise rotation. The magnitude of the vector, $d_{\mathrm{pred}} = \|\mathbf{h}_{\mathrm{pred}}\|$, determines the step size (distance), $s$, for the next forward movement of the drone according to the linear relationship $s = k d_{\mathrm{pred}} + s_{\mathrm{min}}$, in which $k$ is a scaling factor and $s_{\mathrm{min}}$ is the minimum distance it should take. For our experiments, we use a constant scaling factor of $k = 0.13$. The minimum step size, $s_{\mathrm{min}}$, is set to 0.1 during standard tests and increased to 0.5 in windy conditions to ensure that the drone makes sufficient progress against aerodynamic forces. After rotating to the new heading, the drone moves forward a distance $s$. On reaching this new position, the cycle repeats with the capture of a new image. This strategy ensures larger steps when far from home and smaller, more precise steps near the target to prevent overshoot.

### Experiment set-up

**Environment.** Experiments were conducted across a variety of indoor and outdoor environments to validate the robustness and scalability of the system (Extended Data Fig. 6). Initial algorithm validation (visual learning and homing) was performed in the 10 × 10-m CyberZoo at TU Delft, a controlled lab space with motion capture system as ground truth. Full-scale flight tests were conducted in larger, more challenging locations. To test performance in GPS-denied environments, we used two indoor hangars: the 30 × 40-m Unmanned Valley indoor facility in Valkenburg, the Netherlands (the large indoor area), which provides a large, visually structured area, and a 30 × 25-m section of the Delft Drone Initiative flight hall (the small indoor area). To test against more challenging conditions, outdoor experiments were performed in two distinct open-field environments: the outdoor test field at Unmanned Valley, a 400 × 500-m permissible flight area (the large outdoor area), characterized by natural terrain and lighting in an open field, and a 35 × 20-m tennis court in Sardinia, Italy (the small outdoor area), which provided a visually distinct, closed scene.

**Learning flight.** In a new environment, the drone acquires training data during a learning flight conducted in a small area around the designated home location. Because the learning flight trajectory determines the data for learning the homing vector, it has a substantial influence on the subsequent homing performance. In our experiments, we did not mimic the actual, varying honeybee learning trajectories (Supplementary Information Section 7). Given the proposed technological solution, we opted for a trajectory pattern that ensured covering a circular region around the home location. A comparison of an Archimedean spiral with a 'wasp-like' flight pattern[51,61] showed that the latter resulted in better homing performance, as it does not lead to the learning of spurious cues (Supplementary Information Section 6). Hence, the wasp-like pattern was used for all experiments. This pattern is generated algorithmically, starting with a classical Archimedean spiral defined in polar coordinates by the equation $r = b\theta$, in which $r$ is the radius, $\theta$ is the angle and the parameter $b$ controls the distance between the arms of the spiral. The angle $\theta$ is discretized into $m$ steps over a total of $n$ full rotations, spanning from 0 to $2n\pi$. The key 'wasp-like' modification is a periodic mirroring of the trajectory: each time the path crosses the negative $y$-axis, the $x$-coordinates for all subsequent points are inverted. This creates a series of back-and-forth loops that expand outwards from the centre. The parameters are adjusted on the basis of the environment size; for example, for a small 10 × 10-m area, the path consists of $n_{\mathrm{loops}} = 4$ discretized into $m_{\mathrm{points}} = 36$ waypoints with $b_{\mathrm{spacing}} = 0.1$, creating a learning flight with a maximum radius of approximately 2.5 m. For larger outdoor areas, the pattern is expanded to $n_{\mathrm{loops}} = 5$ loops with $m_{\mathrm{points}} = 36$ waypoints and $b_{\mathrm{spacing}} = 0.2$, resulting in a learning flight with a maximum radius of approximately 6.3 m. Once learning is completed, the network is trained using one of the learning set-ups described below.

**Network learning set-up.** Depending on the mission constraints and resource availability (for example, time, hardware and so on), one of three distinct learning set-ups can be used: offboard, offline learning; onboard, offline learning; and onboard, online learning. Extended Data Table 1 summarizes the computational requirements and performance in one of the experiments for each set-up. Offboard, offline learning is used when a ground station with sufficient processing power (a modern CPU) and data transmission (for example, Wi-Fi, capable of transferring approximately 100 MB of data) is available. On completion of the learning flight, the raw omnidirectional images and associated labels (generated from the specific odometry data; see the 'Data labelling' section) are transferred to a ground station laptop. During the experiments, we used an Apple MacBook Air, with Apple M1 chip and 8 GB unified memory. The network is trained using extensive data augmentation to improve generalization, specifically by means of virtual rotation and colour augmentation (also see Extended Data Fig. 3b,c). For virtual rotation, we make use of the panoramic nature of the omnidirectional images to simulate different camera headings from a single captured frame. This is achieved by horizontally shifting the image pixels to rotate the gaze direction of the camera. For each captured image, we generate 360 new training samples by rotating the gaze in 1° increments. The corresponding label vector is recalculated for each shift to reflect the new relative heading to the home position. Subsequently, colour augmentation is applied to account for variable outdoor lighting. We duplicate each virtually rotated image and adjust its brightness by a factor sampled from U(0.9, 1.1) and its contrast by a value from U(−10, 10). For onboard, offline learning, when a ground station is unavailable or data transmission is unreliable, training is performed locally on the drone's onboard computer (Raspberry Pi 4). To accommodate faster training with the more limited computational resources, we introduce two optimizations compared with the offboard approach. First, colour augmentation is omitted to reduce preprocessing overhead and duplication of the dataset. Second, the virtual rotation step size is increased from 1° to 5°, reducing the training dataset size by a factor of five. Despite the slower onboard processor, these optimizations ensure that the total training time remains comparable with offboard methods while maintaining successful homing performance. For missions requiring immediate execution (for example, outbound flights have to be executed less than 2 min after the learning flight), we use onboard, online learning. In this mode, training initiates in-flight immediately after the first image is captured. We use multithreaded processing on the Raspberry Pi: specific cores are dedicated to navigation and communication, whereas two cores are isolated for network training. Image capture and training happen asynchronously. Captured images are continuously appended

to an image bank (stored as tensors in RAM), which serves as a replay buffer for the training thread. Given the short duration of the learning flight and the small number of images, the memory footprint is manageable, eliminating the need for a deletion policy. The training thread continuously samples batches from this buffer and applies random virtual rotation (sampled from virtual rotation step size of 1°). Training concludes exactly one minute after the learning flight ends; this interval ensures that the final learning images are sufficiently represented in the training distribution. All training configurations use the Adam optimizer with a learning rate of $9 \times 10^{-4}$ and a batch size of 4. The offboard and onboard methods are trained for one epoch, whereas the online learning set-up uses a continuous rolling update strategy. All methods were validated in the CyberZoo environment, achieving a 100% success rate across all visual homing trials. Extended Data Table 1 and Fig. 3b summarize the training configuration and navigation performance for each method.

**Full-flight implementation.** After the learning, the forage part of the navigation strategy is implemented as a three-phase process: an outbound search, an inbound return through odometry and a final visual homing phase. During the outbound phase, the drone follows a predefined trajectory designed to cover the test area. Throughout this phase, the drone continuously updates its state estimate (position and heading) relative to its starting point using onboard odometry. Although simple paths are used for most experiments, more complex patterns, such as a grid search, are also used in some indoor environments to simulate real-world applications such as search and rescue. Once the outbound trajectory is complete, the drone switches to the inbound phase. The flight controller is commanded to return to the home coordinates (0, 0) based on the current odometry estimate. The drone executes a direct, high-speed flight towards this estimated home position. On reaching the odometry-based goal, the final visual homing phase is initiated. Owing to the expected accumulation of odometry drift, the estimated position of the drone does not perfectly align with its true starting location. Therefore, the system switches to the neural-network-based visual control strategy, described previously in the 'Use of output' section, to perform the final, precise approach to the home location. In all of our experiments, the home location is not visible by itself, so we stop the experiments when the drone arrives close enough to the home location. A success is hereby defined as the drone ending up within 0.5 m of the home location. Some example trajectories of these full flights can be seen in Extended Data Figs. 7 and 8.

**Obstacle avoidance.** Inspired by Wedgebug[62], we implemented a reactive detect-and-avoid system based on a finite-state machine, in which the sensing hardware evolved to match the computational constraints of the onboard Raspberry Pi. Initial experiments used an Intel RealSense D435i depth camera with the depth map segmented into left, centre and right average-depth bins, whereas the later experiments used three TF-Nova LiDAR sensors monitoring a 1° × 14° field of view for each sector. The avoidance logic interrupts the primary navigation loop whenever the front distance $d_{front}$ drops below a predefined stopping threshold $d_{stop}$, at which point the evasion trajectory is calculated on the basis of the specific flight phase.

For the learning, outbound and inbound phases, the core avoidance logic is identical. The system monitors the left, centre and right sectors. If the centre is blocked, the drone determines which side (left or right) is clear, rotates in that direction and executes a forward movement of $d_{evade}$. Immediately after this evasion, it realigns its yaw to face the original target waypoint. If all three sectors (left, centre and right) are blocked, the system triggers a panic mode: the drone rotates by a larger angle $\psi_{drone}$, travels a distance of $d_{panic}$ and attempts to resume navigation by realigning to the target waypoint again in this new position. However, if after $n_{attempts}$ the same target waypoint is still not reached, the next movement differs by phase:

- Learning phase: if the point remains unreachable after $n$ tries, the system skips it, takes an image and records the label at the current position and heading and proceeds to the next learning waypoint.
- Outbound phase: if blocked, the system skips to the next outbound waypoint. If the outbound path is completed, it automatically switches to the inbound phase.
- Inbound phase: the system persists in trying to reach the odometry coordinates (0, 0). However, if the drone is stuck but within 2 m of the odometry coordinates (0, 0), it aborts the inbound flight and immediately switches to visual homing.

During the homing phase, the detection logic (checking left/centre/right) and the evasion movement $d_{evade}$ remain the same. However, the post-evasion behaviour is different: the drone does not realign to the previous target. Instead, it simply captures a new image at its current position, queries the network for a new prediction, rotates to this new predicted heading and restarts the obstacle detection and evasion loop again.

## Data availability

The biological reanalysis data are available from the corresponding authors on reasonable request. All robotic experimental data are available at https://doi.org/10.4121/a0217a20-0443-48b2-8a04-9609ac267029.

## Code availability

All codebases (running the simulation experiments, onboard the robot, training and evaluating the networks) that have been used to generate the results in this article are available at https://github.com/tudelft/Bee-Nav.

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

**Acknowledgements** We thank M. Mangan for reading and commenting on an earlier version of this manuscript. We are also grateful to S. Stroobants and E. van der Horst for their extensive help with resolving drone hardware issues, T. J. Singh and the MAVLab outdoor test crew for their assistance with the outdoor experiments and M. Yedutenko and the Delft Drone Initiative for lending their flight hall for the indoor experiments. Part of this project was financed by the Dutch Research Council (NWO) under grant number 20663 of the VICI personal grant programme and grant number NNWA.1292.19.298 of the Dutch Research Agenda (NWA).

**Author contributions** All authors contributed to the conception of the study and the analysis and interpretation of the results. M.V.M.F. performed the early proof-of-concept simulation studies under the guidance of G.C.H.E.d.C. and J.J.H. G.C.H.E.d.C. performed the theoretical and simulation analysis in the Supplementary Information. The visual simulation experiments were developed and performed by M.J., J.J.H., D.O. and G.C.H.E.d.C. The neural networks used for visual learning and homing were developed by D.O., who also built the robot, developed the code, performed the real-world robotic experiments and analysed the data, with support

from J.J.H., G.C.H.E.d.C. and C.D.W. Biological data were provided by J.D. Furthermore, F.T.M. and J.D. provided insights into the biological reanalysis. D.O. performed the reanalysis of the biological data. The manuscript was primarily written by G.C.H.E.d.C. and D.O., and the illustrations were made by D.O., G.C.H.E.d.C. and C.D.W. All authors contributed critically to the drafts and gave final approval for publication.

**Competing interests** The authors declare no competing interests.

**Additional information**
**Correspondence and requests for materials** should be addressed to Guido C. H. E. de Croon.

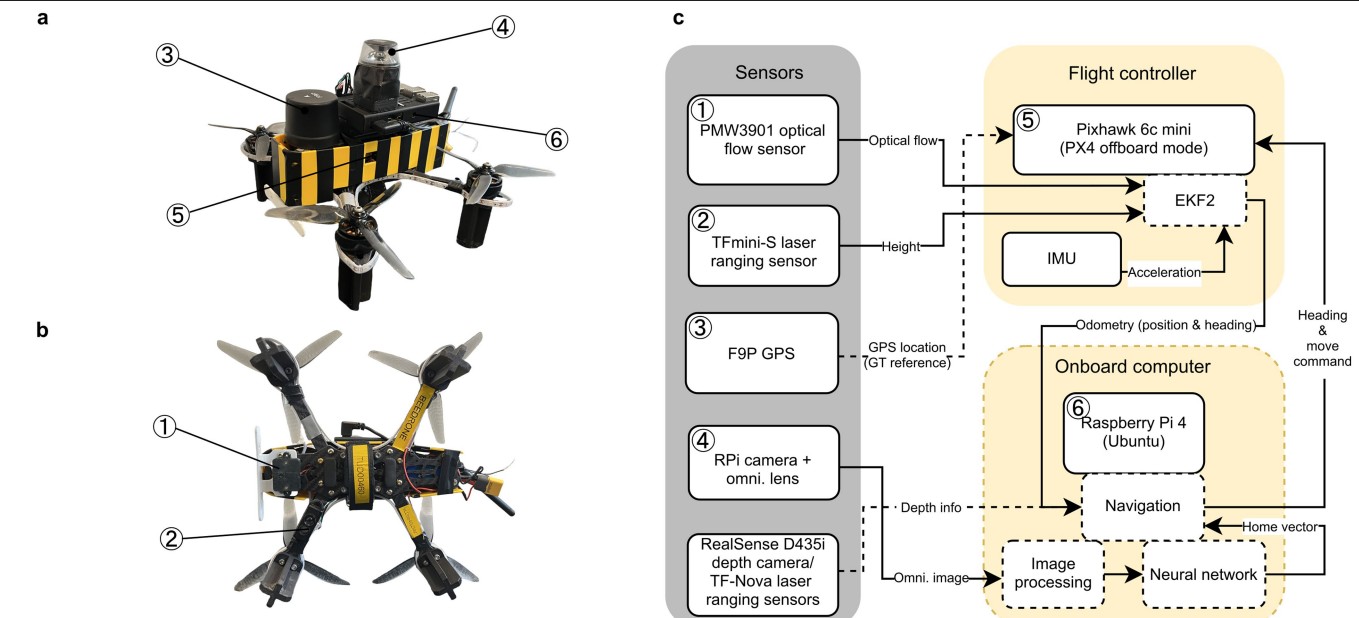

**Extended Data Fig. 1 | Hardware and software architecture of the BeeDrone.** **a**, Isometric view of the drone, highlighting the omnidirectional camera (④), F9P GPS module (③) and the Raspberry Pi 4 onboard computer and Pixhawk 6C Mini flight controller housed in the main body (⑥ and ⑤, respectively). **b**, Bottom view showing the PMW3901 optical flow sensor (①) and the TFmini-S LiDAR sensor (②). **c**, System architecture diagram illustrating the data flow between sensors, the flight controller and the onboard computer.

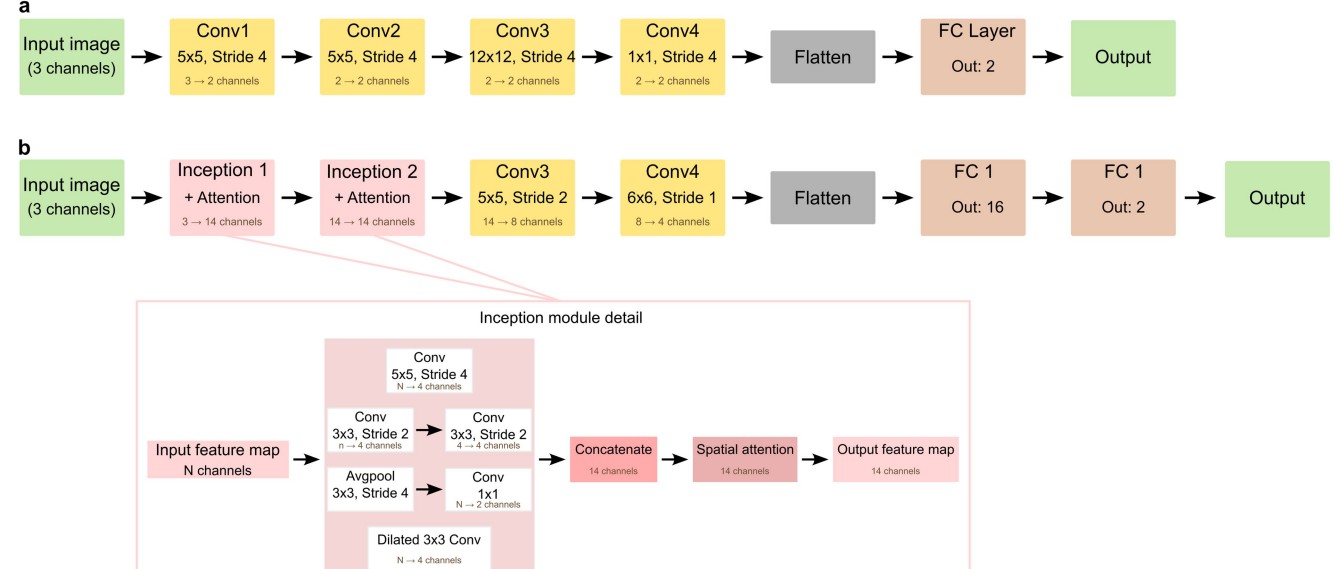

**Extended Data Fig. 2 | Detailed architectures of the neural networks.**
**a**, The lightweight compact network, a sequential model with four convolutional layers and one fully connected layer. The diagram specifies the kernel size, stride and channel depth for each layer. This model, with a total of 868 parameters (3.4 kB), was used in most experimental environments. **b**, The attention network, a deeper model with 10,820 parameters (42.3 kB), was used for the challenging, large and open outdoor environment. The main diagram shows the overall data flow and the inset provides a detailed view of the multibranch Inception module, which includes the spatial attention mechanism.

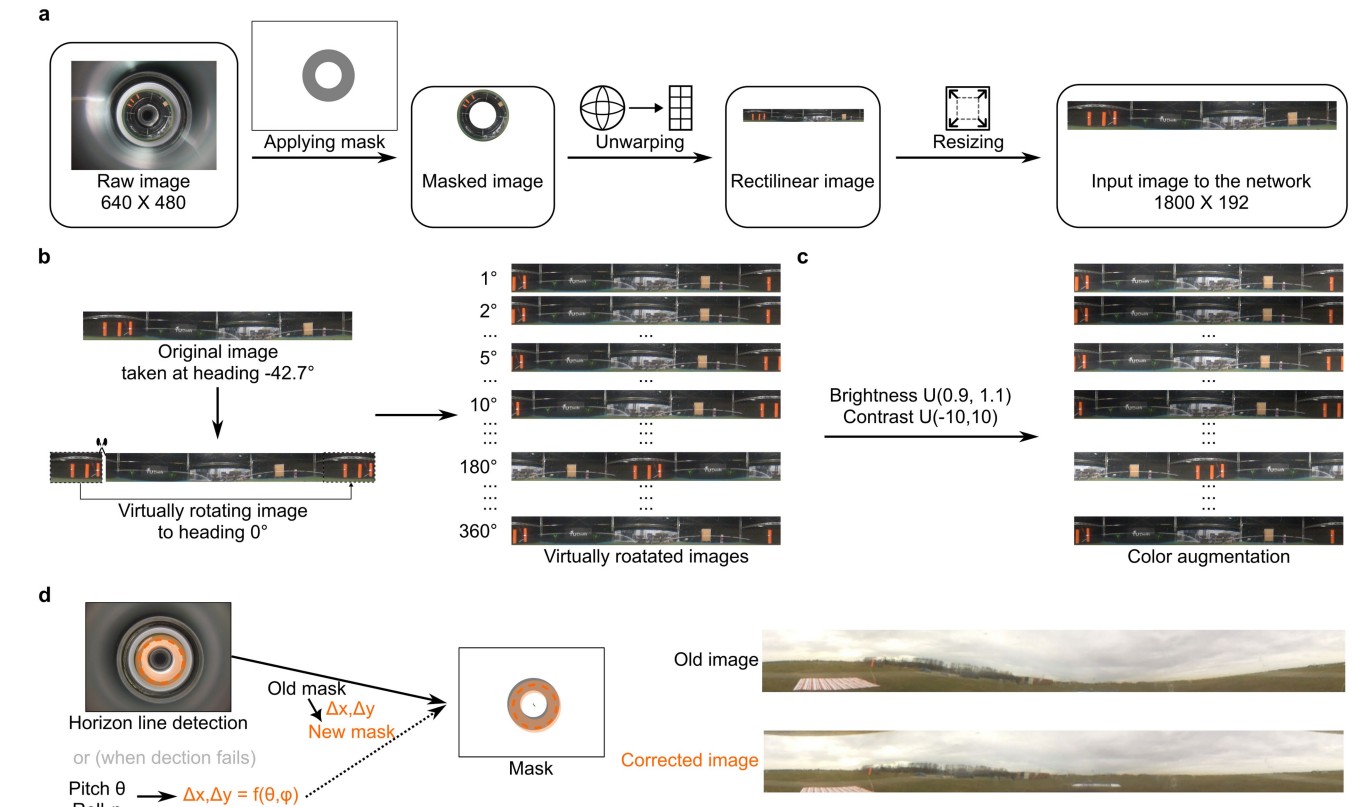

**a**
Raw image 640 X 480 — Applying mask — Masked image — Unwarping — Rectilinear image — Resizing — Input image to the network 1800 X 192

**b**
Original image taken at heading -42.7°
Virtually rotating image to heading 0°
1°
2°
...
5°
...
10°
...
180°
...
360°
Virtually roatated images

**c**
Brightness U(0.9, 1.1)
Contrast U(-10,10)
Color augmentation

**d**
Horizon line detection
or (when dection fails)
Pitch θ
Roll φ → Δx,Δy = f(θ,φ)
Old mask
Δx,Δy
New mask
Mask
Old image
Corrected image

**Extended Data Fig. 3 | Image-processing pipelines in the robotics experiments. a**, The preprocessing pipeline. The raw image taken by the omnidirectional camera is masked, unwrapped and further resized to a format that fits the input size of the neural networks. **b**, Virtual rotation. The preprocessed images are virtually rotated to simulate the view seen when facing different headings, despite only one image being physically taken. This is achieved by cyclically shifting the image content horizontally (moving part of the image from left to right or vice versa). **c**, Colour augmentation.

Each virtually rotated image is duplicated and augmented with a random brightness factor sampled from U(0.9, 1.1) and a contrast factor sampled from U(−10, 10). **d**, Wind correction. When wind affects the drone, the resulting tilt during image capture leads to artefacts in the preprocessed image. This is corrected by adjusting the centre position of the rectilinear unwarping. The new centre is obtained either by detecting the horizon line (dashed orange line) or by a model-based method that calculates the required centre shift using the pitch and roll measured by the drone at the moment of capture.

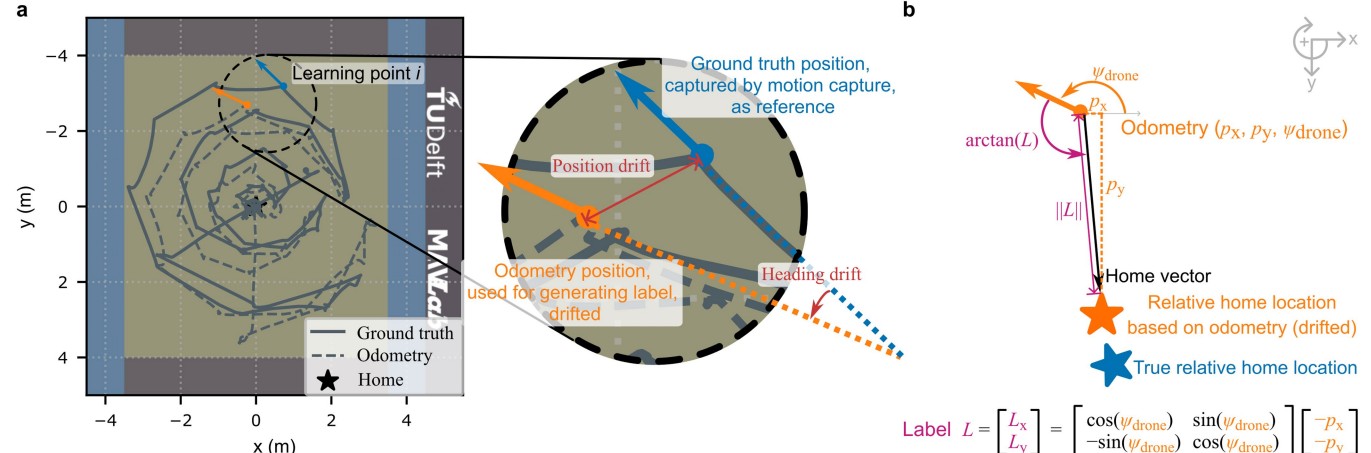

**Extended Data Fig. 4 | Set-up of the self-supervised learning. a**, The learning flight trajectory during one of the CyberZoo experiments. The solid line shows the ground-truth position captured by the OptiTrack motion capture system, serving as a reference. The dashed line shows the odometry position estimation (path integration). The inset highlights one of the learning points in which an image is captured and the corresponding odometry information is used for labelling. The direction of the arrows indicates the heading direction. The drift (red) illustrates the discrepancy between the noisy odometry estimate (orange) and the ground truth (blue). **b**, The odometry information is used to generate the label for the image captured at this location relative to the estimated home position (orange star). Owing to drift, this introduces noise to the label compared with the 'correct' label based on the true relative home location (blue star). The 2D label is generated by performing matrix multiplication using the heading and the current position relative to home. The magnitude of this label represents the distance between the current position and the home location, whereas the arctan of the label represents the desired angle to turn from the current heading to face home.

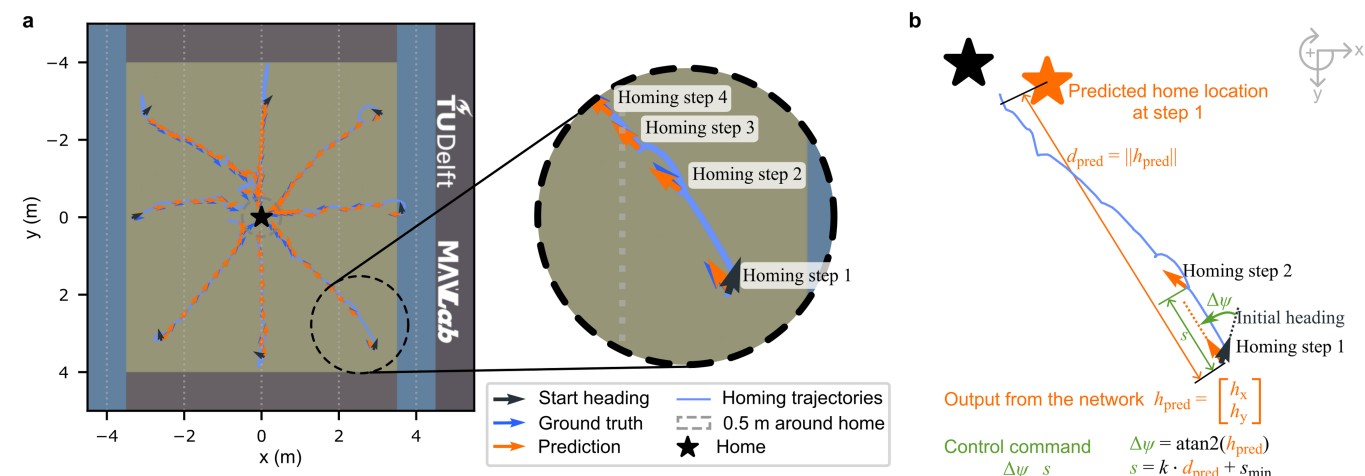

**Extended Data Fig. 5 | Use of network outputs during visual homing.**
**a**, Eight homing trajectories (blue lines) starting at different locations with varying headings (black arrows), reaching home (black star in the middle) using outputs (orange arrows) predicted by the compact network trained offline, offboard. The inset shows the first four steps of one of the homing trajectories. **b**, A detailed description of how the output ($\mathbf{h}_{pred}$) is used to obtain the control command ($\Delta\psi_{drone}$ and $s$). The arctan of the output is used to determine the heading change $\Delta\psi_{drone}$ relative to the current heading to face home. The magnitude of the output predicts the distance from the current position to home and is used to determine the step size $s$ for the drone to move forward along this heading.

**a** 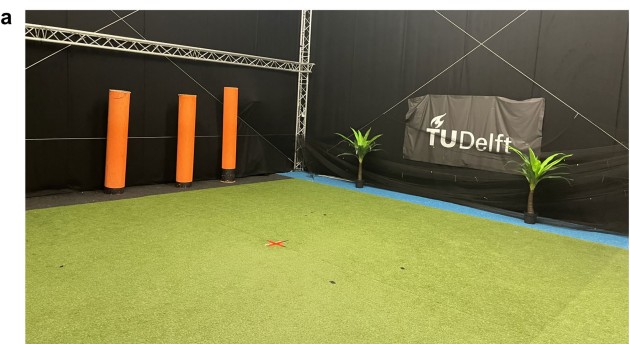

**b** 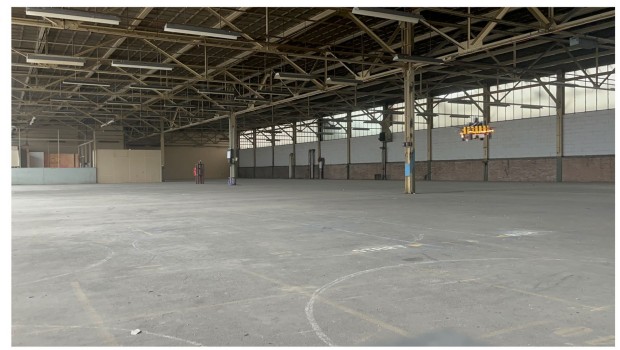

**c** 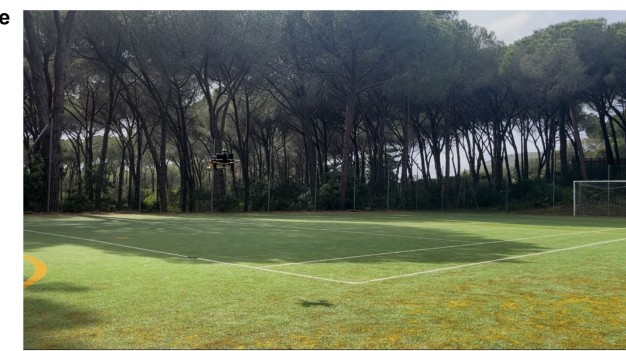

**d** 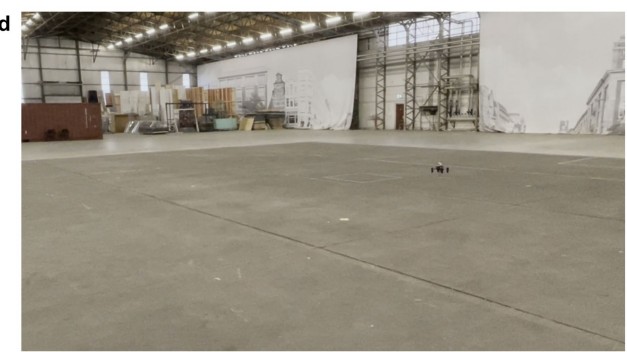

**e** 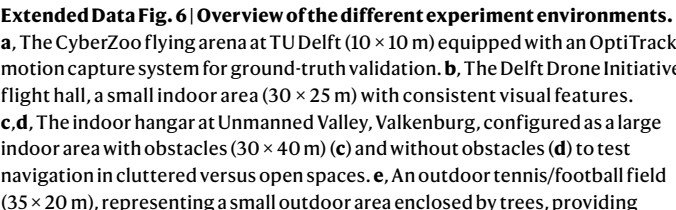

**f** 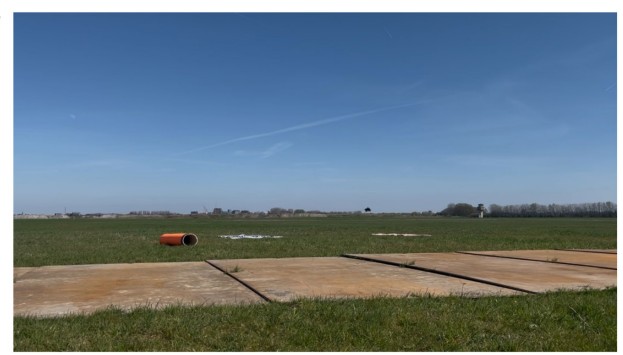

**Extended Data Fig. 6 | Overview of the different experiment environments.**
**a**, The CyberZoo flying arena at TU Delft (10 × 10 m) equipped with an OptiTrack motion capture system for ground-truth validation. **b**, The Delft Drone Initiative flight hall, a small indoor area (30 × 25 m) with consistent visual features. **c,d**, The indoor hangar at Unmanned Valley, Valkenburg, configured as a large indoor area with obstacles (30 × 40 m) (**c**) and without obstacles (**d**) to test navigation in cluttered versus open spaces. **e**, An outdoor tennis/football field (35 × 20 m), representing a small outdoor area enclosed by trees, providing distinct peripheral visual cues. **f**, The open test field at Unmanned Valley, Valkenburg, representing a large outdoor area (400 × 500-m permissible flight area) with natural terrain and a distant horizon, in which accurate GPS ground truth can be obtained. Several objects (a laid-down orange pole and two large features on the ground) were placed on the ground to provide landmarks in the surroundings of the home location. Although the omnidirectional camera could only see these features from some distance owing to the narrow field of view, these objects provided enough information for homing.

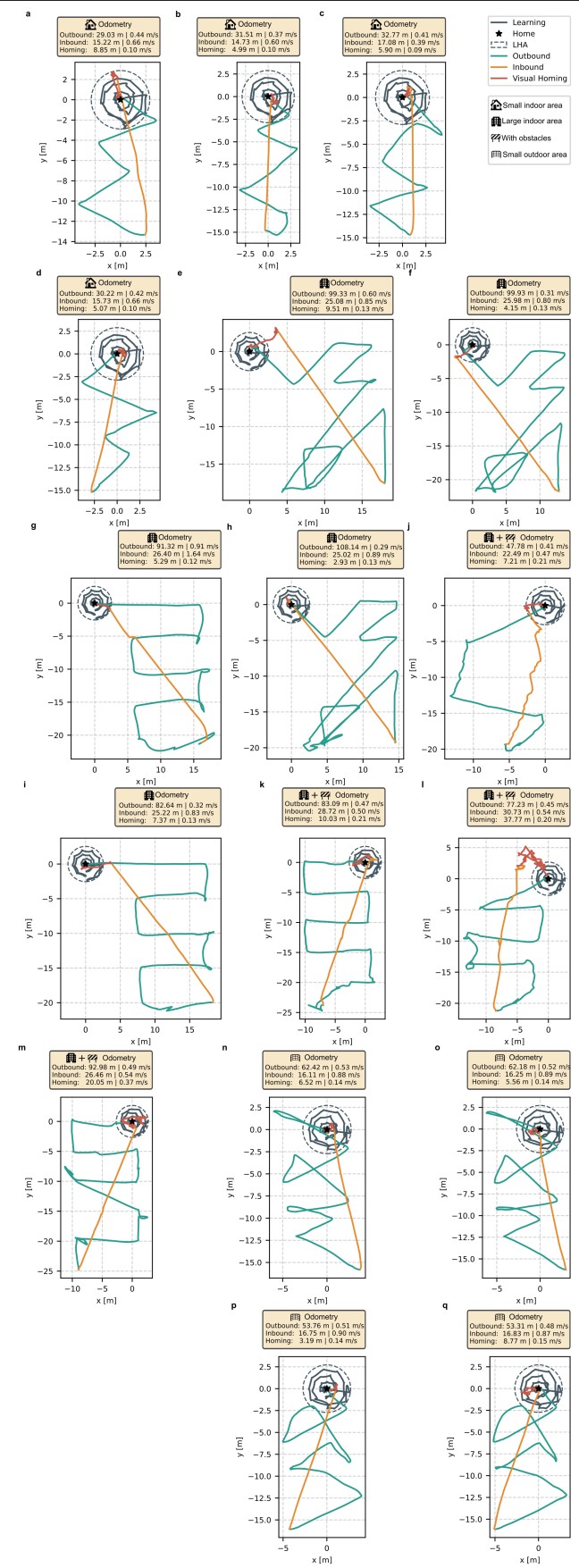

**Extended Data Fig. 7 | Robot learning and full-flight trajectories of the 30–110-m flights.** The figure shows trajectories for each of the 30–110-m flights in four different environments. Each plot shows the robot's learning flight (dark grey), the outbound phase (teal), followed by the inbound phase (orange) and the final visual homing phase (red) of the foraging flight. Trajectories are plotted using onboard odometry. All of the flights successfully ended up within 0.5 m around the home. For visualization purposes, trajectories based on odometry have been globally translated so that their final recorded point aligns with the true home position (0, 0), correcting for accumulated drift to accurately reflect the successful homing. Inset boxes provide quantitative metrics, including total distance and average velocity, for each phase of the flight.

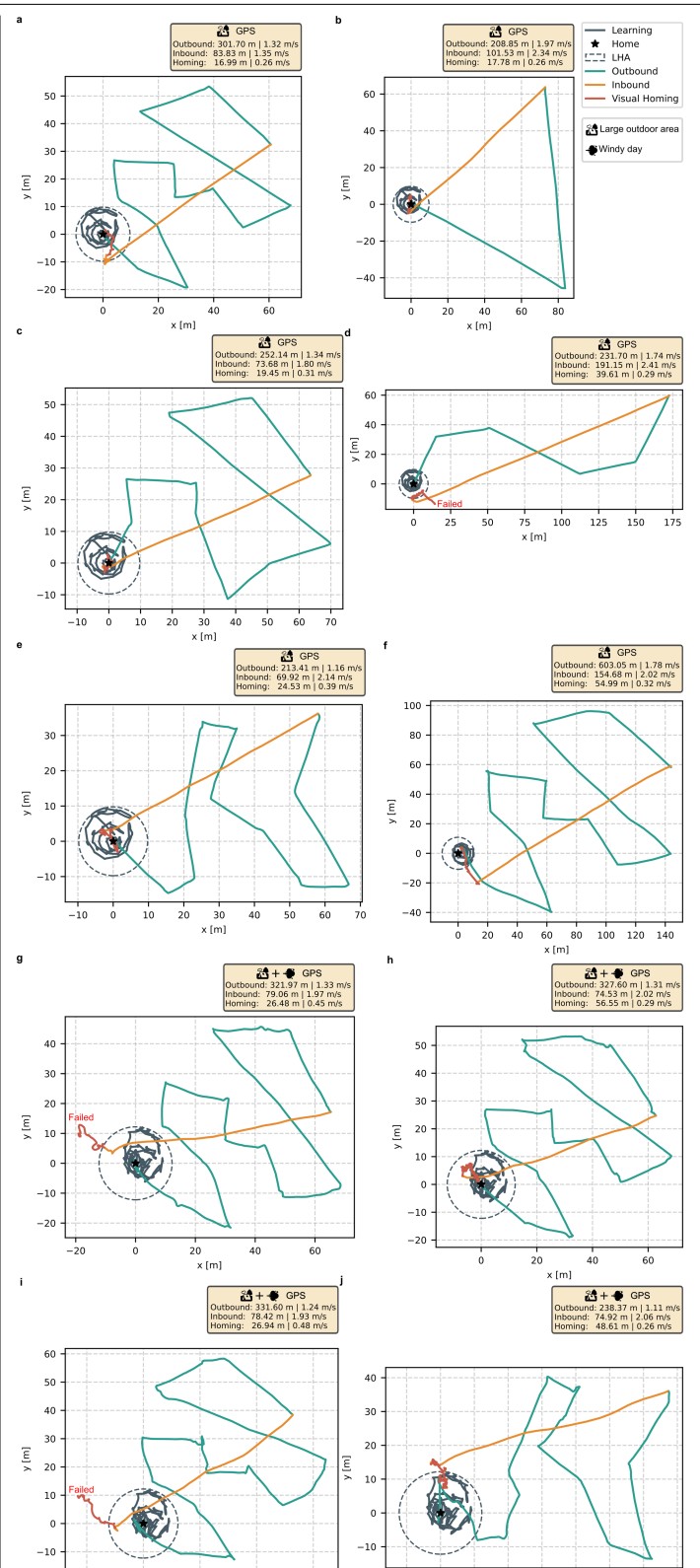

**Extended Data Fig. 8 | Robot learning and full-flight trajectories of the 200–600-m flights.** The figure shows trajectories for each of the 200–600-m flights in the 400 × 500-m Unmanned Valley, Valkenburg test field. Each plot shows the robot's learning flight (dark grey), the outbound phase (teal), followed by the inbound phase (orange) and the final visual homing phase (red) of the foraging flight. Trajectories are plotted using high-precision GPS data. Three flights (marked as 'Failed' in red) did not successfully reach home owing to the challenging conditions in the large outdoor field. The flight in **j** reached 0.5 m within the home area, whereas the GPS data were slightly drifted. Inset boxes provide quantitative metrics, including total distance and average velocity, for each phase of the flight.

**Extended Data Table 1 | Comparison of learning set-up and navigation performance**

| Configuration | | | Training parameters | | | Navigation performance | | | | | |
|---|---|---|---|---|---|---|---|---|---|---|---|
| Strategy | Network | Device | Training time (s) | Color aug. | Rot. step (°) | Inference time (s) | Dir. Err (deg) | Dist. Err (m) | Steps (#) | Total duration (s) | Total dist. (m) |
| Offboard, offline | Attention | Apple M1[a] | 704.9 | Yes | 1 | 0.04 | 15.06±20.91 | 0.39±0.49 | 103 | 348.7 | 46.1 |
| Offboard, offline | Compact | Apple M1[a] | 173.2 | Yes | 1 | 0.004 | 12.28±15.81 | 0.32±0.67 | 92 | 303.2 | 42.9 |
| Onboard, offline | Attention | RPi 4 | 660.3 | No | 5 | 0.04 | 27.72±34.23 | 0.77±0.60 | 147 | 575.1 | 66.8 |
| Onboard, offline | Compact | RPi 4 | 132.2 | No | 5 | 0.004 | 37.45±30.48 | 1.05±0.98 | 158 | 544.2 | 66.9 |
| Onboard, online | Attention | RPi 4[b] | 320~[c] | No | 1[d] | 0.04 | 14.50±11.94 | 0.79±0.79 | 116 | 359.9 | 52.1 |
| Onboard, online | Compact | RPi 4[b] | 320~[c] | No | 1[d] | 0.004 | 31.87±39.51 | 0.65±0.79 | 128 | 490.5 | 63.2 |

[a] MacBook Air (2020) configuration with Apple M1 chip and 8GB Unified Memory.
[b] Training is isolated to 2 specific cores out of the 4 available on the Raspberry Pi 4 BCM2711 SoC.
[c] Approximation includes the duration of the learning flight plus a 1-minute post-flight buffer.
[d] Indicates the discretization step for random sampling; due to the rolling nature of online training, the network may not encounter the full 360° distribution.

The table details the six different learning set-ups and training configurations. These range from training a lightweight compact network onboard a Raspberry Pi 4 online during flight, to training a slightly larger attention network offboard on a MacBook M1 offline after the flight. Each learning set-up features varying training times and data augmentation methods. All tests were conducted in the same environment. The networks were trained using data from the same flight, with the exception of the onboard, online attention set-up, which required a separate but identical flight to facilitate another flight of online learning. Performance was evaluated by conducting eight different homing flights from various directions at the edge of the LHA to achieve 0.5-m area around home, starting with random initial headings. The evaluation metrics used to compare the set-ups include inference time, prediction errors (both absolute angle and distance), number of steps, flight duration and the total distance covered across the eight homing flights.