## [Peer Review file · Nature]

Efficient robot navigation inspired by honeybee learning flights.

Corresponding Author: Professor Guido de Croon

Version 1:

Reviewer comments:

Referee #1

(Remarks to the Author)

This paper describes an efficient navigation strategy for flying robots inspired by insect visual memory. The algorithm is interesting but not strikingly original: images in a small area around the home location are associated (by training a neural net using the images and the ongoing odometry estimate) to a vector to the home location. This is coupled with odometry (path integration) to allow much larger excursions to be made, provided the accumulated odometry error still returns the robot to the area in which visual learning took place. The algorithm has already been presented in ref. 37, with the main advance here being demonstration of its deployment for successful navigation on real flying robots tested in several indoor and outdoor environments, a substantial effort and important demonstration of efficacy. It is also usefully shown that the error induced by using noisy odometry estimates during learning is still sufficient for navigating home, i.e., the information available purely from the robot's own viewpoint is sufficient for the task.

However, I have two main reservations about the contributions of this paper:

1) Although it is repeatedly stressed that the presented strategy is highly efficient, using a "tiny network", hence suitable for resource-constrained robots such as small flying drones, the key process of learning to associate views to vectors depends on offline training of the network, and the computational demand this requires is unclear. The reader actually has to work quite hard to discover this limitation. The first description of the strategy reads: "...a robot performs a learning flight (Figure 1a), during which it trains a small on-board neural network to map its visual inputs directly to a home vector". One might reasonably infer this is on-board, online learning. The caption to fig 1a specifies that the training of the neural net occurs after all the images are captured (so not online) but not that this is offboard. One has to delve into the methods to read (in the section "Neural networks") that the networks are trained offline on a ground station laptop (not further specified) using data augmentation, and in the section "Learning flight" that the training on this unknown specification laptop takes 2-3 minutes for the simple network and 30 minutes for the more complex network needed for the real world scenarios.

This issue is particularly relevant because the authors do not provide sufficient discussion of or comparison to 'conventional' map-based navigation, simply dismissing this as something for which lightweight drones "can simply not carry or power the required computational systems", providing only one reference to such systems, a survey paper from 2018. However, there is substantial recent research into 'lightweight' SLAM algorithms and alternative robotic mapping approaches. Moreover, the problem solved here is different; many applications (e.g. surveying) require that a map is produced or that the robot's position is continuously estimated to a precision better than odometry allows. It is reasonable to present a solution for the more constrained problem of returning to a known location, which indeed does not necessitate constructing a map. But comparison of computational demands should be on a level playing field, i.e., for solving the same task.

2) The most relevant 'insect-inspired' algorithm to which the current work can be compared is the use of 'familiarity'-learning of views towards the nest, collected during learning walks or flights, e.g. doi.org/10.1016/j.cub.2015.12.052, doi.org/10.1177/1059712313516132, [arXiv:2507.09725](https://arxiv.org/abs/2507.09725)). Notably, this approach uses PI to obtain the nest-directed viewpoint, or the relation of the current view to the nest direction, so implicitly learns the association of views to the home vector direction. The latter paper is briefly mentioned here, but somewhat dismissed for using GPS to estimate the location of

the robot relative to home, although it would clearly have been straightforward for that work to have substituted onboard odometry for GPS. The learning in this algorithm is extremely simple (could be executed online and onboard) and has been plausibly mapped to insect brain circuits. It would certainly seem that this algorithm would be a much more appropriate comparison than the snapshot algorithm used for comparison here (fig 2d) as it seems self evident that any algorithm storing only one image/location will under-perform compared to an algorithm trained on 100+ image/location pairs. Additionally, many points made in the discussion would apply equally to the familiarity algorithm, e.g., it also assumes path integration is required for learning (to structure the viewpoints stored).

Further comments:

The abstract highlights that the robot only needs to learn small area (0.5%) of the total flight range. However, this result depends entirely on how noisy the odometry is assumed to be in simulation. In the methods, the authors provide some detail on how the odometry noise was modelled, and note that this is conservative (i.e. more noisy) with respect to prior work with real drones. However, some brief explanation of the noise model is needed in the main text, as well as reference to how well it reflects real drones and also real insects. This is also needed for fig 2b, e.g. it is impossible to understand what a 'noise level', as shown on the x-axis, means.

The review of previous insect-inspired strategies seems a little unbalanced. E.g. research on route-following has focused on view memory alone because the behavior of interest was the ability of insects to travel home along familiar routes after displacement, i.e., when path integration is zero or conflicting, and in cluttered environments. It seems unfair to imply earlier researchers have 'missed' the obvious advantages of combining view memory and path integration, and that this paper has made a breakthrough by combining them.

In general, the experimental descriptions could benefit from a little more detail in the main text, as it is hard to get a clear idea of the implementation. E.g. that the visual input is omnidirectional, unwrapped, visually rotated; that the odometry is based on optic flow and IMU; how the recovered vector is used to control movement, etc. Some but not all this information is in figure 1.

How important is the learning flight trajectory? This is described as bee-like in the text but does not obviously resemble the learning flights shown in extended data fig 4. In the methods it is described as wasp-like.

(Remarks on code availability)

Referee #2

(Remarks to the Author)

This is interesting research in the area of lightweight robot navigation systems, building on a substantive body of work in the general area by the authors, who are one of the leading groups in this broad area internationally, and in this specific domain (lightweight drone-specific visual navigation) arguably the leading group. The capability demonstrations are impressive and as far as I know firsts in the specific area of lightweight visual out and back robot navigation in terms of scale and size.

We break down our review into several components:

Novelty and delta over previous work

The authors have a range of prior work such as their 2024 Science Robotics paper, "Visual route following for tiny autonomous robots". The authors are primarily "competing" against themselves here, in terms of their string of relatively recently publications in the general area. The work described in their 2024 paper already broadly described an out and back lightweight visual navigation system for drones. In terms of additions here: there is a substantially different approach, especially around the local homing area and associated training, the comparison to bee flights, and the general sophistication of the approach. The scale and challenge of the experimental demonstrations is much larger and more impressive.

Practical Relevance and Utility

The idea of widely deployed small robots as detailed in the article has long been a dream of many roboticists: it has not yet happened in any enduring capability, and it's telling that new enduring widespread deployments of any largely autonomous robots (large or small) has been extremely limited over the past few decades, with a few notable exceptions like logistics robots in warehouses, and possibly autonomous robotaxis as they scale up. That is to say, it's a fiendishly difficult problem, lots of advances are needed, this research makes a valid contribution to some of the challenges but there are so many general challenges that at this stage this is technology that is still likely a long way off widespread deployment beyond proof-of-concepts and pilot studies. This comment is more to set the context in which this work occurs. There is a positive here - being by no means a solved problem - the value of research like this is more significant than if this was a mature field.

From a pragmatic perspective - how much could this work (eventually) revolutionize aspects of robotics navigation in commercial and deployed applications - the biological inspiration is largely irrelevant except indirectly as to desirable

properties like lightweight compute etc...

In this regard, there are some clarifications that are needed:

The rationale around the benefits of this lightweight system need more detail and more specifics. How much does only having 40 kB of storage versus 100s to 1000s of MB (a space inefficient SD card has been able to store this amount of data for many years) matter practically. Obviously less is always better, all other things being equal, but how significant is this specific memory claim in practical terms of how a complete system (which will have other requirements and capabilities) perform - especially if some of the capabilities require more compute / storage anyway. Or alternatively, what is the class of real-world problems where that wouldn't be the case?

I am surprised that the claim focused in significant part around storage - I would assume the primary practical benefit here is around the amount of data / training / compute / energy consumption that is required to train the larger / more memory intensive systems - rather than the memory itself, per se - if that impression is wrong, some detailed cost / bulk / energy consumption statistics for the lower memory footprint itself would be helpful in making the case.

There is a significant treatment of wind, and the tilt effects and compensating for these. As far as I can determine, this is a pure engineering-based approach, to deal with a practical real world issue. There is a range of related work in area like high performance drones (e.g. work out of Scaramuzza group, the work that Opteran does for its main sensor) so this is largely a needed capability enhancement but not particularly interesting from a research perspective. It's needed practically, and a practical contribution, but I would evaluate those components as not particular novel or state-of-the-art compared to other work in the field, including in commercially deployed drones.

Similar comment about "The uniformly textured floor posed challenges to visual odometry" - this is not a problem for SOTA VO methods, is this a limitation of this specific method's shortcomings? (compute, sensor, ???). Again, VO is not a claimed contribution here, this is more about understanding what is being used and what its limitations are compared to other more sophisticated systems (that might use a lot more compute).

Indoors versus outdoors - there is some commentary here: how much of this commentary isn't about absolute differences in indoor / outdoor environments, and how much is just conditions of wind, lighting e.g. there are likely indoor environments (flickering lighting, strong internal ventilation, highly aliased visuals) that this system would struggle in more than certain outdoor environments (no wind, unique and abundant landmarks). I would suggest phrasing this less as indoor versus outdoor, and more about the properties of the environment that make it easier or more challenging.

Approach

The approach essentially boils down to: make a path integration system that doesn't drift too much, go out on an arbitrary route whilst path integrating, then beeline straight back to the start location, hope that the drift isn't too large to end up in the LHA (Learned Homing Area), then visually home in on the starting location. That is still an impressive achievement, especially as demonstrated in this domain / with actual drones indoors and out, but the core task is fairly straightforward to describe.

There are multiple aspects to this:

Path integration that doesn't drift too much: this has been done extensively in drones and autonomous vehicles, and here it's either abstractly simulated in simulation or implemented on the drone.

The LHA approach is neat, but it will not work in a large variety of environments. That does not negate the potential value and impact of this work, but the wide range of assumptions that the LHA approach makes about the environment in order to be functional need to be detailed. What would happen in an area with inter-meshed with opaque barriers / walls? I assume it will not work in a completely visually ambiguous environment? Highly dynamic environments are also likely an issue. What are the exact properties of the LHA area in order for the method to likely be performant? Sometimes this could be more convincingly answered, albeit indirectly, by, "we ran 1000s of real-world trials" (which is very difficult to do) but that is not the case here given a small number of (real world) trials (which are hard).

The proposed system uses two different networks for small and large environments. There is a rationale provided - but rather than tweaking for a specific environment, some treatment of how this could be generalized as a process would help. Demonstrating the performance of the small network in the large environment (rather than just describing) and more importantly the performance of the large network in the small environment would also be interesting here and add more depth to the results.

Further explanation for how the Attention-based Inception Network predicts the home vector in the featureless outdoor environment would be helpful.

In the simulated environment, the manuscript should explain how the 40 trees were differentiated from one another within the network, or if they were considered as similar looking, how they were placed to avoid failure.

Biological Relevance

The authors make an effort to relate the behaviour of their proposed system to natural systems, with some limited statistical analysis. The claims here seem to distill down to a comparison in the property tortuosity between the artificial system and bees - more tortuous within LHA, straighter outside, as well as a relatively velocity match: faster during path integration, slower during view memory-based navigation. There is a prediction of the extrapolation properties of the view-based navigation memory outside the learn homing area - arriving at a 30% extrapolation capability.

How strong / relevant is this to biology? The proposed system is undoubtedly at a high level interesting in terms of the breakdown between path integration and local homing, which is the proposed navigation strategy for bees and other flying insects. In terms of the specifics and predictions - this is perhaps not that convincing (yet) - this analysis shows that the artificial system is broadly similar to the natural system, and makes an (untested) prediction around extrapolation beyond the LHA. Regardless, it's likely to be of interest to both roboticists and biologists.

Writing and Claims

This is probably the biggest issue with the paper as it currently stands - it appears to be making some pretty strong claims, many of which are either arguably overblown or more a natural byproduct of the assumptions the system is built on. I think toned down and more specific versions of these claims would still make it worth of consideration for Nature and also more appropriate what has actually been demonstrated.

"Simulation experiments showed that the neural network only requires training on ~0.5% of the total flight area" - this claim is correct but is essentially a very "obvious" product of two components: an assumption that the path integration system doesn't drift too much, and hence path integration back into or near the LHA is achievable, and that the LHA homing works. I think a more appropriate claim might be something like, "when path integration can be relied upon to get the system back into the learnt homing area, that learnt homing area only needs to be X% of the total navigation range" or some similar claim. And then the authors' analysis of the ratio of the two depending on various noise factors becomes more important too. The authors make a contribution in providing systems that meet these two requirements - no small feat, with the novelty relative to the field being more I think about the learning homing area than the path integration.

Some of the language could be tightened up e.g. "randomized, tortuous outbound trajectories" - it's implied this is bad, but is it actually bad / a hindrance / extra challenge for this system, in these experiments? Can the authors be more specific / clarify here.

General / Presentation

In the simulation experiments section, the text appears to reference Figure 2e instead of Figure 2f.

In Figure 2d, the last two x tick labels are overlapped, which should be placed one after another.

In Figure 3f, the data points should not be connected horizontally into a line as the x labels are the environment types and even though they are placed in increasing difficulty order, they have no correlation.

The extended data figures 4 and 5 do not appear to be cited in the manuscript.

In Figure SI-13, the order of the figures for the spiral and wasp-like flight patterns in the two environments is spiral and then wasp-like, but the figure caption and its description in SI-6 section for figures SI-13d and e use the reverse order when referring to their quantitative results from the plots.

Some figures such as Figure 3, all Extended data figures 1-6, and all figures SI 1-13 have images which have been rendered with low quality, which makes it hard to read.

Supplemental

Contains a large amount of relatively theoretical / example-based analysis of various homing conditions, with visually unique and non-unique landmarks, of varying numbers.

"similarly looking landmarks" - I find this phrasing a bit awkward - "similar appearance landmarks"?

Videos are good additions.

(Remarks on code availability)

I've (as an experienced researcher who does plenty of coding in the robotics field) examined the provided source code, configurations, and data. The following points summarise my findings regarding clarity, documentation, and the ability to run the code and replicate the manuscript's findings.

Code Execution and Replication

- The environment installation and execution instructions for the theoretical simulation analysis worked correctly, allowing for the successful replication of the experiments.
- The robot offboard training and testing module could also be successfully run.
- Unfortunately, it was not possible to run the robot onboard experiments and the visual simulator's simple experiments. The system configurations in the visual simulator's `run_simple_experiments.py` use hardcoded local paths within the `.json` files. Even after attempts to modify these paths, errors related to finding the image folders persisted, making it impossible to

determine the root cause and execute the experiments.

Suggestions for Improved Documentation and Structure

- System inputs, outputs, and hardware: the README files for all modules should clearly detail the expected inputs and outputs for each system and explicitly state the hardware requirements needed to run each module. This is particularly important for clarifying which experiments can be run with the provided data and which require additional hardware (like the onboard system).
- Data usage clarity: although the input data is provided separately, the repository's README files and the data folder's overview should clearly map which specific dataset within the provided data is necessary for each experiment (e.g. for the visual simulator experiments). This is because not all datasets contain the required data types for every system.
- Output location: the README files for all modules should also clearly explain what the expected outputs are and where the resulting files are stored.
- Environment installation: providing instructions to use an environment manager like Conda or pixi for the theoretical simulation and robot offboard/onboard systems could significantly streamline the installation process.
- Visual simulator (IsaacSim): the README's linked documentation for the recommended IsaacSim version (4.2.0) is broken. While the initial environment setup worked on version 4.5.0 (with minor import modifications), the instructions should include all additional Python packages required for the IsaacSim environment to run the visual simulator experiments.
- Visual simulator (IsaacSim): the reliance on hardcoded local paths in the configuration files, particularly for finding image data, should be resolved to ensure the experiments are runnable on other systems.
- Code organisation: The source code's structure could be improved for easier navigation. Sorting different file types, such as putting all .json configuration files into a dedicated configurations subfolder, would make it clearer to find the scripts necessary for execution.

Referee #3

(Remarks to the Author)

I co-reviewed this manuscript with one of the reviewers who provided the listed reports.

(Remarks on code availability)

Version 2:

Reviewer comments:

Referee #1

(Remarks to the Author)

Overall, this is a very thorough evaluation of the navigational capabilities that can be obtained for a flying robot by combining larger scale path integration with compact neural networks that associate path integration state to visual input in the nearer vicinity of the target; which might also be a plausible model for insect behaviour. I'm not convinced this is a major conceptual breakthrough for either robotics or biology but it is very interesting work, and likely to have significant impact.

I greatly appreciate the substantial effort by the authors to respond to the points raised in my previous review, specifically:

- 1) Clearer contextualisation of how the approach relates to state of the art navigation approaches in small flying robots.
- 2) Direct evaluation of the extent to which on-line/on-board learning is plausible for the approach.
- 3) Extended comparison (in simulation) of alternative insect-inspired visual localisation methods.
- 4) A more thorough exploration of the path integration noise assumptions.

There are still some points that should be addressed:

Abstract, lines 18-20: "in real-world experiments, 18 a tiny 42-kB neural network sufficed for a small drone to fly outbound trajectories up to 600 meters and return using direct inbound trajectories shorter than 150 meters". This is using offline, offboard training, and appears to refer to a single trial using a larger radius learning flight than the main data reported in the paper (as described in the results on line 175-177, and provided as a video supplement, but not otherwise mentioned in the methods or figures). The *directness* of the inbound trajectory (150m vs 600m) seems somewhat irrelevant to highlight, as this is largely due to using rather standard odometry to convert a convoluted outward path to a straightline direction home. What actually matters is repeatable accuracy in pinpointing the home location, and my understanding of the key results is that for flights of a similar range (100s of meters), the success rate in reaching the desired level of accuracy was 6/9 (fig 3) - maybe 7/10 if this extra 600m trial is included? A more representative 'headline result' should be provided in the abstract, e.g., "100% success in returning within 0.5m of home for real-world indoor and outdoor flights of 50-100m, and 70% for flights of up to 600m, including windy conditions". N.B. I am inferring the definition of "success" from lines 152-153, as a search on "success" in the paper did not locate any other definition, e.g., in the methods.

It would be helpful to clarify how the two outdoor environments (described line 636-637 (methods) as a tennis court and an outdoor test field, without any dimensions provided) correspond to the “small outdoor” and “large outdoor” environments in fig 4c. This is also not adequately clear in lines 165-177 (main text). In fact, it would be preferable to provide (in the figure and main text) some actual dimension rather than ‘large’ and ‘small’, e.g., the average outbound distance flown for each condition. Some of this information can be inferred from extended data figure 7, which shows that that “large indoor” flights were in fact similar in scale to “small outdoor” flights, rather than “large outdoor” flights. There seems no reason not to quantify these more clearly.

Given the relatively small number of real-world flights, all the data for all the flights could be included in extended fig 7. In this figure, the illustrated failures appear to occur despite the path integration returning the drone to within the LHA. This seems worth some explicit discussion in the main text, given the previous tests (fig 3b) that show 100% success for homing within the LHA.

There does seem to be a significant difference in the error for the different online/onboard variants of learning (fig 3c), leading to decreased path efficiency (3b), which does not seem adequately captured in the brief description (lines 155-157). It seems worth commenting that onboard online seems to do as well as onboard offline. In methods, line 675, should this be Onboard offline?

I find the discussion rather underwhelming, both as an analysis of the main strengths and limitations of the approach for robotics, and as providing insight for biology (n.b. line 183, should be ‘purposely’ rather than ‘purposively’). For example, is there anything in the network training approach that limits the number of views (or size of the LHA) that could be successfully encoded? Would learning a more extensive distribution of views, perhaps more sparsely, improve robustness, e.g. to sudden displacement of the robot (kidnapped robot problem)? The supplementary material includes a brief comment (lines 747-8) that dynamic visual elements contribute to the lower success rate in outdoor environments – this seems a point worth discussing in the main text. Lines 196-198 in the main text are very vague about the most crucial next steps to evaluate this approach. From the biological perspective, could the MB learning model be used, in the same way as the more abstract neural networks, to associate visual inputs to vectors rather than just left/right directions? In other words, is the difference observed between the algorithms due to differences in the learning mechanism (e.g. in capacity or generalisation), or differences in what type of outputs are associated to views? And could this suggest ways in which current models of neural circuitry for learning in the insect are incomplete? There is some discussion of this point in the response to reviewers but it is not provided in the paper. Lines 218-219 “the strategy could be extended to include the learning of different places and the neural machinery to take shortcuts between them” could cite some of the earlier discussions of how association of vectors to places might play a role in insect navigation, including the enabling of shortcuts, e.g., Cartwright, B. A., and T. S. Collett. “Landmark maps for honeybees.” *Biological cybernetics* 57.1 (1987): 85-93.

(Remarks on code availability)

Referee #2

(Remarks to the Author)

The authors have made a very substantial effort including significant new work to respond to the reviewers' comments. This review breaks down into a few main components: 1) what the authors have done in response to the reviews and the adequacy / compellingness of those changes / additions / responses 2) the resulting significance and appropriateness of the work as it is presented now, also held up against the updated set of claims, and 3) specific technical suggestions.

Changes / updates:

Onboard / offboard: I agree with referee #1 that much more clarity was needed around where the training occurs - this appears to have been addressed in the revisions. I however also agree with the authors that offboard training is not necessarily a major pragmatic limitation for the specific niche of applications that this work is targeting - this is not general purpose, map-based SLAM / navigation, this is an attempt to create a highly efficient navigation system that works under specific circumstances, and initial offboard training is likely OK for many of these. Where this might be a concern is the general maintainability of the system over time, but that's addressed in a separate point in this review. The authors' response updates 4 potential scenarios: realistically some of these likely start invalidating some of the simplicity benefits of the proposed approach by bringing in more learning and complexity - that's OK.

Comparisons: comparisons have been added / updated for some other lightweight approaches: more computationally intensive systems have mostly been disregarded (except ORB-SLAM 2) as being of a different class of systems - this is OK.

Practical potential: there is plenty of response by the authors to various aspects of the initial reviews. One is around the initial focus on compute: the authors acknowledge that compute energy usage may be affected by other factors like other compute tasks beyond navigation (and of course, general actuation of the micro drone). And the argument that, all other things being equal, less compute / power consumption is always helpful - this is true, but the actual extent to which this is practically significant or not really depends on the deployed system and context. There is a much longer ongoing discussion to be had here around the various implications of trade offs, scope etc... and whether those will actually unfold in practice, but that doesn't really effect my decision on the paper.

On wind and tilt: the authors have tweaked their claims to be more appropriate - and I re-iterate there is nothing wrong with including important engineering considerations. I would suggest the authors also note that the approach to adjusting for tilt in the image rotations is very brittle sensitive to the tilt measurement output by the onboard sensors / the horizon detection: this is a valid approach used in much past work and is probably valid for the subset of tasks this work is envisaged to help with, but can go wrong very quickly if the tilt estimate is even slightly off - this is where more high end feature-based approaches shine, because they can be made much more inherently robust to pose variations in the platform.

In general, the authors have responded to and refined claims around issues like level of biological plausibility / relevance, and clarified intended interpretations around some of the text and made changes in the manuscript accordingly.

I am not fully satisfied with the response of the authors (both scientifically and in terms of practical impact, as well as in the context of how important this component is for the whole research effort) is the response to the request for a better characterisation of the situations under which this approach (the LHA component) will or won't work. This sort of analysis is also expected for much more complex / high end navigation systems - like much of the work in semantic / language-based navigation in computer vision - it's not a unique request to this particular approach. Nevertheless, many of the points have been at least responded to, and much of my remaining hesitation isn't around the inherent value of the work but around a) the claims and b) the likely significant / impact, which I address now.

Assessment of the updated body of work.

The paper is generally improved across the board, in terms of more appropriate claims, extra experiments / evidence to provide more justification for some of the claims, and text and graphical changes / updates to support all of this.

The strongest endorsement of the paper is that the authors have taken a compelling observed navigation phenomenon from nature: the usage of approximate dead reckoning on long paths and the assumption that dead reckoning will get the agent / robot / animal back to the approximate vicinity of the home goal, combined with a computationally lightweight approach to learning how to navigate within that home area, to create and demonstrate both in real world experiments and in simulation the system functioning at a high level, within the specified task and environmental constraints. This is novel, almost certainly of widespread interest to much of the community, and an impressive systems / engineering feat as well (and a major effort, although a truism: so are most submissions to Nature I imagine). The revisions have in general improved the paper and the quality of the research.

I am still slightly uncomfortable about how the contribution is presented, given its heavy dependence on a path integration system - which, although expanded in this revision, isn't really a part of the research contributions of this work - that reliably brings the system back to the vicinity of the home goal, at which point the learnt homing system kicks in. The discomfort isn't about this point in isolation, but in combination with what I would still regard as a somewhat underwhelming and not-sufficiently-rigorous characterisation of the types of home environments where the homing system is expected to learn and successfully operate. The revisions have provided some qualitative observations about some dynamic obstacles, and bit of commentary on ambiguity, opaque obstacles etc... but this is more commentary than anything rigorous. It's also been addressed as "future work" in the response so far. To be clear, I am expecting the system to only operate in a small fraction of potential environmental configurations - just like any other mapping / navigation system including even the more heavyweight ones - but more clarity around this would significantly improve the strength and rigour of the work, and usable relevance to the community who might follow on from this.

A minor pragmatic consideration too is that path integration / dead reckoning systems in practice of course often don't work in the way that noise models suggest - almost always there are catastrophic failures caused by any of an infinite variety of effects - this again doesn't impact the core value of this work but would help better place its potential impact. Recovery for this system would be different (probably some sort of expanded search) to recovery of a system that has mapped the entire (not just local) area - again this is fine, but implies a different practical impact.

Specific details:

SVO-GTSAM LHA percentage of the total flight area is 0.74% in the paper, but the results obtained from running the code shows that the LHA percentage of the total flight area for SVO-GTSAM is 0.0148%. It would be great to clarify if the value presented in the paper is the area achieved using SVO-GTSAM or the area achieved using the proposed approach with properties similar to SVO-GTSAM.

In figure 3c, $n=8$ homing flights/learning steps but the x-axis label mentions different n values for the different scenarios.

Also, the variable n is used for the number of full rotations, the number of learning steps, and the number of attempts the same target waypoint is not reached?

In line 500, the investigated radius factors are (0.5, 1.0, 1.5, 2.0) but the results in Figure 2d, show that it is evaluated from 0.5 to 2.5 in steps of 0.25.

In SI-7 under larger dataset analysis and bootstrapping test subsections, the figure that the text refers to is Extended Data Figure 6, but it should be Figure SI-16.

In SI-8 under the results section, the text refers to Figure SI-20, but it should be Figure SI-19.

In the figure captions where the LHA is described, the area is referred to as a dotted circle, but shown as a dashed circle.

(Remarks on code availability)

Code Quality

The updated code repository has sufficiently addressed all the concerns I raised previously.

All instructions are given for code installation and execution of each project within the code repository. To properly replicate the visual homing simulation experiments, instructions for a docker environment are given, which is easy to follow.

The README files have provided all details about the system input and outputs and hardware requirements, as well as describing the expected results and where the results are stored.

The README file of the provided data folder clearly explains the details of each data and how they can be used and/or visualised.

The revised code repository has removed all hard-coded local paths, and has replaced them with relative file paths, which now work without requiring any modification.

There were only a few minor issues that I came across, which would be worth addressing when releasing the code:

In `code_theoretical_simulation_analysis/insect_navigation.py`, the configuration property `cancel_checked_neurons` is used in one of the conditions. However, none of the `.json` files have that property.

In `robot_network_training/home_learning_laptop/evaluate.py`, the `plot_vectors` function call for the condition where the ground truth is `mocap` has missed the `'heading_mocap'` variable to be passed as one of the required parameters.

For ease of the user, it would be great to provide the option to save the generated plots across all modules with meaningful names, as opposed to the current approach of showing them using the `matplotlib`'s popup window. This is because when too many figures are generated from the same script, it would be hard to keep track of what they represent.

I could test all possible modules except the robot onboard due to specific hardware requirements:

Theoretical simulation analysis (generated supplemental results, figures 1-12)

Provides the visualisations for the learning of the home vectors within the learned homing area.

Path integration noise simulation (generated figures 2a and 2b)

The model for a drone flying outwards and then attempting to return home using path integration with noisy sensors and uses different noise simulations.

Robot network training (generated extended data figure 5a)

The code for training the home learning network using the 3 different collected data that was provided: Cyberzoo, outdoor goodday, outdoor orangepole.

Robot onboard

The code for the onboard robot system requires Raspberry Pi 4 with PX4 flight controller, and a PiCamera. As it needs these specific hardware to run the system, I could not test it.

The README files describe all components of the system and provide information on the prerequisites and installation.

The provided code seems to contain the required files.

Visual simulator docker (generates figures 2c, 2d, 2e and 2f)

The simulation environment. Each script runs and generates random forest environments, saves the generated maps, onboard images and training data (forests) and saves the results and trained models.

Visualised obtained results from the onboard robot testing:

Cyberzoo: using the provided python script (generated figure 3f – homing trajectories)

Followed the instructions to view the PX4 data from the flights for robot learning and full flight trajectories from 7 different collected data. The visualisations include the performance of the controller and the estimated trajectory. The extended data figure 7 shows the flight trajectories of these data.

Referee #3

(Remarks to the Author)

I co-reviewed this manuscript with one of the reviewers who provided the listed reports.

(Remarks on code availability)

Dear editor,

First of all, we thank you and the reviewers for investing time in the review and evaluation of our manuscript. The comments made by the reviewers were very thorough and helpful in improving the quality of the article, which we have modified substantially.

Here we mention a few of the main changes:

- We performed a more thorough comparison with the current state of the art. Additional simulation experiments for comparing the proposed strategy with two other methods that make use of the same learning images have been conducted. Moreover, we discuss map-based navigation approaches in greater depth in the main article.
- We have performed a range of additional robot experiments, including experiments in which the robot learns its view memory onboard. We cover two types of onboard learning: offline, where the robot learns after the learning flight, and online, where the robot learns during the learning flight.
- Where necessary, claims have been better formulated, explained, and substantiated. In particular, we further deepened the investigation of the required percentage of the learned homing area with respect to the total flight area, and adjusted the corresponding claim.
- We have de-emphasized the biological data analysis. It is now only referred to as a “preliminary analysis” in the discussion. Hence, the analysis is no longer featured in the main article, but is now part of the supplementary information. In order to still show the biological inspiration for the proposed navigation strategy, we now include an image of a honeybee and example honeybee flight trajectory in Figure 1, and we name the proposed strategy “Bee-Nav”.

Below, we address all points raised per reviewer with their comments in *blue, italic font* and our responses in black, normal font. We hope that our response and the revised article meet your and the reviewers’ expectations.

On behalf of all authors,

Guido de Croon

Full Professor at Delft University of Technology

Referee #1:

This paper describes an efficient navigation strategy for flying robots inspired by insect visual memory. The algorithm is interesting but not strikingly original: images in a small area around the home location are associated (by training a neural net using the images and the ongoing odometry estimate) to a vector to the home location. This is coupled with odometry (path integration) to allow much larger excursions to be made, provided the accumulated odometry error still returns the robot to the area in which visual learning took place. The algorithm has already been presented in ref. 37, with the main advance here being demonstration of its deployment for successful navigation on real flying robots tested in several indoor and outdoor environments, a substantial effort and important demonstration of efficacy. It is also usefully shown that the error induced by using noisy odometry estimates during learning is still sufficient for navigating home, i.e., the information available purely from the robot's own viewpoint is sufficient for the task.

However, I have two main reservations about the contributions of this paper:

We thank the reviewer for the thorough review, and for acknowledging the importance of the robotic demonstration of the proposed efficient navigation strategy, using only information available from the robot's own viewpoint. Showing that the robot's own odometry suffices for training a home-vector neural network, and identifying the consequences of using drifting odometry to this end is indeed one of the main contributions of this work. Below we will address the reviewer's concerns.

1) Although it is repeatedly stressed that the presented strategy is highly efficient, using a "tiny network", hence suitable for resource-constrained robots such as small flying drones, the key process of learning to associate views to vectors depends on offline training of the network, and the computational demand this requires is unclear. The reader actually has to work quite hard to discover this limitation. The first description of the strategy reads: "...a robot performs a learning flight (Figure 1a), during which it trains a small on-board neural network to map its visual inputs directly to a home vector". One might reasonably infer this is on-board, online learning. The caption to fig 1a specifies that the training of the neural net occurs after all the images are captured (so not online) but not that this is offboard. One has to delve into the methods to read (in the section "Neural networks") that the networks are trained offline on a ground station laptop (not further specified) using data augmentation, and in the section "Learning flight" that the training on this unknown specification laptop takes 2-3 minutes for the simple network and 30 minutes for the more complex network needed for the real world scenarios.

We agree with the reviewer that the description of the learning process in the main article was not sufficiently clear in terms of when and where the learning process took place. The reviewer is right that in the experiments, the robot first gathered images during the learning flight and that training the network took place on a laptop (an Apple MacBook Air with an M1 chip 8 GB). After learning, the network was uploaded to the drone and used for navigation.

Based on this important remark by the reviewer we further reflected on the when and where of the learning. This learning can either be done during the learning flight, i.e., "online", or after the flight, i.e., "offline". Moreover, the learning can either take place on an external computer, i.e., "offboard", or on the robot itself, i.e., "onboard". This leads to four possibilities for the implementation of the learning process, which we evaluate below.

1. Offline, offboard learning:

This is the setup employed in the first submitted version of the article. The main reason for choosing this setup was that it sufficed for showing that (1) the proposed small neural networks can learn various real-world areas based on odometry to a sufficient level of detail to return home, and (2) the proposed

approach enables long-range navigation in real-world environments. Moreover, we believe that for many applications, this setup would be acceptable. After performing a learning flight, the drone will need to land and recharge at the home location. During the recharge, the images of the drone can be transferred to a server, which can train a network and deploy it on the drone. Such a connection to a server would be common in many of the envisaged application scenarios, such as greenhouse and warehouse monitoring. The training process originally took 30 minutes, but this was with naïve, unoptimized code. With a modest amount of recoding effort during the revision period, this was brought down to 10 minutes. This duration is currently quite much shorter than the battery recharging, so that no time is lost. Moreover, it is negligible with respect to the total potential utilization time, if we assume that the robot can use the trained network for many different flights.

2. Offline, onboard learning:

In this setup, the drone stores images during the learning flight, and trains its neural network when it returned to the home location with purely onboard resources. This is an important option, as it implies improved autonomy of the drone, without any reliance on external computation. In remote locations onboard learning may be critical.

3. Online, offboard learning:

This is a setup in which the drone sends its images to an external computer during the learning flight. The computer updates the network and sends the updated parameters back to the drone. Although this approach is technically possible, e.g., with the help of 5G infrastructure, it requires ample communication bandwidth and still relies on external processing.

4. Online, onboard learning:

With this setup, the drone learns during the learning flight. The advantage of online learning is that it potentially allows the drone to already use its view memory when returning to the home location at the end of the learning flight.

Concluding, the offline, offboard learning setup is sufficient to show the main tenets of the proposed approach and is viable for various application scenarios. However, onboard learning is of interest to improve the autonomy of the robot.

Modifications

Based on the reviewer's remark, we have implemented both onboard, offline learning and onboard, online learning and performed additional robot experiments in the flight arena. The results can be seen in the revised article in Figure 3b,c. Because the offline, offboard learning process uses more learning iterations and hence more data augmentation in terms of rotations and lighting variations, it leads to the best results. Still, all three networks succeed in navigating to the home location, where the offboard-trained network finds the home location most quickly. Finally, we now more clearly emphasize in the main article when and where learning takes place for the various experiments.

This issue is particularly relevant because the authors do not provide sufficient discussion of or comparison to 'conventional' map-based navigation, simply dismissing this as something for which lightweight drones "can simply not carry or power the required computational systems", providing only one reference to such systems, a survey paper from 2018. However, there is substantial recent research into 'lightweight' SLAM algorithms and alternative robotic mapping approaches. Moreover, the problem solved here is different; many applications (e.g. surveying) require that a map is produced or that the robot's position is continuously estimated to a precision better than odometry allows. It is reasonable to present a solution for the more constrained problem of returning to a known location,

which indeed does not necessitate constructing a map. But comparison of computational demands should be on a level playing field, i.e., for solving the same task.

We agree with the reviewer that the discussion of SLAM algorithms was too succinct and does not do justice to the wide range of SLAM algorithms in the literature. Moreover, we agree that metric SLAM algorithms (1) produce a map, which may be directly useful to certain real-world applications, and (2) allow for optimal planning between any two locations in the map. With the proposed insect-inspired navigation approach, we purposively sacrifice these two SLAM characteristics in order to substantially reduce the computational and memory requirements for navigation.

Below, we first discuss state-of-the-art SLAM algorithms, including more efficient versions of SLAM. Then, we comment on the comparison of the proposed strategy and SLAM. Finally, we indicate in which way we modified the article.

State-of-the-art SLAM algorithms

In the main article, we now refer to more recent SLAM comparisons, which show that most state-of-the-art methods are still computationally and memory intensive (Sharafutdinov et al. 2023, Herrera-Granda et al. 2023). For example, in (Sharafutdinov et al. 2023) it is shown that the new open-source method OpenVINS uses 6490 MB for an office corridor (the TUM-VI dataset). The most memory-efficient algorithm in that comparison, Basalt, uses 107.8 MB for that corridor, although it does require more computation, saturating 4 CPUs.

There are works that focus on computation and memory efficiency for resource-constrained systems (Tourani et al. 2022). Some of these works share the computation between an edge device and a more powerful server (Xu et al. 2020). Other works do aim for running visual SLAM on autonomous robots, but still need a high-end laptop (Ferrera et al. 2021) or a GPU-enabled embedded computer (Bavle et al. 2020). Alternatively, topological SLAM methods come closer in spirit to insect-inspired navigation methods (Boal et al. 2014, Garcia-Fidalgo et al. 2015), as they sacrifice map accuracy and content for reduced computational and memory requirements. Some of these methods first construct a detailed, metric map with visual SLAM, and then use that metric map to build a topological map for more efficient path planning (Oleynikova et al. 2018, Blochliger et al. 2018). Hence, these methods still require ample processing and memory in the initial phase. Other methods immediately construct a topological graph. For example, in (Khan & Labrosse, 2020) a method is proposed that constructs a topological map online, automatically storing the current image as a new node when it becomes substantially different from the previous node image. Storing an image at each node does still lead to considerable memory requirements as the robot travels further. Moreover, the authors of (Khan & Labrosse, 2020) remark that the need for loop closure and localization lead to increased memory and computational requirements for larger environments – a common challenge for topological SLAM methods (Boal et al. 2014).

In terms of efficiency, the example that comes closest to the efficiency of insect-inspired methods is NanoSLAM by (Niculescu et al. 2023). NanoSLAM is a graph-based SLAM method that is actually very similar to insect-inspired route-following methods, except for the fact that it performs loop closure in order to construct and maintain a global, sparse graph-based map. A tiny quad rotor equipped with a BitCraze autopilot board and a custom AI chip, a GAP9 parallel processor, uses NanoSLAM to explore a cardboard box arena of 4×5 m. The sensor modality used in the experiments was a “laser deck”, i.e., four tiny lasers that measure distances in four orthogonal directions. Using lasers leads to less memory usage but also less expressivity for capturing differences between places in the environment. With this sensor modality, the algorithm takes only 500 kB of memory and requires 250 ms of computation time on the small processors onboard. The computational requirements of the algorithm, and specifically the loop closure module, do scale supra-linearly with covered distance (and hence the number of nodes in the graph).

Importantly, we are not aware of any very lightweight SLAM variant that has been shown to work for long-range autonomous navigation.

Comparison with SLAM:

As the reviewer remarks, the proposed, bee-inspired navigation strategy is different from SLAM, permitting a strict subset of SLAM's navigation capabilities. However, they do both aim at enabling autonomous robot navigation. If one wants to create robots that are able to travel out and come back to the same location, one could think of employing SLAM for this task. Hence, we believe that it will be insightful for readers to have an impression of how much memory a conventional SLAM algorithm would use in the same environment. For this reason, we now also run ORB SLAM 2 in one of the experimental environments (Mur-Artal and Tardós, 2017).

Although ORB SLAM 2 was released already in 2017, it is still one of the most popular methods due to its computational efficiency and robustness (Sharafutdinov et al. 2023, Herrera-Granda et al. 2023). Because it was not designed for low-resolution omnidirectional images, we apply it to a higher-resolution forward-looking camera (which we mounted on our robot solely for this purpose). Our main interest, for the comparison, is in the memory consumption of the map. Hence, the ORB SLAM 2 experiments only involve the mapping phase, not autonomous navigation. The experiments show that the memory consumption in the same large indoor area of a 50-m flight is 1200 MB, out of which 600 MB is used for the map. Comparing the memory usage of the map with that for the neural network (including an input image) gives a memory difference of three orders magnitudes, for the relevant experimental environment. We have included the SLAM experiment in the supplementary information.

Modifications:

Based on the reviewer's comment, we have adjusted the sentence at the start of the introduction to indicate that light-weight robots cannot accommodate *high-precision* SLAM algorithms, adding references to more recent overviews and comparisons. Furthermore, in the introduction we now mention more efficient versions of SLAM in order to do better justice to the wide range of SLAM algorithms. Moreover, in the discussion, we now elaborate on the differences between SLAM and the proposed navigation approach. Finally, we include an experiment with ORB SLAM 2 in the supplementary information (SI-9) in order to determine the map's memory expenditure for one of the experimental environments, referring to it in the discussion.

References:

- Tourani, A., Bavle, H., Sanchez-Lopez, J. L., & Voos, H. (2022). Visual slam: What are the current trends and what to expect?. *Sensors*, 22(23), 9297.
- Sharafutdinov, D., Griguletskii, M., Kopanev, P., Kurenkov, M., Ferrer, G., Burkov, A., ... & Tsetserukou, D. (2023). Comparison of modern open-source visual SLAM approaches. *Journal of Intelligent & Robotic Systems*, 107(3), 43.
- Herrera-Granda, E. P., Torres-Cantero, J. C., Rosales, A., & Peluffo-Ordóñez, D. H. (2023). A comparison of monocular visual SLAM and visual odometry methods applied to 3D reconstruction. *Applied Sciences*, 13(15), 8837.
- Niculescu, V., Polonelli, T., Magno, M., & Benini, L. (2023). NanoSLAM: Enabling fully onboard SLAM for tiny robots. *IEEE Internet of Things Journal*, 11(8), 13584-13607.
- Mur-Artal, R.; Tardós, J.D. Orb-slam2: An open-source slam system for monocular, stereo, and rgb-d cameras. *IEEE Trans. Robot.* 2017, 33, 1255–1262.
- Ferrera, M., Eudes, A., Moras, J., Sanfourche, M., & Le Besnerais, G. (2021). OV²SLAM: A fully online and versatile visual SLAM for real-time applications. *IEEE robotics and automation letters*, 6(2), 1399-1406.

- Xu, J.; Cao, H.; Li, D.; Huang, K.; Qian, C.; Shangguan, L.; Yang, Z. Edge assisted mobile semantic visual slam. In Proceedings of the IEEE INFOCOM 2020-IEEE Conference on Computer Communications, Toronto, Canada, 6–9 July 2020; pp. 1828–1837.
- Bavle, H.; De La Puente, P.; How, J.P.; Campoy, P. VPS-SLAM: Visual planar semantic SLAM for aerial robotic systems. *IEEE Access* 2020, 8, 60704–60718.
- Oleynikova, H., Taylor, Z., Siegwart, R., & Nieto, J. (2018, October). Sparse 3d topological graphs for micro-aerial vehicle planning. In 2018 IEEE/RSJ International Conference on Intelligent Robots and Systems (IROS) (pp. 1-9). IEEE.
- Blochliger, F., Fehr, M., Dymczyk, M., Schneider, T., & Siegwart, R. (2018, May). Topomap: Topological mapping and navigation based on visual slam maps. In 2018 IEEE International Conference on Robotics and Automation (ICRA) (pp. 3818-3825). IEEE.
- Boal, J., Sánchez-Miralles, A., & Arranz, A. (2014). Topological simultaneous localization and mapping: a survey. *Robotica*, 32(5), 803-821.
- Garcia-Fidalgo, E., & Ortiz, A. (2015). Vision-based topological mapping and localization methods: A survey. *Robotics and Autonomous Systems*, 64, 1-20.
- M. A. Khan, F. Labrosse, Visual topological mapping using an appearance-based location selection method, in Towards Autonomous Robotic Systems: 21st Annual Conference, TAROS 2020, Nottingham, UK, September 16, 2020, Proceedings, A. Mohammad, X. Dong, M. Russo, Eds., vol. 12228 of Lecture Notes in Computer Science (Springer, 2020), pp. 90–102.

2) The most relevant ‘insect-inspired’ algorithm to which the current work can be compared is the use of ‘familiarity’-learning of views towards the nest, collected during learning walks or flights, e.g. doi.org/10.1016/j.cub.2015.12.052, doi.org/10.1177/1059712313516132, arXiv:2507.09725). Notably, this approach uses PI to obtain the nest-directed viewpoint, or the relation of the current view to the nest direction, so implicitly learns the association of views to the home vector direction. The latter paper is briefly mentioned here, but somewhat dismissed for using GPS to estimate the location of the robot relative to home, although it would clearly have been straightforward for that work to have substituted onboard odometry for GPS. The learning in this algorithm is extremely simple (could be executed online and onboard) and has been plausibly mapped to insect brain circuits. It would certainly seem that this algorithm would be a much more appropriate comparison than the snapshot algorithm used for comparison here (fig 2d) as it seems self evident that any algorithm storing only one image/location will under-perform compared to an algorithm trained on 100+ image/location pairs. Additionally, many points made in the discussion would apply equally to the familiarity algorithm, e.g., it also assumes path integration is required for learning (to structure the viewpoints stored).

We agree with the reviewer that it is valuable to directly compare the approach proposed in our article with those in the mentioned studies.

Perfect memory

Based on the reviewer’s comment, we now additionally compare visual homing performance with a “perfect memory” method, inspired by Stürzl et al. (2016) and Dewar et al. (2014) in the article. Specifically, the implemented perfect memory method uses the same data as the proposed neural network approach, but instead of training a network it just stores all the learning images as “snapshots” together with the path integration targets. When performing visual homing, the method compares the current image with all stored snapshots, always rotating the current image 360 times (1 degree more for each rotation). For each rotation, the image difference is calculated, and for each snapshot, the minimal image difference is stored. After doing this for all N images, the best matching k snapshot locations are determined. The target vectors from these snapshots are averaged for determining the simulated robot’s action. The results in the article are for $k = 3$. The results for other values of k , such as $k = 1$ were similar.

This perfect memory approach is based on two of the studies mentioned by the reviewer, namely Stürzl et al. (2016) and Dewar et al. (2014). Both these methods are more efficient in different ways. In Dewar et al. only four snapshots are chosen out of all snapshot images made on a grid, based on an observation that more snapshots did not substantially improve homing performance. Here we included all snapshots to ensure maximal performance at the cost of more computation. In Stürzl et al., the navigation was purely based on left and right commands, depending on the home being visible in the left or right visual hemisphere during a real wasp's learning flight. Here, we stored real-valued target vectors, i.e., with a direction and distance, for better performance and more adequate comparison with the proposed navigation method.

The results are included in the main article, Figure 2d. We observe that the performance of the computationally intensive perfect memory method is comparable to the neural network inside of the learned homing area (LHA), but degrades faster outside of the LHA. Outside of the LHA, the matches with the snapshots get worse, up to a point where the current image matches with snapshots located on the other side of the home location. In that case, the robot will move completely into the wrong direction. It may perhaps seem surprising that this approach, which perfectly retains all image information, underperforms with respect to the neural network, which compresses the relevant information in its weights. However, please note that image matching of the perfect memory approach always treats all pixels equally, whereas neural networks can attend to specific relevant (landmark) objects in the image, even if these objects are small compared to the total image. This can allow neural networks to move towards a landmark even if it is further away.

Mushroom-body-inspired strategy

We have also implemented the mushroom-body-inspired (MB) strategy from Gattaux et al. (2025b) in our simulator. We did so by adapting the code available from the route-following work presented in Gattaux et al. (2025a) with the help of the description in (Gattaux et al. 2025b).

We first tested the code in an environment with characteristics similar to the environments in Gattaux et al. (2025b). It is quite open and has geometric objects around the horizon. This resulted in low-resolution edge images looking like the ones of the original experiments. It led to results with homing trajectories that are very similar to those in the robotic experiments of Gattaux et al. (2025b), see the Figure below. Please note that these trajectories, just like the ones in the original experiments, meander substantially, taking quite a long time to converge on the home location. For comparison, we also include the homing trajectories of our "compact" network in the same environment (f in the Figure).

After this verification of the code, we also applied the MB strategy to the visually cluttered forest environments used in our visual homing simulator. Unfortunately, the strategy did not function well in this environment. The main problem seemed to be the ample amount of texture in the environment, leading to pruning too many weights to the mushroom body output neurons (MBONs). We adapted various parameters, such as the number of Kenyon cells, increasing them from 5,000 in (Gattaux et al. 2025b) to even 50,000. Moreover, we varied other settings, such as the number of connections pruned per learning cycle, the frequency of learning, etc. However, we were still not able to get good performance in the more cluttered forest environment used in our simulation study.

[Figure panels redacted]

Figure: [Text redacted] *c,d*: Open environment with geometric objects for verifying the MB code, with an example captured image (left) and the environment overview (right). *e,f*: Trajectories in the environment for the MB approach from (Gattaux et al. 2025b) (left) and the compact network proposed in our study (right), when starting from the same initial positions.

The MB strategy is very interesting, since it is highly computationally and memory efficient. Moreover, the MB neural network structure and learning mechanism more closely approximate real insects' neural structures that are likely involved in this function. However, in the current form, its learning capacity seems lower than that of the backpropagation-trained neural networks proposed in our study. In order

to verify this, we applied both the MB network and the neural networks we used in our experiments to the MNIST digit classification task, a basic benchmark for AI in computer vision. The results of the different networks are shown in the following table, with the average and standard deviation of the results over 10 different training runs. The compact and attention networks trained with backpropagation considerably outperform the MB networks.

Network type	Classification performance
MB network, 5,000 Kenyon cells, 4 active connections from PN to each KC.	47.38% \pm (1.31)
MB network, 20,000 Kenyon cells, 10 active connections from PN to each KC.	52.36% \pm (2.71)
Attention network (5 epochs)	95.97% \pm (0.23)
Compact network (5 epochs)	79.57% \pm (1.95)

We think that the MB approach has great potential. However, the current implementation of the MB neural network and learning mechanism likely do not yet sufficiently approximate the performance of the animal's corresponding brain circuits. As the authors of (Ardin et al. 2016), who introduced the method for visual navigation, put it:

“This is a simplification of the insect MB circuit, which in reality includes significant feedback connectivity, synaptic adaptation at other levels including in the calyx [41,42] and has a substantial compartmentalisation of its inputs and outputs both between and within the lobes [40].”

It may be that further developments of the method will bring the MB method's performance closer to that of the backpropagation-trained neural networks used in the proposed strategy. However, such adaptations and the corresponding amount of work go beyond the timeline and effort possible in the revision period.

We now refer to the comparison with the MB method in the main article and include it in the supplementary information (SI-12).

References:

- Stürzl, W., Zeil, J., Boeddeker, N., & Hemmi, J. M. (2016). How wasps acquire and use views for homing. *Current Biology*, 26(4), 470-482.
- Dewar, A. D., Philippides, A., & Graham, P. (2014). What is the relationship between visual environment and the form of ant learning-walks? An in silico investigation of insect navigation. *Adaptive Behavior*, 22(3), 163-179.
- Gattaux, G. G., Wystrach, A., Serres, J. R., & Ruffier, F. (2025a). Route-centric ant-inspired memories enable panoramic route-following in a car-like robot. *Nature Communications*, 16(1), 8328.
- Gattaux, G. G., Serres, J. R., Ruffier, F., & Wystrach, A. (2025b). Visual Homing in Outdoor Robots Using Mushroom Body Circuits and Learning Walks. arXiv preprint arXiv:2507.09725.
- Ardin, P., Peng, F., Mangan, M., Lagogiannis, K., & Webb, B. (2016). Using an insect mushroom body circuit to encode route memory in complex natural environments. *PLoS computational biology*, 12(2), e1004683.

Further comments:

The abstract highlights that the robot only needs to learn small area (0.5%) of the total flight range. However, this result depends entirely on how noisy the odometry is assumed to be in simulation. In the

methods, the authors provide some detail on how the odometry noise was modelled, and note that this is conservative (i.e. more noisy) with respect to prior work with real drones. However, some brief explanation of the noise model is needed in the main text, as well as reference to how well it reflects real drones and also real insects. This is also needed for fig 2b, e.g. it is impossible to understand what a 'noise level', as shown on the x-axis, means.

The reviewer rightly remarks that the percentage depends on the accuracy of the odometry. In fact, this is the reason that we included a figure in the main article that shows the percentage at different noise levels – illustrating that, at the various noise levels, the LHA remains small compared to the total flight area. However, we acknowledge that a more thorough explanation is required, especially to better substantiate specific percentages mentioned.

In the original submission, we used an Allan variance model for the gyro noise, and we used somewhat higher noise values than mentioned in the literature to better capture the higher noise due to the mechanical vibrations and electronic interference on a flying drone. We also looked at how reasonable the odometry drift estimations were after the inbound flight, i.e. compared with our experience in outdoor flights. However, we did not investigate in detail how well all the drift statistics matched our exact robotic system.

Based on the reviewer's remark, we took the following actions, which will be further explained below:

- (1) We modified the noise model for the heading angle drift, so that it can be better fitted to robotic data and more easily explained in the main article.
- (2) We performed path integration experiments with our drone, measuring specifically the position and heading angle drift.
- (3) We then searched for noise parameters that fit our robot. Moreover, we also fit parameters to match with data of other path integration methods from the literature, which can form a reference for different robotic systems.

All these steps have led to a more accurate determination of the percentages that can be expected from real robotic systems.

(1) Noise model

Based on the reviewer's remark that we should explain the model in the real text, we decided to adapt the noise model slightly. In the initial submission, we used a detailed Allan variance model for simulating gyro noise. However, this model is more difficult to explain and also more difficult to fit to the data from our robotic experiments and from data reported in the literature (see action 3 below). Hence, we slightly simplified the heading drift model, aligning it with noise models used in relevant work (Wystrach et al. 2015, van Dijk et al. 2024). Specifically, we now add heading drift noise according to the normal distribution $\mathcal{N}(0, t\sigma_\psi^2)$, where t is the time in seconds. So, if we have a time step of 1 second, then each time step we draw a random variable from $\mathcal{N}(0, \sigma_\psi^2)$ and add it to the heading estimation. This leads to an accumulation of heading drift over time. The distance estimation noise is still modelled as $\mathcal{N}(0, d\sigma^2)$, with d the distance in meters. This new noise model is easy to explain, while capturing the two main relevant types of path integration noise for a system with heading drift.

(2) Path integration experiments

We performed various path integration experiments with our drone, measuring the position and heading drift over time (see the figure below for an example). We did this in different conditions: flying above artificial grass with little texture vs. flying above a highly textured underground, and flying with vs. without wind (both above the low-texture artificial grass). The experiments provide data for position drift after different flight distances and heading drift at given flight times.

Figure: Robotic experimental result for measuring path integration drift in our motion tracking arena. *Top left:* The test environment, with artificial grass and therefore limited texture. *Top right:* Motion capture ground truth position vs. odometry-based position estimate. *Bottom left:* Position error over flown distance. *Bottom right:* Heading drift over time.

(3) Noise parameters

Subsequently, we searched for noise parameters that approximated the robot’s drift statistics. Please note that the path integration used by our robot is very standard and does not feature a heading measurement (no “compass”). Path integration without a compass of any sorts is referred to as “idiothetic” path integration in the biology literature (Heinze et al. 2018) and is known to be relatively inaccurate. In the case of our robot, we use PX4’s visual odometry estimate from a downward looking camera and laser. For the heading estimation, the Extended Kalman Filter depends on noisy MEMS gyro measurements, which are essentially integrated over time. We finally selected $\sigma_\psi = 0.63^\circ$, and $\sigma_d = 0.10\text{m}$ for our robotic system, which gave a mean heading drift after 100s of 5.03° and a mean position drift after 100m of 0.91m. When simulating 1000 outbound and inbound travels, this setting results in a Learned Home Area (LHA) that is 3.84% of the total flight area. This means that even with the quite crude path integration of our robot, the percentage of the LHA with respect to the total flight area is very low. We then repeated the simulation-based “noise level” experiment, scaling both σ_ψ and σ_d with $\{0.10, 0.5, 0.75, 1.0, 1.25, 1.5\}$. With a requirement of the LHA containing 99% of the return positions, this gives LHAs that span from 0.1% to 10.82% of the total flight area. The expected percentage for our robot is indicated by a circle marker (Figure 2b).

In order to give a more comprehensive idea of what could be achieved with robotic systems, we also fitted our noise model to what is known of two methods from the literature. First, more advanced versions of Visual Inertial Odometry (VIO), using a frontal camera and IMU, feature a better path integration accuracy. To illustrate this, we also determined parameters for the noise model that led to a

similar drift as that of the visual-inertial odometry method “SVO-GTSAM”, as reported in (Delmerico and Scaramuzza, 2018). In that article, SVO-GTSAM is reported to have a mean drift of $\sim 0.3\text{m}$ and heading drift of $\sim 0.8^\circ$ after 35 meters of flight. Our noise model, with $\sigma_\psi = 0.25^\circ$, and $\sigma_d = 0.015\text{m}$ leads to similar statistics: a mean position drift after 35m of 0.32m and mean heading drift of 0.79° . With this VIO, the LHA percentage of the total flight area would be 0.74% according to the simulation experiments. It has to be noted that SVO-GTSAM requires substantial processing. Hence, it would require considerable onboard computational and memory resources – which goes against the efficiency of the proposed navigation method.

Second, we also fit the noise model to a robotic odometry method from the literature that does use a compass, in the form of a magnetometer (Stankiewicz and Webb, 2020). They use an efficient, bio-inspired VIO method and report a position drift of 1.5m after 100m of robotic flight. Since their heading does not drift, we slightly adapted the noise model. In particular, in the case of their setup, the heading noise no longer accumulates, since the magnetometer can provide an absolute heading measurement. Instead, the Gaussian noise is added at each time step to the true heading angle. The so-adapted noise model, with $\sigma_\psi = 5.5^\circ$, and $\sigma_d = 0.15\text{m}$ also leads to 1.5m drift after 100m of travel. The corresponding percentage of the LHA is 0.24%. This example demonstrates that sensing of an absolute heading angle (called “allothetic” path integration in the biological literature) substantially improves the accuracy of path integration and hence the efficiency of the proposed navigation strategy.

Insect path integration accuracy:

Determining path integration accuracy for real insects is challenging (Heinze et al. 2018). In the biological literature, in (Wystrach et al. 2015) a path integration noise model is fit to ant data from (Merkle and Wehner, 2010). In that latter study, ants were captured 10 meters away from the nest and displaced in darkness to a different site. After displacement, the ants would first use their path integration estimate to go back to the nest location. However, because the other site did not have the nest, they would commence a search upon not detecting the nest at the expected location. We essentially use the same model as (Wystrach et al. 2015) for the case of allothetic path integration. Translating the settings from that article to the standard deviation used in our model, we arrive at $\sigma_\psi = 57^\circ$, and $\sigma_d = 0.38\text{m}$ for the simulations. Note that although the angular noise seems large, it does not accumulate over time. This setting leads to an LHA percentage of 7.6%.

We have searched for a similar model for honeybees, but were not able to find any. However, in (Wang et al. 2025), honeybee experiments are performed that form an indication of the path integration accuracy. In particular, the researchers tracked honeybees that were recruited to go to different foraging locations for the first time. They were then able to measure where the honeybees would start searching. Notably, for a foraging site 2300m away from their “hive 2”, the mode of the search location distribution was only $\sim 200\text{m}$ away. Although there are many factors that can contribute to the inaccuracy here (including “noise” in the communication of the honeybee dance), this offset can be regarded as an indicator for the path integration drift. As an approximation of honeybee odometry accuracy, we have fitted the allothetic noise model to the data for the furthest foraging site. This site was selected because larger distances lead to more reliable parameter estimates than smaller distances. The best fit was achieved with settings of $\sigma_\psi = 34.5^\circ$, and $\sigma_d = 0.38\text{m}$. This leads to 201.1m drift after 2300m of straight flight, and an LHA percentage of 3.4%.

We now also mention these percentages in the text, adding that they are based on a coarse approximation of real insect path integration accuracy.

Final remark:

We have incorporated the new results in the article, now also briefly explaining the noise model in the main text. The details explained above have been incorporated in the supplementary information (SI-

11). Moreover, we have rephrased the claim in the abstract to clarify that the exact percentage depends on the odometry accuracy: “Simulation experiments showed that for realistic path integration accuracies the neural network only requires training on ~0.25-10% of the total flight area.”. Thus, we now refer to a wider range of percentages and clearly make the link to the accuracy of path integration. Finally, we also mention the percentages for the coarse approximation of ant and honeybee path integration accuracy.

References:

- Wystrach, A., Mangan, M., & Webb, B. (2015). Optimal cue integration in ants. *Proceedings of the Royal Society B: Biological Sciences*, 282(1816), 20151484.
- Thrun, S. and Burgard, W. and Fox, D. (2005). *Probabilistic Robotics*. The MIT Press, Cambridge, MA, Intelligent Robotics and Autonomous Agents series.
- van Dijk, T., De Wagter, C., & de Croon, G. C. (2024). Visual route following for tiny autonomous robots. *Science Robotics*, 9(92), eadk0310.
- A Benchmark Comparison of Monocular Visual-Inertial Odometry Algorithms for Flying Robots, by Jeffrey Delmerico and Davide Scaramuzza, 2018 IEEE International Conference on Robotics and Automation (ICRA), May 21-25, 2018, Brisbane, Australia.
- Heinze, S., Narendra, A., & Cheung, A. (2018). Principles of insect path integration. *Current Biology*, 28(17), R1043-R1058.
- Stankiewicz, J., & Webb, B. (2020, July). Using the neural circuit of the insect central complex for path integration on a micro aerial vehicle. In *Conference on Biomimetic and Biohybrid Systems* (pp. 325-337). Cham: Springer International Publishing.
- Narendra, A. (2007). Homing strategies of the Australian desert ant *Melophorus bagoti*. I. Proportional path-integration takes the ant half-way home. *J. Exp. Biol.* 210, 1798–1803.
- Cheung, A., Hiby, L., and Narendra, A. (2012). Ant navigation: Fractional use of the home vector. *PLoS One* 7, e50451.
- Wang, Z., Chen, X., Okada, R., Walter, S., & Menzel, R. (2025). Encoding and decoding of the information in the honeybee waggle dance. *Behavioral Ecology and Sociobiology*, 79(5), 53.
- Using the Neural Circuit of the Insect Central Complex for Path Integration on a Micro Aerial Vehicle, by Jan Stankiewicz and Barbara Webb, *Living Machines 2020*, LNAI 12413, pp. 325–337, 2020.
- Merkle, T., & Wehner, R. (2010). Desert ants use foraging distance to adapt the nest search to the uncertainty of the path integrator. *Behavioral Ecology*, 21(2), 349-355.

The review of previous insect-inspired strategies seems a little unbalanced. E.g. research on route-following has focused on view memory alone because the behavior of interest was the ability of insects to travel home along familiar routes after displacement, i.e., when path integration is zero or conflicting, and in cluttered environments. It seems unfair to imply earlier researchers have ‘missed’ the obvious advantages of combining view memory and path integration, and that this paper has made a breakthrough by combining them.

We are sorry that we created this impression, as it was not our intention to do so. We have rephrased that part of the introduction: “*Route-following is a suitable strategy for navigating in highly cluttered environments, but in relatively open areas it can make the return journey unnecessarily long.*”

Moreover, it was not our intention to claim that the idea of combining path integration and view memory itself is the breakthrough presented by our work. Although we tried to convey this already in the initial submission, we have now made this clearer, by writing: “*The idea of combining path integration and view memory in this manner formed the basis for the seminal work on “Sahabot 2”, a mobile ground robot that navigated in the desert*²⁵.” However, please note that in the Sahabot 2 study, separate robot

experiments were performed for path integration and visual homing. To the best of our knowledge, also subsequent robotics studies did not combine path integration and view memory to achieve real-world demonstration of efficient, long-range robot navigation.

In general, the experimental descriptions could benefit from a little more detail in the main text, as it is hard to get a clear idea of the implementation. E.g. that the visual input is omnidirectional, unwrapped, visually rotated; that the odometry is based on optic flow and IMU; how the recovered vector is used to control movement, etc. Some but not all this information is in figure 1.

We have now added more details on the robot setup to the main text and figure captions. Furthermore, we have introduced Extended Data Figures 3–5, which visually describe the visual processing pipeline, the label generation process, and how the predicted homing vector is utilized for robot control.

How important is the learning flight trajectory? This is described as bee-like in the text but does not obviously resemble the learning flights shown in extended data fig 4. In the methods it is described as wasp-like.

The learning flight trajectory is important since it determines the data for learning the homing vector and hence has a substantial influence on the subsequent homing performance. The reviewer is right that we did not mimic the actual honeybee learning trajectories. Honeybees typically seem to fly a “loop” in a specific direction away from the hive, varying this direction somewhat when making subsequent learning flights. They often do not cover all directions. Given the proposed technological solution, we opted for a pattern that would ensure that the neural network obtains homing vectors in all directions around the home location, providing better coverage of the entire circular learned homing area.

Initially, we opted for an Archimedean spiral pattern around the home location. However, with this learning flight pattern, the same side of the drone would always face the home. If the drone’s body has a structure in sight in that direction (e.g., an antenna as in our experiments), the neural network may falsely associate that fixed body direction with the direction home. That is why, later, we changed the pattern to a “wasp-like” pattern (inspired by the nest inspection flights of wasps, *Cerceris australis*). In this pattern, the drone reverses direction at the end of each arc of the trajectory. In the supplementary information (SI-6), we provide experimental results comparing these two flight patterns. They show that the network trained with the wasp-like pattern achieved a mean angular error of 6.68° , substantially outperforming the 17.23° error from the spiral-trained network.

We now more clearly explain in the method section why we do not mimic honeybee trajectories, and why we converged on the wasp-like pattern.

Finally, we thank the reviewer for the thoughtful review of our article. We hope that the additional experiments and textual changes have addressed their main concerns.

Referee #2 and #3:

Referee 3 performed the review together with referee 2 (referee 3: *“I co-reviewed this manuscript with one of the reviewers who provided the listed reports.”*). Hence, here, we address the joint comments of both reviewers.

This is interesting research in the area of lightweight robot navigation systems, building on a substantive body of work in the general area by the authors, who are one of the leading groups in this broad area internationally, and in this specific domain (lightweight drone-specific visual navigation) arguably the leading group. The capability demonstrations are impressive and as far as I know firsts in the specific area of lightweight visual out and back robot navigation in terms of scale and size.

We thank the reviewers for their kind words and their recognition of the navigational capabilities achieved with the proposed lightweight navigation scheme.

We break down our review into several components:

Novelty and delta over previous work

The authors have a range of prior work such as their 2024 Science Robotics paper, “Visual route following for tiny autonomous robots”. The authors are primarily “competing” against themselves here, in terms of their string of relatively recent publications in the general area. The work described in their 2024 paper already broadly described an out and back lightweight visual navigation system for drones. In terms of additions here: there is a substantially different approach, especially around the local homing area and associated training, the comparison to bee flights, and the general sophistication of the approach. The scale and challenge of the experimental demonstrations is much larger and more impressive.

We thank you for this assessment of the novelty, with which we agree. Our previous work was able to fly a distance of 100 meters in an indoor environment. However, as it performed route-following, it was not able to follow a shorter, straight flight home. Moreover, the visual processing in that paper was much less sophisticated and robust (performing a Fourier transform of the omnidirectional image, assuming objects in view to be equidistant from the camera).

The newly proposed approach is more robust and path efficient, enabling for the first time to tackle challenging, large-scale experimental demonstrations.

Practical Relevance and Utility

The idea of widely deployed small robots as detailed in the article has long been a dream of many roboticists: it has not yet happened in any enduring capability, and it’s telling that new enduring widespread deployments of any largely autonomous robots (large or small) has been extremely limited over the past few decades, with a few notable exceptions like logistics robots in warehouses, and possibly autonomous robotaxis as they scale up. That is to say, it’s a fiendishly difficult problem, lots of advances are needed, this research makes a valid contribution to some of the challenges but there are so many general challenges that at this stage this is technology that is still likely a long way off widespread deployment beyond proof-of-concepts and pilot studies. This comment is more to set the context in which this work occurs. There is a positive here - being by no means a solved problem - the value of research like this is more significant than if this was a mature field.

From a pragmatic perspective - how much could this work (eventually) revolutionize aspects of robotics navigation in commercial and deployed applications - the biological inspiration is largely irrelevant except indirectly as to desirable properties like lightweight compute etc...

We agree with the reviewers' analysis here. It is indeed also our dream to deploy many small robots to perform real-world tasks. Let us discuss one application on which we perform substantial work in order to illustrate the importance of the lightweight compute.

In our group, we perform a substantial amount of research to enable the deployment of swarms of small flying drones in greenhouses to monitor the crop. This will allow for early disease or pest detection and, eventually, support precision agriculture applications in general, thereby increasing crop yield and reducing waste and pesticides.

From a pragmatic perspective, small robots / drones bring unique properties to the table that will unlock novel applications. For example, small, light-weight drones will be very suitable for use in greenhouses, since:

1. They are inherently safe for human workers. Humans will not have to adapt their way of working or their clothing to the presence of these drones (no helmets / safety glasses / etc.). Please note that this property is vital; even if the drones completely fail, they will not pose any harm – we believe this could lead to earlier adoption than that of larger, heavier, and potentially physically dangerous robots (Svatý et al., 2025).
2. They can fly in narrow spaces, initially over low crops, but later in between ranges of e.g. tomatoes, and eventually in between plants – if we can reach a true insect-scale.
3. If lower-end processors are used, they will likely be much cheaper than larger robots, so that they can be produced in large numbers – allowing for swarms of these robots to quickly cover large areas.

Of course, the main disadvantage of light-weight drones (and very small robots in general) is that they are extremely resource-restricted, with little payload capability for sensors and processing. Consequently, conventional approaches to robotic AI are typically not within reach.

Most importantly, autonomous navigation is key to almost all real-world applications one can think of. However, state-of-the-art navigation algorithms like metric SLAM do not fit within the available resource budget. In the greenhouse, a drone that cannot make its way back to a charging station would have extremely limited use. Even though this can be solved with external infrastructure like ultrawideband beacons, in our experience greenhouse companies find the associated investments in installing and maintaining that infrastructure prohibitive.

Thus, we expect this work to have a high impact, as it provides, for the first time, a feasible option for providing the rudimentary but necessary home-navigation skills to resource-restricted robots. Of course, we agree with the reviewers that we are currently still far from a product stage. In terms of technological readiness level, our research reaches Technological Readiness Level (TRL) 4 or 5: demonstrations of the technology in lab / real-world environments (Mihaly, 2017). Much is to be researched and developed before we reach the product-stage of TRL 9. However, it at least shows us a path forward to achieve the dream of widespread, real-world deployment of autonomous small robots.

We modified the discussion to better reflect on these aspects in the article.

References:

Svatý, Z., Vrtal, P., Mičunek, T., Kohout, T., Nouzovský, L., Frydrýn, M., ... & Kocián, K. (2025). Impact analysis assessment of UAS collision with a human body. *PloS one*, 20(3), e0320073.

Mihaly, Heder (September 2017). "From NASA to EU: the evolution of the TRL scale in Public Sector Innovation", The Innovation Journal. 22: 1–23.

In this regard, there are some clarifications that are needed:

The rationale around the benefits of this lightweight system need more detail and more specifics. How much does only having 40 kB of storage versus 100s to 1000s of MB (a space inefficient SD card has been able to store this amount of data for many years) matter practically. Obviously less is always better, all other things being equal, but how significant is this specific memory claim in practical terms of how a complete system (which will have other requirements and capabilities) perform - especially if some of the capabilities require more compute / storage anyway. Or alternatively, what is the class of real-world problems where that wouldn't be the case?

I am surprised that the claim focused in significant part around storage - I would assume the primary practical benefit here is around the amount of data / training / compute / energy consumption that is required to train the larger / more memory intensive systems - rather than the memory itself, per se - if that impression is wrong, some detailed cost / bulk / energy consumption statistics for the lower memory footprint itself would be helpful in making the case.

We thank the reviewers for bringing this important matter into focus. Here, we first discuss the importance of saving memory and then computation and power.

Memory:

In the first version of the article, we indeed focused primarily on the memory footprint of the neural network necessary for successful home-navigation. On the one hand, despite the availability of small storage devices such as SD cards, the memory required for navigation is still of importance. This is due to the different types of memories in processors, where usage of the stored data requires interfacing with RAM. Having a small memory footprint could allow for using the network even onboard of small, light-weight, low-power, and cheap microcontrollers. For instance, STM recently released the STM32N6 (see link under References below). This microcontroller contains a Neural Processing Unit (NPU). It has 4.2MB of RAM, 128 Kb of ECC RAM and 64 kB of instruction TCM RAM.

On the other hand, as the reviewers mention, one may need more computational power and memory anyway for the application at hand. In that case as well, minimizing the memory necessary for navigation increases the general capabilities of the robot, as it enables performing more other tasks. Moreover, the used small neural networks do not only imply little memory usage, but – as the reviewers emphasize – also enhanced computational efficiency.

Computation and power:

Compute is indeed also really important, as it largely determines the size, weight, power and economic costs of the onboard computer.

The most important compute is, in our eyes, that which is necessary for inference, so after the learning process. For small networks, this compute is very limited. The larger, inception network takes 300 ms of compute on a Raspberry Pi, which has a quite limited 1.5 GHz processor. It weighs 46 grams and consumes 2.7 - 6.4 Watt, while costing in the order of 35\$. In fact, we even expect the STM32N6 chip mentioned above to be able to run the type of small deep neural networks we use in our experiments, while being much smaller, more lightweight, and power efficient. The chip's specifications mention 3 Trillion Operations Per Second per Watt, i.e. 3 TOPS/W. The chip will typically consume in the order of a few hundred of milliWatts, and typically stay below 1.5W when running the network, at a cost of ~17\$.

In order to put these numbers in context: Works that aim for running visual metric SLAM on autonomous robots still need a high-end laptop (Ferrera et al. 2021) or a GPU-enabled embedded computer (Bavle et al. 2020). An example of the latter is the NVidia Orin NX, which together with a fan and carrier board weighs ~100 grams, consumes between 10 and 25 Watt of power, and costs in the order of ~700\$. This implies a larger, heavier and – very important for production and real-world application – a much more expensive robot. Moreover, even high-end processors may be saturated by map-based navigation alone (Sharafutdinov et al. 2023), leaving little place for other tasks the autonomous robot will have to perform.

There is also compute required for training. In the setup presented in the first submitted version of the article, training was done offline and offboard on an Apple MacBook Air M1 chip 8GB. Training happens only once, and takes only ~10 minutes. This is currently less time than that necessary for recharging the drone’s battery. Moreover, it will be negligible compared to the potential total flight time when using the trained network for many flights. For the new article version, we now also investigated onboard training, running on the Raspberry Pi, both after the learning flight (offline) and during the learning flight (online). Both forms of onboard learning also lead to successful homing, albeit with less accuracy and hence slightly longer visual homing (see the new Figure 3b,c).

Based on the reviewers’ remark, we have revisited the text (including the discussion), better emphasizing also the computational efficiency of the proposed strategy.

References:

- <https://www.st.com/en/microcontrollers-microprocessors/stm32n6-series.html> (STM accessed December 2025)
- Ferrera, M., Eudes, A., Moras, J., Sanfourche, M., & Le Besnerais, G. (2021). OV²SLAM: A fully online and versatile visual SLAM for real-time applications. *IEEE robotics and automation letters*, 6(2), 1399-1406.
- Bavle, H.; De La Puente, P.; How, J.P.; Campoy, P. VPS-SLAM: Visual planar semantic SLAM for aerial robotic systems. *IEEE Access* 2020, 8, 60704–60718.
- Sharafutdinov, D., Griguletskii, M., Kopanev, P., Kurenkov, M., Ferrer, G., Burkov, A., ... & Tsetserukou, D. (2023). Comparison of modern open-source visual SLAM approaches. *Journal of Intelligent & Robotic Systems*, 107(3), 43.

There is a significant treatment of wind, and the tilt effects and compensating for these. As far as I can determine, this is a pure engineering-based approach, to deal with a practical real world issue. There is a range of related work in area like high performance drones (e.g. work out of Scaramuzza group, the work that Opteran does for its main sensor) so this is largely a needed capability enhancement but not particularly interesting from a research perspective. It’s needed practically, and a practical contribution, but I would evaluate those components as not particular novel or state-of-the-art compared to other work in the field, including in commercially deployed drones.

We agree that the method we developed for dealing with the tilt due to the wind is a practical, engineering-based method.

Although we did not intend to present this method as part of the scientific contribution of this work, we did not mention related studies when describing our approach to tilt correction in the method section. We now mention that correcting for tilt is a known problem in the visual homing literature (Berganski et al. 2023) and that the vision-based algorithm we developed is reminiscent of the type of solution proposed in (Natraj et al. 2013). Concerning this latter point, they use Markov random fields together with an optimization algorithm to fit an ellipse to the omnidirectional image. Although different from the method we propose in terms of implementation, the method is similar in spirit.

We hope that the textual modifications, presenting it mainly as an engineering effort in the methodology the reference additions address the reviewer's concerns in a satisfactory manner.

References:

Berganski, C., Hoffmann, A., & Möller, R. (2023). Tilt correction of panoramic images for a holistic visual homing method with planar-motion assumption. *Robotics*, 12(1), 20.

Natraj, A., Ly, D. S., Eynard, D., Demonceaux, C., & Vasseur, P. (2013). Omnidirectional vision for UAV: Applications to attitude, motion and altitude estimation for day and night conditions. *Journal of Intelligent & Robotic Systems*, 69(1), 459-473.

Similar comment about "The uniformly textured floor posed challenges to visual odometry" - this is not a problem for SOTA VO methods, is this a limitation of this specific method's shortcomings? (compute, sensor, ???). Again, VO is not a claimed contribution here, this is more about understanding what is being used and what its limitations are compared to other more sophisticated systems (that might use a lot more compute).

Here, we use a downward looking optical flow sensor PWM3901. This is a standard VO solution for drones, which is low weight and low power (0.6 gram and 33 milliwatts). The limitations of this approach derive from the very limited field of view and the basic optical flow algorithm running in hardware on the board: The optical flow here consists of a single two-dimensional optical flow vector for the entire field of view. When there is enough visual texture below the drone, this VO solution gives good results. However, when the floor below has little texture, the velocity estimate will be erroneous leading to more odometry drift. With the proposed navigation strategy, this drift is cancelled by the homing network.

In contrast, state-of-the-art VO methods typically use a forward looking camera with a wider field of view. For instance, in the study that presented semi-direct monocular visual odometry (SVO), a Matrix Vision BlueFox, global shutter, 752×480 pixel resolution was used (Forster et al., 2014). These methods typically track a large number of features – or even all pixels – in the field of view. They use the resulting optical flow to determine both the structure of the environment (distances of all tracked points) and the ego-motion of the camera. This much more advanced visual processing requires more compute, e.g., in the case of SVO, an Odroid-U2, ARM Cortex A-9, with 4 cores at 1.6 GHz. Still, they do also need visual texture, as in the absence of texture there is no information to deduce ego-motion. With forward-looking, broad field-of-view cameras, the probability of having low texture is much smaller, especially in human-made indoor environments where floors can be quite uniform. However, when the camera starts moving much faster (as in drone racing), the motion blur can become so considerable that even such VO methods fail. Visual-Inertial Odometry (VIO) methods use inertial sensors like accelerometers to partially compensate for temporary absence of suitable visual texture, but will still lead to drift on the longer run.

In any case, more advanced VO methods will enable a drone to either fly further or learn a smaller homing area. Still, even more expensive VO methods will always drift, and more so for longer journeys. So, a method for cancelling the drift remains necessary, as that proposed in the article.

We now mention the visual odometry method we used in the main article text leading up to the text quoted by the reviewers. Moreover, in the noise analysis in simulation, we show the effects of having a more accurate form of visual odometry on the efficiency of the proposed navigation method (Fig. 2b).

References:

- Forster, C., Pizzoli, M., & Scaramuzza, D. (2014, May). SVO: Fast semi-direct monocular visual odometry. In 2014 IEEE international conference on robotics and automation (ICRA) (pp. 15-22). IEEE.

Indoors versus outdoors - there is some commentary here: how much of this commentary isn't about absolute differences in indoor / outdoor environments, and how much is just conditions of wind, lighting e.g. there are likely indoor environments (flickering lighting, strong internal ventilation, highly aliased visuals) that this system would struggle in more than certain outdoor environments (no wind, unique and abundant landmarks). I would suggest phrasing this less as indoor versus outdoor, and more about the properties of the environment that make it easier or more challenging.

We agree with the reviewers that the differences in the environments do not lie in the fact that an environment is indoors or outdoors per se, but in the characteristics of the environments. The difficulties we encountered in the large outdoor environment were due to factors such as the wind, leading to larger pitch and roll angles, changing lighting, and the openness of the environment, i.e., absence of close-by landmarks. However, not all outdoor environments present the same challenges. For example, the outdoor tennis court environment had trees around, which formed relatively close-by landmarks and shielded the environment from the wind. Finally, as the reviewers remark, also indoor environments can have their challenges, such as little texture on the floor or highly aliased visuals.

We now discuss this matter with more nuance in the text.

Approach

The approach essentially boils down to: make a path integration system that doesn't drift too much, go out on an arbitrary route whilst path integrating, then beeline straight back to the start location, hope that the drift isn't too large to end up in the LHA (Learned Homing Area), then visually home in on the starting location. That is still an impressive achievement, especially as demonstrated in this domain / with actual drones indoors and out, but the core task is fairly straightforward to describe.

We agree with the reviewers that the approach is easy to explain, and we strongly believe that this is one of its strengths.

Currently, many people believe that for autonomous navigation robots need to construct *maps* of their environment. The reason for this successful adoption of maps is partially because it is easy for people to understand how this would work: We understand that if we have a map, and we know where we are, then we can easily plan to go to some other place that is on the map.

When proposing an alternative to maps, it is important for general acceptance that it can also be easily understood. The reviewers perfectly express the concept of learning a homing area that is large enough to capture the maximal expected drift on a homeward travel. Please note that both the theoretical analysis and the results in the visual simulator show that the visual homing network can even generalize beyond the learned homing area, further reducing the probability of not making it home.

Bringing this idea to life with odometry-based, self-supervised learning of a homing neural network and implementing and executing it on a real flying robot was indeed far from trivial. However, the core concept is definitely easy to grasp, where we prefer the term “elegant” but are prepared to accept terms like “straightforward” or “simple” as a badge of honour.

There are multiple aspects to this:

Path integration that doesn't drift too much: this has been done extensively in drones and autonomous vehicles, and here it's either abstractly simulated in simulation or implemented on the drone.

There is indeed ample work on path integration, such as the large literature on visual-inertial odometry. Any path integration will work with the proposed approach, but its accuracy will determine how large the learned homing area has to be and / or how far the robot can travel away from its home location. In our robotics experiments, we have employed a quite standard method for path integration; The robot's extended Kalman filter fuses and integrates the velocity as estimated with the optical flow from a downward looking camera and the height measurement of a downward pointing laser. The heading is updated with the help of gyro measurements, and is hence subject to substantial drift. We employed this standard method for drone VIO to show that the Bee-Nav also enables long-range navigation when employing cheap, standard hardware. More elaborate visual odometry methods will allow for traveling further and / or learning a smaller homing area.

We will revisit this point in detail when we will address the reviewers' concern on the claim of the size of the learned homing area with respect to the total flight area. There we will also explain how we modified the article in order to better explain the path integration noise model, its parameters, and the impact of path integration accuracy on the relative size of the learned homing area.

The LHA approach is neat, but it will not work in a large variety of environments. That does not negate the potential value and impact of this work, but the wide range of assumptions that the LHA approach makes about the environment in order to be functional need to be detailed. What would happen in an area with inter-meshed with opaque barriers / walls? I assume it will not work in a completely visually ambiguous environment? Highly dynamic environments are also likely an issue. What are the exact properties of the LHA area in order for the method to likely be performant? Sometimes this could be more convincingly answered, albeit indirectly, by, "we ran 1000s of real-world trials" (which is very difficult to do) but that is not the case here given a small number of (real world) trials (which are hard).

It is currently hard to fully answer the question what the exact limitations of the proposed visual homing method are. Still, since visual homing is learned and performed with a deep neural network, we expect that the LHA approach can work in a wide range of environments. Deep neural networks form the state of the art in computer vision, since they are able to capture complex visual concepts. This means that if there *is* visual information on the direction and distance to the home location, we expect that a deep neural network can extract it. The question is how large or complex the network will have to be for this. The tiny "compact" neural network used in Bee-Nav has proven to be sufficient for all environments covered in our study, except for the largest outdoor environment. For that environment a slightly larger "attention" neural network did provide satisfactory results. Future work should investigate the relationship between the complexity of the environment and the capacity of the required neural networks.

This said, we now comment on the specific potential problems mentioned by the reviewers.

Opaque walls obstructing the view on the home location are by themselves not a problem, since the learning of the home vector is based on path integration, not on the recognition of a home object in view. In our simulations, we see that the network is perfectly able to map images to a home direction when this direction is obstructed by a tree for instance. Based on the reviewers' remark we now have also performed additional experiments with opaque obstacles inside the learning area (SI-8).

However, **ambiguous environments** are problematic. In a sense, this was the case in the large outdoor environment, since there were no close-by landmarks. Because far-away landmarks only change with heading and not with position, all positions in the learning area resulted in highly similar images. This

is why in the large outdoor environment we added some features to the ground. Of course, a similar thing could happen in an indoor environment for a different reason; if around the home location there are four completely identical corridors, then there is no visual information available for retrieving the right direction. Also animals are subject to such ambiguities. For example, in greenhouses, bumblebee boxes are often deployed for pollination purposes. However, if these are located at the end of a row of plants that is highly similar to other rows, bumblebees take much longer to return to their box. Hence, in practice the greenhouse keepers add some visually conspicuous object close to the box in order to facilitate navigation for the bumblebees. In nature, it could be that honeybees or bumblebees take this matter into account when choosing a novel home location.

With coarse, wide field-of-view vision, the presence of some small to medium-sized **dynamic objects** is not a problem. For example, we noticed that during the learning flights the experimenter was often visible in the omnidirectional camera. During the homing part of the flight, the experimenter was often at a different place. Because the experimenter only occupied a small part of the image this has not led to any substantial performance loss. However, if the dynamic objects dominated the field of view, then this would impact homing performance (as in a very busy city square). If the approach is to be used in such a scenario, one could draw inspiration from other methods that use motion detection to mask specific image regions during learning.

Figure: In most learning and homing images an experimenter is visible in the view, at different places. The red circle shows experimenters in images taken at the outdoor location.

In the discussion we now recommend as future work the investigation of the conditions under which neural network visual homing will perform well. Moreover, we performed extra experiments to show that visual obstruction is no problem, which we include in the main article (Figure 3a) and supplementary information (SI-8). Finally, we also include some onboard images showing the experimenter as a “dynamic object” during the learning and homing flight (SI-8).

The proposed system uses two different networks for small and large environments. There is a rationale provided - but rather than tweaking for a specific environment, some treatment of how this could be generalized as a process would help. Demonstrating the performance of the small network in the large environment (rather than just describing) and more importantly the performance of the large network in the small environment would also be interesting here and add more depth to the results.

We indeed used two different neural networks, which is due to the chronology of our experiments: We first performed almost all experiments in medium-sized environments, for which the tiny “compact”

neural network sufficed. Subsequently, we made the step to a more challenging, very large and open outdoor environment, which benefited from a slightly larger network.

It would be undesirable if one would have to tweak a network to each different environment, as it would mean continuous, expensive engineering work. We do not expect this to be necessary. Ultimately, we think it should be possible to have a network – like the slightly larger attention network – that will work for most environments.

Based on the reviewers' remark, we have now performed extra experiments. We first tested both the larger attention network and the small compact network under the same scene setup in the Cyberzoo indoor environment. This indeed shows that, in this structured environment, the (only slightly larger) network provides better performance than the compact network, especially with the online, onboard learning setup (see the new Figure 3b,c).

Furthermore we also performed extra tests with a dataset we gathered during the largest outdoor environment experiments. The results are presented in SI-13. During training, the compact network learns slower than the attention network, meanwhile converging to higher loss value. This shows its learning capacity is less good compared to the larger attention network in this more challenging environment. Furthermore, when comparing the angle prediction performance in the test dataset that we manually labelled, the compact network is showing poor performance (indicating by majority of the angle error greater than 90 *deg*), while the attention network can still perform well (with most errors remaining under 50 *deg*). We now also explicitly mention this in the supplementary information. Please note that the visual homing experiments in simulation have also been performed with the attention network.

Finally, we note that the example of the failing compact neural network also shows a strength of self-supervised learning. With this type of learning it is always possible to evaluate the error on a test set gathered during the learning flight. So, in the case of the homing network not working well enough, at least the system could detect insufficient performance before performing any outbound flight. One could even imagine that in such a case more data could be gathered or a larger / deeper network could be trained, automatically.

We now include the matter of the complexity of the network vs. the complexity of the environment in the discussion section of the main article.

Further explanation for how the Attention-based Inception Network predicts the home vector in the featureless outdoor environment would be helpful.

Based on the reviewers' remark, we fear that one of our statements in the first version of the article caused some confusion. Specifically, in the caption of the network architecture, we mentioned: "The Attention-based Inception Network, a deeper model with 10,820 parameters (42.3 kB), was used for challenging featureless outdoor environments."

To clarify: the outdoor environment was not completely featureless, in which case it would have been impossible to perform homing. Outdoors, we did have to place the home location in a very open area. Many of the natural features of the environment captured by the omnidirectional camera were very far away, such as trees in the far distance. Although far-away features carry information on heading, they do not carry any valuable information on translation. Hence, homing with only the natural, far-away features did not work reliably.

Consequently, we added a few features to the ground in the vicinity of the home location. These features served as stand-ins for absent natural landmarks. In the first version of the article, this was shown in extended figure 3f, with one pole and two mats placed on the ground. These features on the ground are

only visible from some distance by the robot, as the omnidirectional camera has a narrow vertical field of view (45 degrees). Although not always visible when within the LHA, these features added the information necessary for successful translation to the home location.

We expect that a vision system that can look down as well, will be able to exploit potential patterns of grass and dirt on the ground. Nevertheless, we believe that when choosing a home location, a robot or insect should choose a location with some close-by visual landmarks for successful translation to the home location – or, as in the case of desert ants, build their own landmarks (Freire et al., 2023).

We have now clarified this matter in the text (stating that we added objects on the ground in the main text and rephrasing the caption).

References:

Freire, M., Bollig, A., & Knaden, M. (2023). Absence of visual cues motivates desert ants to build their own landmarks. *Current Biology*, 33(13), 2802-2805.

In the simulated environment, the manuscript should explain how the 40 trees were differentiated from one another within the network, or if they were considered as similar looking, how they were placed to avoid failure.

Distinctiveness of trees

The environment generation script selected the trees randomly. Trees were selected from the standard set of trees available in the ISAAC simulation ecosystem. Some of the trees look quite similar, but many trees look very different from one another. Below, we show twenty-five example trees, placed on a 5 × 5 grid in a 20 × 20m area from the 3D object set on a white surface:

Figure: randomly sampled trees from the Omniverse tree assets.

It can be seen that the trees are generally quite visually distinct. Hence, we expect that the network can identify many of these trees as individual and unique landmarks (e.g., by means of the distinct color or shape) – although we did not analyse this up to the neural level. Please note that the visual distinctiveness of the trees suggests that the case faced by the neural network in the visual simulator is

more like the theoretical case in which each landmark has its own identity than the case in which each landmark is identical (referring to the analysis in the supplementary information).

In the supplementary information (SI-10), we now also show multiple omnidirectional images of differently generated environments. We include a few example images here:

Figure: Example omnidirectional images from the generated forest environments.

Avoiding failure in and around trees

Concerning placement, trees were placed randomly, uniformly in the environment with at least 5 meters distance from already existing trees, in order to prevent them from intersecting with each other. We also kept a clearance of 5 meters around the home location, in order to ensure that it was accessible. Furthermore, we allowed a maximum number of three trees in the 10m radius around the home location to ensure that the learned homing area was not too cluttered.

In the experiments for the original submission, we did not take tree locations into account in any other way for avoiding failure. Hence, it could happen that (1) a learning view was inside a tree, (2) an initial homing location was inside a tree, or (3) during the simulated flight, the simulated drone could go inside of a tree. Whereas condition (1) likely worsened network learning, conditions (2) and (3) led to more failed runs than necessary.

In the revised article, we now avoid conditions (1) and (2) by displacing the learning or initial position away from the landmark. We avoid condition (3) by adding an artificial potential field repulsion to the movement vector when within 2 meters of a tree (Khatib, 1986).

In order to better clarify this matter, we now include extra information in the methodology section. The discussion on tree distinctiveness is now part of the supplementary information (SI-10).

References:

Khatib, O. (1986). Real-time obstacle avoidance for manipulators and mobile robots. *The international journal of robotics research*, 5(1), 90-98.

Biological Relevance

The authors make an effort to relate the behaviour of their proposed system to natural systems, with some limited statistical analysis. The claims here seem to distill down to a comparison in the property tortuosity between the artificial system and bees - more tortuous within LHA, straighter outside, as well as a relatively velocity match: faster during path integration, slower during view memory-based

navigation. There is a prediction of the extrapolation properties of the view-based navigation memory outside the learned homing area - arriving at a 30% extrapolation capability.

How strong / relevant is this to biology? The proposed system is undoubtedly at a high level interesting in terms of the breakdown between path integration and local homing, which is the proposed navigation strategy for bees and other flying insects. In terms of the specifics and predictions - this is perhaps not that convincing (yet) - this analysis shows that the artificial system is broadly similar to the natural system, and makes an (untested) prediction around extrapolation beyond the LHA. Regardless, it's likely to be of interest to both roboticists and biologists.

Before delving into detail on the biological findings, we would like to mention that we agree with the limitations of the analysis of biological data, and have relegated that part to the discussion, and the analysis to the supplementary information.

To answer the reviewers' question: At a high level, the proposed strategy is of interest to biologists, because it forms a concrete proposal on how path integration and view memory can be combined to achieve long-range navigation. For instance, we show that noisy, drifting path integration can be used to effectively learn view memory for visual homing. The fact that Bee-Nav successfully passes the test of the real world will be important to biologists, as it demonstrates the strategy's viability despite being comparatively simple. We expect that Bee-Nav and its possible future extensions can significantly contribute to a deeper understanding of insect navigation.

The reviewers summarize the biological findings in the original submission well: The data shows that honeybees have a more tortuous flight path and a lower velocity closer to the hive than far away. The hypothesis that we test with the biological data is whether the learned homing area (LHA) captures the separation between more straight and more tortuous flight. The data supports this separation, with the maximal homing tortuosity occurring at 130% of the LHA.

Please note that we define / approximate the LHA as a home-centred circle tightly enclosing the learning flight trajectories. However, as shown in our simulation and robotic experiments, the proposed homing neural network typically generalizes beyond that approximated area (cf. Figure 2d). Hence, the observation that the separation between straight and more tortuous flight occurs beyond the tightly enclosing circle is in line with expectations.

We consider that this more detailed hypothesis and corresponding data is also of interest to biologists. However, we agree with the reviewers that it does not yet provide strong evidence for the exact underlying mechanisms proposed in our article, i.e., that learning with path integration is a main cause of tortuosity. For this reason, we now relegate the biological data analysis to the extended figures and supplementary material, referring to it in the discussion. In the discussion we make clear that additional, targeted experiments will be necessary to investigate the underlying mechanisms utilized by honeybees.

Writing and Claims

This is probably the biggest issue with the paper as it currently stands - it appears to be making some pretty strong claims, many of which are either arguably overblown or more a natural byproduct of the assumptions the system is built on. I think toned down and more specific versions of these claims would still make it worth of consideration for Nature and also more appropriate what has actually been demonstrated.

We have carefully checked all claims made in the article, making them more specific and providing additional evidence where necessary. Below we go into more detail on a main claim mentioned by the reviewers, concerning the required size of the learned homing area in relation to the total flight area.

“Simulation experiments showed that the neural network only requires training on ~0.5% of the total flight area” - this claim is correct but is essentially a very “obvious” product of two components: an assumption that the path integration system doesn’t drift too much, and hence path integration back into or near the LHA is achievable, and that the LHA homing works. I think a more appropriate claim might be something like, “when path integration can be relied upon to get the system back into the learnt homing area, that learnt homing area only needs to be X% of the total navigation range” or some similar claim. And then the authors’ analysis of the ratio of the two depending on various noise factors becomes more important too. The authors make a contribution in providing systems that meet these two requirements - no small feat, with the novelty relative to the field being more I think about the learning homing area than the path integration.

The reviewers are right that having to learn only a small area of the total flight area directly flows from the proposed navigation concept. This property constitutes one of its main advantages. Moreover, successful navigation back to the home location indeed relies on the path integration being sufficiently accurate to bring the robot back to the LHA.

However, we do feel that we need to clarify something based on the reviewers’ remark that *“when path integration can be relied upon to get the system back into the learnt homing area, that learnt homing area only needs to be X% of the total navigation range”*. Based on this remark, we reflected upon the fact that there are three elements that determine the percentage of the LHA with respect to the total feasible travel area: (i) the travel distance one wants to achieve for the task, (ii) the path integration system and its accuracy, and (iii) the neural network’s capacity to learn a given area size. In our article we have taken the required travelled distance and path integration accuracy as a departure point. We then let this determine how large the LHA has to be. Let us illustrate this with an example. If we want the robot to travel 250 meters, and the path integration of a robot allows it to travel outwards for 250 meters and reliably come back to within 20 meters (with “reliable” meaning, e.g., 99% of all travels), *then* the LHA only has to be 20 meters in radius. Dividing $\pi 20^2$ by $\pi 250^2$ then gives a percentage of 0.64%, where this low percentage is due to the area scaling with the radius squared.

We acknowledge the reviewers’ remark that the claim in the abstract should be made more specific, as the percentage depends on the exact accuracy of the robot’s path integration. Concerning the 0.5% mentioned in the original submission, it was based on an Allan variance model for the gyro noise, using a somewhat higher noise values than mentioned in the literature to better capture the higher noise due to the mechanical vibrations and electronic interference on a flying drone. At the time, we looked at how reasonable the odometry drift was after the inbound flight, i.e., compared with our experience in outdoor flights. However, we did not investigate in detail how well all the drift statistics matched our exact robotic system.

Modifications:

In order to better substantiate any claims, we took the following three actions:

- (1) We modified the noise model for the heading angle drift, so that it can be better fitted to robotic data and more easily explained in the main article.
- (2) We performed path integration experiments with our drone, measuring the position and heading angle drift.
- (3) We then searched for noise parameters that fit with both our robot but also with path integration methods from the literature, which can form a reference for other robotic systems.

All these steps have led to a more accurate determination of the percentages that can be expected from real robotic systems.

(1) Noise model

In the initial submission, we used a detailed Allan variance model for simulating gyro noise. However, this model is more difficult to fit with the data from our robotic experiments and from data reported in the literature (see action 3 below). We now employ a more straightforward Gaussian noise model, aligning it with models used in the literature (Thrun et al. 2005, Wystrach et al. 2015, van Dijk et al. 2024). Specifically, we now add heading drift noise according to the normal distribution $\mathcal{N}(0, t\sigma_\psi^2)$, where t is the time in seconds. So, if we have a time step of 1 second, then each time step we draw a random variable from $\mathcal{N}(0, \sigma_\psi^2)$ and add it to the heading estimation. This leads to an accumulation of heading drift over time. The distance estimation noise is still modelled as $\mathcal{N}(0, d\sigma^2)$, with d the distance in meters.

(2) Path integration experiments

We performed various path integration experiments with our drone, measuring the position and heading drift over time (see the figure below for an example). We did this in different conditions: flying above artificial grass with little texture vs. flying above a highly textured underground, and flying with vs. without wind (both above the low-texture artificial grass). The experiments provide data for position drift after different flight distances and heading drift at given flight times.

Figure: Robotic experimental result for measuring path integration drift in our motion tracking arena. *Top left:* The test environment, with artificial grass, so limited texture. *Top right:* Motion capture ground truth position vs. odometry-based position estimate. *Bottom left:* Position error over flown distance. *Bottom right:* Heading drift over time.

(3) Noise parameters

Subsequently, we searched for noise parameters that approximated the robot's drift statistics. Please note that the path integration used by our robot is very standard and does not feature a heading measurement (no "compass"). Path integration without a compass of any sorts is referred to as "idiothetic" path integration in the biology literature (Heinze et al. 2018), and is known to be relatively inaccurate. In the case of our robot, we use PX4's visual odometry estimate from a downward looking camera and laser. For the heading estimation, the Extended Kalman Filter depends on noisy MEMS gyro measurements, which are integrated over time. We finally selected $\sigma_\psi = 0.63^\circ$, and $\sigma_d = 0.10\text{m}$ for our robotic system, which gave a mean heading drift after 100s of 5.03° and a mean position drift after 100m of 0.91m. When simulating 1000 outbound and inbound travels, this setting results in a Learned Home Area (LHA) that is 3.84% of the total flight area.

This means that even with the quite crude path integration of our robot, the percentage of the LHA with respect to the total flight area is very low. We then repeated the simulation-based "noise level" experiment, scaling both σ_ψ and σ_d with $\{0.10, 0.5, 0.75, 1.0, 1.25, 1.5\}$. With a requirement of the LHA containing 99% of the return positions, this gives LHAs that span from 0.1% to 10.82% of the total flight area. The expected percentage for our robot is indicated by a circle marker (Figure 2b).

In order to give a more comprehensive idea of what could be achieved with robotic systems, we also fitted our noise model to what is known of two methods from the literature. First, more advanced versions of Visual Inertial Odometry (VIO), using a frontal camera and IMU, feature a better path integration accuracy. To illustrate this, we also determined parameters for the noise model that led to a similar drift as that of the visual-inertial odometry method "SVO-GTSAM", as reported in (Delmerico and Scaramuzza, 2018). In that article, SVO-GTSAM is reported to have a mean drift of $\sim 0.3\text{m}$ and heading drift of $\sim 0.8^\circ$ after 35 meters of flight. Our noise model, with $\sigma_\psi = 0.25^\circ$, and $\sigma_d = 0.015\text{m}$ leads to similar statistics: a mean position drift after 35m of 0.32m and mean heading drift of 0.79° . With this VIO, the LHA percentage of the total flight area would be 0.74% according to the simulation experiments. It has to be noted that SVO-GTSAM requires substantial processing. Hence, it would require considerable onboard computational and memory resources – which goes against the efficiency of the proposed navigation method.

Second, we also fit the noise model to a robotic odometry method from the literature that does use a compass, in the form of a magnetometer (Stankiewicz and Webb, 2020). They use an efficient, bio-inspired VIO method and report a position drift of 1.5m after 100m of robotic flight. Since their heading does not drift, we slightly adapted the noise model. In particular, in the case of their setup, the heading noise no longer accumulates, since the magnetometer can provide an absolute heading measurement. Instead, the Gaussian noise is added at each time step to the true heading angle. The so-adapted noise model, with $\sigma_\psi = 5.5^\circ$, and $\sigma_d = 0.15\text{m}$ also leads to 1.5m drift after 100m of travel. The corresponding percentage of the LHA is 0.24%. This example demonstrates that sensing of an absolute heading angle (called "allothetic" path integration in the biological literature) substantially improves the accuracy of path integration and hence the efficiency of the proposed navigation strategy.

Final remark:

We have incorporated the new results in the article, now also briefly explaining the noise model in the main text. The details explained above have been incorporated in the supplementary information (SI-11). Moreover, we have rephrased the claim in the abstract to clarify that the exact percentage depends on the odometry accuracy: "Simulation experiments showed that for realistic path integration accuracies the neural network only requires training on $\sim 0.25\text{-}10\%$ of the total flight area.". So, we now refer to a wider range of percentages and clearly make the link to the accuracy of path integration.

References:

- Wystrach, A., Mangan, M., & Webb, B. (2015). Optimal cue integration in ants. *Proceedings of the Royal Society B: Biological Sciences*, 282(1816), 20151484.
- Thrun, S. and Burgard, W. and Fox, D. (2005). *Probabilistic Robotics*. The MIT Press, Cambridge, MA, Intelligent Robotics and Autonomous Agents series.
- van Dijk, T., De Wagter, C., & de Croon, G. C. (2024). Visual route following for tiny autonomous robots. *Science Robotics*, 9(92), eadk0310.
- A Benchmark Comparison of Monocular Visual-Inertial Odometry Algorithms for Flying Robots, by Jeffrey Delmerico and Davide Scaramuzza, 2018 IEEE International Conference on Robotics and Automation (ICRA), May 21-25, 2018, Brisbane, Australia.
- Heinze, S., Narendra, A., & Cheung, A. (2018). Principles of insect path integration. *Current Biology*, 28(17), R1043-R1058.
- Stankiewicz, J., & Webb, B. (2020, July). Using the neural circuit of the insect central complex for path integration on a micro aerial vehicle. In *Conference on Biomimetic and Biohybrid Systems* (pp. 325-337). Cham: Springer International Publishing.

Some of the language could be tightened up e.g. “randomized, tortuous outbound trajectories” - it’s implied this is bad, but is it actually bad / a hindrance / extra challenge for this system, in these experiments? Can the authors be more specific / clarify here.

The reviewers refer to the description of the outbound flights in the simulation experiments that determine the required LHA ratio. This description was not meant to have a positive or negative connotation. To clarify: the proposed approach to navigation specifies how a robot can return to its home location. However, it does not specify the flight behavior during the outbound travel, other than that the robot will have to perform path integration during this phase. The outbound flight behavior will depend on the specific task. It may be quite tortuous, for example, if a robot is tasked with searching for an object in the environment. It could also be quite straight, if the robot flies towards a known location.

In order to study the required ratio of the LHA with simulation experiments, we need to assume a certain outbound flight scenario. In the initial submission we opted for a tortuous trajectory with randomized turns to capture an erratic search behavior. In the revised article we employed a scanning search pattern similar to that used in the robotic experiments, as could be useful in a search mission. We now mention this in the text.

General / Presentation

In the simulation experiments section, the text appears to reference Figure 2e instead of Figure 2f. In Figure 2d, the last two x tick labels are overlapped, which should be placed one after another. In Figure 3f, the data points should not be connected horizontally into a line as the x labels are the environment types and even though they are placed in increasing difficulty order, they have no correlation.

The extended data figures 4 and 5 do not appear to be cited in the manuscript.

In Figure SI-13, the order of the figures for the spiral and wasp-like flight patterns in the two environments is spiral and then wasp-like, but the figure caption and its description in SI-6 section for figures SI-13d and e use the reverse order when referring to their quantitative results from the plots.

Some figures such as Figure 3, all Extended data figures 1-6, and all figures SI 1-13 have images which have been rendered with low quality, which makes it hard to read.

Thank you for noticing these issues – we have corrected them in the new version of the article.

The rendering is due to the conversion from Word to pdf by the Nature submission system. Where possible we have tried to enlarge details to improve the readability. Figures are also uploaded separately in high resolution, but we do not know if these are available to the reviewers. In any case, we apologize for this inconvenience and will make sure that in a possible final version, the figures are all in a suitably high resolution.

Supplemental

Contains a large amount of relatively theoretical / example-based analysis of various homing conditions, with visually unique and non-unique landmarks, of varying numbers.

“similarly looking landmarks” - I find this phrasing a bit awkward - “similar appearance landmarks”?

Videos are good additions.

Thank you for your evaluation of the supplementary material and videos. We have rephrased the section heading as suggested.

Referee #2 (Remarks on code availability):

I've (as an experienced researcher who does plenty of coding in the robotics field) examined the provided source code, configurations, and data. The following points summarise my findings regarding clarity, documentation, and the ability to run the code and replicate the manuscript's findings.

Code Execution and Replication

- The environment installation and execution instructions for the theoretical simulation analysis worked correctly, allowing for the successful replication of the experiments.*
- The robot offboard training and testing module could also be successfully run.*
- Unfortunately, it was not possible to run the robot onboard experiments and the visual simulator's simple experiments. The system configurations in the visual simulator's run_simple_experiments.py use hardcoded local paths within the .json files. Even after attempts to modify these paths, errors related to finding the image folders persisted, making it impossible to determine the root cause and execute the experiments.*

Suggestions for Improved Documentation and Structure

- System inputs, outputs, and hardware: the README files for all modules should clearly detail the expected inputs and outputs for each system and explicitly state the hardware requirements needed to run each module. This is particularly important for clarifying which experiments can be run with the provided data and which require additional hardware (like the onboard system).*
- Data usage clarity: although the input data is provided separately, the repository's README files and the data folder's overview should clearly map which specific dataset within the provided data is necessary for each experiment (e.g. for the visual simulator experiments). This is because not all datasets contain the required data types for every system.*

- *Output location: the README files for all modules should also clearly explain what the expected outputs are and where the resulting files are stored.*
- *Environment installation: providing instructions to use an environment manager like Conda or pixi for the theoretical simulation and robot offboard/onboard systems could significantly streamline the installation process.*
- *Visual simulator (IsaacSim): the README's linked documentation for the recommended IsaacSim version (4.2.0) is broken. While the initial environment setup worked on version 4.5.0 (with minor import modifications), the instructions should include all additional Python packages required for the IsaacSim environment to run the visual simulator experiments.*
- *Visual simulator (IsaacSim): the reliance on hardcoded local paths in the configuration files, particularly for finding image data, should be resolved to ensure the experiments are runnable on other systems.*
- *Code organisation: The source code's structure could be improved for easier navigation. Sorting different file types, such as putting all .json configuration files into a dedicated configurations subfolder, would make it clearer to find the scripts necessary for execution.*

We thank you for making the time and taking the effort to try and run our code – we truly appreciate this. Our apologies that the visual simulator experiments could not be replicated by you at the time of this first review. We value the suggestions given and have now followed these with the hope that a potential next attempt will be successful.

Specifically, we now provide a containerized environment (Docker) based on Isaac Sim 4.2.0 to run the visual homing simulation experiments. This setup minimises the steps to installed the required software and configuring paths. However, there are still some hardware and software requirements. Notably, NVIDIA Isaac Sim required an NVIDIA GPU to run. More information can be found in the README document.

We hope that the improvements of the installation setup, documentation, and code structure will enable both the reviewer and potential future readers of the article to reproduce also the visual simulator's experiments. Concerning the robot's onboard code, we have updated it. However, as is quite common in robotics, specific hardware requirements make it more difficult for others to use.

Response Letter

Dear Editor,

We thank you and the reviewers for the thorough evaluation of the revised manuscript. Based on this we have revised our manuscript, as described in the point-by-point review reply text below.

To summarize, the main points we addressed in this second revision are:

- We have carefully considered the reviewers' comments for better contextualizing and caveating the proposed approach and its capabilities. For instance, in the discussion we now more extensively discuss how visual homing with neural networks will generalize to different environments, and more clearly indicate the next steps to further improve the approach.
- Based on your comments, we propose a new title, namely: "Efficient robot navigation inspired by honeybee learning flights." It emphasizes the efficiency of the method and avoids the general descriptor of "insect-inspired", specifying the inspiration from honeybee learning flights. Moreover, it indicates that the method is suitable for robot navigation (not just foraging, but also not just for flying robots as the method can be applied to robots with other means of locomotion as well). We also removed references to "foraging" robots from the abstract and discussion.
- We have removed excess references, moving some of them to the methods section reference list, and added the author contributions.

In the review reply document below, we address all points raised per reviewer with their comments in *blue, italic font* and our responses in black, normal font. We hope that our response and the revised article meet your and the reviewers' expectations.

On behalf of all authors,

Guido de Croon

Full Professor at Delft University of Technology

Referee #1

Overall, this is a very thorough evaluation of the navigational capabilities that can be obtained for a flying robot by combining larger scale path integration with compact neural networks that associate path integration state to visual input in the nearer vicinity of the target; which might also be a plausible model for insect behaviour. I'm not convinced this is a major conceptual breakthrough for either robotics or biology but it is very interesting work, and likely to have significant impact.

I greatly appreciate the substantial effort by the authors to respond to the points raised in my previous review, specifically:

- 1) Clearer contextualisation of how the approach relates to state of the art navigation approaches in small flying robots.*
- 2) Direct evaluation of the extent to which on-line/on-board learning is plausible for the approach.*
- 3) Extended comparison (in simulation) of alternative insect-inspired visual localisation methods.*
- 4) A more thorough exploration of the path integration noise assumptions.*

There are still some points that should be addressed:

We thank the reviewer for the kind words and recognition of the effort that went into the revision. We feel that the reviewer's initial comments and ensuing changes in the revised version have substantially improved the article and its potential impact.

We address the remaining points one by one below.

*Abstract, lines 18-20: "in real-world experiments,18 a tiny 42-kB neural network sufficed for a small drone to fly outbound trajectories up to 600 meters and return using direct inbound trajectories shorter than 150 meters". This is using offline, offboard training, and appears to refer to a single trial using a larger radius learning flight than the main data reported in the paper (as described in the results on line 175-177, and provided as a video supplement, but not otherwise mentioned in the methods or figures). The *directness* of the inbound trajectory (150m vs 600m) seems somewhat irrelevant to highlight, as this is largely due to using rather standard odometry to convert a convoluted outward path to a straightline direction home. What actually matters is repeatable accuracy in pinpointing the home location, and my understanding of the key results is that for flights of a similar range (100s of meters), the success rate in reaching the desired level of accuracy was 6/9 (fig 3) - maybe 7/10 if this extra 600m trial is included? A more representative 'headline result' should be provided in the abstract, e.g., "100% success in returning within 0.5m of home for real-world indoor and outdoor flights of 50-100m, and 70% for flights of up to 600m, including windy conditions". N.B. I am inferring the definition of "success" from lines 152-153, as a search on "success" in the paper did not locate any other definition, e.g., in the methods.*

We agree with the reviewer that the difference between the outbound and inbound trajectory length is not the most illustrative main result, because this heavily depends on the outbound behaviour.

Hence, we followed up on the reviewer's advice and have replaced this result with a phrase on the success rate of the robot experiments. Moreover, we have added a clear definition of success to the methodology section.

It would be helpful to clarify how the two outdoor environments (described line 636-637 (methods) as a tennis court and an outdoor test field, without any dimensions provided) correspond to the “small outdoor” and “large outdoor” environments in fig 4c. This is also not adequately clear in lines 165-177 (main text). In fact, it would be preferable to provide (in the figure and main text) some actual dimension rather than ‘large’ and ‘small’, e.g., the average outbound distance flown for each condition. Some of this information can be inferred from extended data figure 7, which shows that that “large indoor” flights were in fact similar in scale to “small outdoor” flights, rather than “large outdoor” flights. There seems no reason not to quantify these more clearly.

We agree with the reviewer that some of the dimensions of the experimental environments and the distances were not clear. We have now added the dimensions of the environments at the relevant places in the manuscript (the main article, figure caption, and methodology). Moreover, we made clearer which areas are referred to as the large indoor / outdoor areas in the captions and methodology.

Given the relatively small number of real-world flights, all the data for all the flights could be included in extended fig 7. In this figure, the illustrated failures appear to occur despite the path integration returning the drone to within the LHA. This seems worth some explicit discussion in the main text, given the previous tests (fig 3b) that show 100% success for homing within the LHA.

This remark consists of two parts. The first part concerns including all real-world flight trajectories. We currently include only two runs per environment in order to keep the extended figure size limited. In fact, we performed four to five experiments in each environment. Including all these experiments in the figure, would make it 2.5 times as large. We now include all the full flight trajectories in two figures (Extended Data Figure 7 and 8).

The second part is about failed experiments in the large, outdoor terrain, whilst the robot seems to end up within the LHA. We use “seems to”, because the circle enclosing the learning flight trajectory was actually scaled by a factor of 1.5. This factor was chosen based on the observed generalization beyond the strict LHA in most simulated and smaller indoor and outdoor areas. The below plots show the failed outdoor trajectories where the LHA is now shown with a scale factor of 1.0.

The plots show that the robot starts homing at the edge of the LHA, quite far away from the locations on the learning flight trajectory (grey lines). Hence, no training sample was taken close to those locations. In an environment with more easily visible landmarks this may have posed no problem. However, it did lead to failure due to the challenging conditions in this large, wide-open outdoor environment, with few natural close-by features. In particular, when it is very windy, the drone needs to angle (30°) hover with a substantial trim angle. Despite the algorithmic corrections for tilt, this worsened the visual homing accuracy (likely because the visual ground features appeared differently or were even not visible from those locations). We now comment on this in the main article (in the “Robot experiments” section).

Figure: Failed outdoor runs with the LHA plotted at a factor 1.0. Visual homing (red lines) in the failed runs starts at the edge of the LHA, away from locations on the learning flight trajectory (grey lines).

There does seem to be a significant difference in the error for the different online/onboard variants of learning (fig 3c), leading to decreased path efficiency (3b), which does not seem adequately captured in the brief description (lines 155-157). It seems worth commenting that onboard online seems to do as well as onboard offline. In methods, line 675, should this be Onboard offline?

The reviewer is correct that onboard, online learning leads to similar results as onboard, offline learning. We initially did not mention this for brevity, but have now added this observation to the main text. Moreover, we thank the reviewer for noticing the mistake in line 675, which has now been corrected.

I find the discussion rather underwhelming, both as an analysis of the main strengths and limitations of the approach for robotics, and as providing insight for biology (n.b. line 183, should be 'purposely' rather than 'purposively').

We thank the reviewer for the suggestions to improve the discussion. The word “purposively” has now been corrected to “purposely”. We will address the other points below.

For example, is there anything in the network training approach that limits the number of views (or size of the LHA) that could be successfully encoded? Would learning a more extensive distribution of views, perhaps more sparsely, improve robustness, e.g. to sudden displacement of the robot (kidnapped robot problem)?

The reviewer raises a fundamental question here. The visual homing neural network forms a function that maps an image to a home vector. The potential complexity of this function, i.e., the network’s “capacity”, is determined by the architecture and general setup of the neural network (numbers and types of layers, numbers of neurons or channels per layer, activation functions, etc.). In general, deep neural networks are highly effective at extracting task-relevant visual information. In terms of the task at hand, this means that if there *is* information in the images on the home vector, it is likely that a deep neural network can be designed that will extract it. Of course, when the area that needs to be learned is larger or more difficult, then the deep neural network will have to be larger as well.

So, given a neural network of limited size, such as the tiny 3.4kB and 42kB networks in the robot experiments, the LHA will be limited. The exact limits for a network will be hard to determine though, since it heavily depends on the objects in the environment and their visual appearance. Of course, one could choose to learn a larger LHA, possibly by using larger networks. This would indeed help with the robot kidnapping problem. Please note that when learning a larger region, the views included in training do not have to be sparser, as the views themselves are not stored – the network weights implicitly represent the information that needs to be extracted for successful homing. Denser views will lead to better homing accuracy, but may increase the duration of learning flights (if the drone needs to stop for each view) and may increase training time.

We now discuss this matter in the article’s discussion, suggesting this as an important topic for future research. Moreover, in the supplementary information (SI-6) we now mention that future work could explore different learning flight patterns to see what is most effective for learning a given area.

The supplementary material includes a brief comment (lines 747-8) that dynamic visual elements contribute to the lower success rate in outdoor environments – this seems a point worth discussing in the main text.

We now mention this in the main text when we comment on the lower success percentage in the large, open outdoor test environment.

Lines 196-198 in the main text are very vague about the most crucial next steps to evaluate this approach. From the biological perspective, could the MB learning model be used, in the same way as the more abstract neural networks, to associate visual inputs to vectors rather than just left/right directions? In other words, is the difference observed between the algorithms due to differences in the learning mechanism (e.g. in capacity or generalisation), or differences in what type of outputs are associated to views? And could this suggest ways in which current models of neural circuitry for learning in the insect are incomplete? There is some discussion of this point in the response to reviewers but it is not provided in the paper.

We now more clearly emphasize and explain the most crucial next steps in the discussion:

- (1) Complement the approach with a way to land precisely at the home location.
- (2) Have the robot perform a systematic search if visual homing does not lead to the home location,
- (3) Achieve more accurate path integration by including a compass measurement, from magnetometer or polarization sensors.
- (4) Have the visual homing neural networks output uncertainty, so that the robot can start to perform visual homing when it recognizes the LHA, and fuse the visual homing outputs with its path integration. This can also be used to trigger a new learning flight when the environment around the home location changes.
- (5) Better understanding of the capacities of the visual homing neural networks – in terms of the size of the LHA, and the types of environments that can be learned,

In terms of biological relevance, the next steps would involve:

- (1) Perform biological experiments that aim to ascertain that noisy path integration is involved in the learning of view memory.
- (2) Investigate even more efficient, biologically more plausible neural networks, e.g., as inspired by the mushroom body.
- (3) Extend the approach to enable navigation to different places.

Concerning the reviewer's questions: We do believe that the mushroom-body model can be modified to associate visual inputs to vectors and expect that this would lead to shorter and more successful visual homing trajectories. One could stick to the current familiarity-style learning and use a set of neurons to represent discrete directions (in a similar vein as the identified ring attractor circuit). Further research is needed to determine whether this works sufficiently well, or whether additional changes are needed to perform continuous regression. It would be interesting to revisit known neural data of the insect mushroom-body circuit and investigate whether the neural structure could encode directional data.

We have also added the discussion on biological plausibility of the networks to the discussion.

Lines 218-219 "the strategy could be extended to include the learning of different places and the neural machinery to take shortcuts between them" could cite some of the earlier discussions of how association of vectors to places might play a role in insect navigation, including the enabling of shortcuts, e.g., Cartwright, B. A., and T. S. Collett. "Landmark maps for honeybees." Biological cybernetics 57.1 (1987): 85-93.

We have now extended this discussion, mentioning how vector subtraction has been envisaged for taking novel routes, referring to the mentioned article.

Finally, we thank the reviewer again for the thoughtful reviews, which have in our opinion considerably improved the manuscript.

Referee #2 and #3:

The authors have made a very substantial effort including significant new work to respond to the reviewers' comments. This review breaks down into a few main components: 1) what the authors have done in response to the reviews and the adequacy / compellingness of those changes / additions / responses 2) the resulting significance and appropriateness of the work as it is presented now, also held up against the updated set of claims, and 3) specific technical suggestions.

We thank the reviewers for recognizing the substantial effort that went into the revision. We will address the remaining remarks and suggestions below.

Changes / updates:

Onboard / offboard: I agree with referee #1 that much more clarity was needed around where the training occurs - this appears to have been addressed in the revisions. I however also agree with the authors that offboard training is not necessarily a major pragmatic limitation for the specific niche of applications that this work is targeting - this is not general purpose, map-based SLAM / navigation, this is an attempt to create a highly efficient navigation system that works under specific circumstances, and initial offboard training is likely OK for many of these. Where this might be a concern is the general maintainability of the system over time, but that's addressed in a separate point in this review. The authors' response updates 4 potential scenarios: realistically some of these likely start invalidating some of the simplicity benefits of the proposed approach by bringing in more learning and complexity - that's OK.

Comparisons: comparisons have been added / updated for some other lightweight approaches: more computationally intensive systems have mostly been disregarded (except ORB-SLAM 2) as being of a different class of systems - this is OK.

Practical potential: there is plenty of response by the authors to various aspects of the initial reviews. One is around the initial focus on compute: the authors acknowledge that compute energy usage may be affected by other factors like other compute tasks beyond navigation (and of course, general actuation of the micro drone). And the argument that, all other things being equal, less compute / power consumption is always helpful - this is true, but the actual extent to which this is practically significant or not really depends on the deployed system and context. There is a much longer ongoing discussion to be had here around the various implications of trade offs, scope etc... and whether those will actually unfold in practice, but that doesn't really effect my decision on the paper.

We agree with the reviewers that the relevance of the computational and energy savings will depend on the exact system. We expect that the efficiency brought by the proposed approach will be vital to small, resource-constrained robots. The efficiency will still be useful for larger, less constrained robots, but the future will tell for what applications the approach will be most useful. We now comment on this aspect in the discussion.

On wind and tilt: the authors have tweaked their claims to be more appropriate - and I re-iterate there is nothing wrong with including important engineering considerations. I would suggest the authors also note that the approach to adjusting for tilt in the image rotations is very brittly sensitive to the tilt measurement output by the onboard sensors / the horizon detection: this is a valid approach used in much past work and is probably valid for the subset of tasks this work is envisaged to help with, but can go wrong very quickly if the tilt estimate is even slightly off - this is where more high end feature-based approaches shine, because they can be made much more inherently robust to pose variations in the platform.

We now more clearly mention in the methodology that both methods used to correct for tilt have their specific weaknesses. The vision-based correction needs a clearly visible horizon (an open outdoor environment), whereas the model-based correction can be subject to drift if it is based on the inertial measurement unit.

In general, the authors have responded to and refined claims around issues like level of biological plausibility / relevance, and clarified intended interpretations around some of the text and made changes in the manuscript accordingly.

I am not fully satisfied with the response of the authors (both scientifically and in terms of practical impact, as well as in the context of how important this component is for the whole research effort) is the response to the request for a better characterisation of the situations under which this approach (the LHA component) will or won't work. This sort of analysis is also expected for much more complex / high end navigation systems - like much of the work in semantic / language-based navigation in computer vision - it's not a unique request to this particular approach. Nevertheless, many of the points have been at least responded to, and much of my remaining hesitation isn't around the inherent value of the work but around a) the claims and b) the likely significant / impact, which I address now.

The reviewers refer here to the characterization of when the proposed approach will work, and in particular the conditions under which a deep neural network can successfully map images to home vectors for visual homing. They rightly state that this matter is also of relevance to other applications of deep neural networks, from computer vision to language processing and beyond. We will go into more detail on this comment below, where the reviewers elaborate on this point.

Assessment of the updated body of work.

The paper is generally improved across the board, in terms of more appropriate claims, extra experiments / evidence to provide more justification for some of the claims, and text and graphical changes / updates to support all of this.

The strongest endorsement of the paper is that the authors have taken a compelling observed navigation phenomenon from nature: the usage of approximate dead reckoning on long paths and the assumption that dead reckoning will get the agent / robot / animal back to the approximate vicinity of the home goal, combined with a computationally lightweight approach to learning how to navigate within that home area, to create and demonstrate both in real world experiments and in simulation the system functioning at a high level, within the specified task and environmental constraints. This is novel, almost certainly of widespread interest to much of the community, and an impressive systems / engineering feat as well (and a major effort, although a truism: so are most submissions to Nature I imagine). The revisions have in general improved the paper and the quality of the research.

We thank the reviewers for recognizing the novelty of the work and for expressing their expectation that it will be of widespread interest.

I am still slightly uncomfortable about how the contribution is presented, given its heavy dependence on a path integration system - which, although expanded in this revision, isn't really a part of the research contributions of this work - that reliably brings the system back to the vicinity of the home goal, at which point the learnt homing system kicks in. The discomfort isn't about this point in isolation, but in combination with what I would still regard as a somewhat underwhelming and not-sufficiently-rigorous characterisation of the types of home environments where the homing system is expected to learn and successfully operate. The revisions have provided some qualitative observations about some dynamic obstacles, and bit of commentary on ambiguity, opaque obstacles etc... but this is more commentary than anything rigorous. It's also been addressed as "future work" in the response so far. To

be clear, I am expecting the system to only operate in a small fraction of potential environmental configurations - just like any other mapping / navigation system including even the more heavyweight ones - but more clarity around this would significantly improve the strength and rigour of the work, and usable relevance to the community who might follow on from this.

We agree with the reviewers that this is an important matter, which may have been insufficiently addressed even in the revised version of the manuscript.

We expect that deep neural networks will be able to learn visual homing in many different environments. The main reason for this expectation is that deep neural networks have a high “capacity”, i.e., they can model complex functions. Generally, deep neural networks are highly effective at extracting task-relevant visual information. In the context of the proposed navigation approach: if the image contains visual landmarks, then a sufficiently large deep neural network will likely be able to extract this information for predicting the home vector. Many environments will have visual landmarks, i.e., visually recognizable objects with a fixed spatial relation to the home location.

One question is about what the size of a deep neural network needs to be to extract visual homing information. Our experiments suggest that strikingly small neural networks (3.4 – 42kB) may already suffice for moderately sized to large environments. However, it is notoriously difficult to determine exactly which functions a specific deep neural network can or cannot represent. In the context of the article: there may be environments for which these networks do not have sufficient capacity. In that case, the training performance will already be sub-par. Because the proposed approach relies on self-supervised learning, this is a failure case that can be automatically detected before the drone would fly further away. One could hence imagine an automatic procedure for scaling up the complexity of the network if the environment requires it.

Of course, difficult, information-poor environments may exist, such as long office corridors that have many similarly-looking places or large, wide-open terrains. As we suggested in the first response letter, drones could take this into account when choosing a home location. That is, they should ensure the presence of identifiable, close-by landmarks. Dynamic objects form a different challenge. As shown in the supplementary information, dynamic objects that only occupy a small part of the image (such as the experimenter at some distance of the drone) actually have a negligible influence on the predictions. However, if dynamic objects take up a large part of the images, they may become problematic, requiring an extension of the approach (e.g. masking them out).

Based on this important remark, we have extended the reflections on this matter in the discussion section of the article. Moreover, we have performed additional simulation experiments in different virtual environments than the forest environment. The ensuing simulations were all successful, further corroborating our expectation that the neural networks will function in various environments (SI-14).

A minor pragmatic consideration too is that path integration / dead reckoning systems in practice of course often don't work in the way that noise models suggest - almost always there are catastrophic failures caused by any of an infinite variety of effects - this again doesn't impact the core value of this work but would help better place its potential impact. Recovery for this system would be different (probably some sort of expanded search) to recovery of a system that has mapped the entire (not just local) area - again this is fine, but implies a different practical impact.

The reviewers are right that path integration may not bring a robot back to the learned homing area, potentially due to a catastrophic failure, after which it could get “lost”. In fact, for insects, expanding search behaviours are part of their navigational toolkit (cf. Lambrinos et al. 2000). We think that such an expanding search behaviour would be a prime candidate for extending the proposed approach, and now mention this in the discussion.

Reference:

- Lambrinos, D., Möller, R., Labhart, T., Pfeifer, R. & Wehner, R. A mobile robot employing insect strategies for navigation. *Rob. Auton. Syst.* 30, 39–64 (2000).

Specific details:

SVO-GTSAM LHA percentage of the total flight area is 0.74% in the paper, but the results obtained from running the code shows that the LHA percentage of the total flight area for SVO-GTSAM is 0.0148%. It would be great to clarify if the value presented in the paper is the area achieved using SVO-GTSAM or the area achieved using the proposed approach with properties similar to SVO-GTSAM.

We thank the reviewer for taking the time to run our code and for identifying this issue. The discrepancy was caused by an incorrect default setting in the submitted version of the code: the parameter *heading_measurement* was mistakenly set to *True* rather than *False* for the SVO-GTSAM experiment. Because absolute heading measurements should not be available in this scenario, this error led to the extremely low ratio value observed (0.0148% vs 0.74%). We have corrected this parameter in the updated code.

In figure 3c, $n=8$ homing flights/learning steps but the x-axis label mentions different n values for the different scenarios.

Also, the variable n is used for the number of full rotations, the number of learning steps, and the number of attempts the same target waypoint is not reached?

We agree with the reviewer that there is confusion of the different meaning of n on different appearance in the article.

To resolve this we have added subscripts to all n -parameters. See below the revisions made:

Meaning	Appearance	Previous representation	Corrected representation
Number of homing flights	Methods-Simulation experiments	$n = 8$	$n_{\text{homing-flights}} = 8$
Number of homing prediction steps	Figure 3c x axis	$n = 128, n = 158 \dots$	$n_{\text{steps}} = 128, n_{\text{steps}} = 158 \dots$
Number of full flights	Figure 4c y axis	$N = 4 \dots$	$n_{\text{full-flights}} = 4 \dots$
Number of loops of the learning flight	Methods-Simulation experiments & Experiment setup	$n = 6 \text{ loops} \dots$	$n_{\text{loops}} = 6$
Number of attempts to reach waypoint during obstacle avoidance	Methods-Experiment setup	$n \text{ attempts}$	n_{attempts}

In line 500, the investigated radius factors are (0.5, 1.0, 1.5, 2.0) but the results in Figure 2d, show that it is evaluated from 0.5 to 2.5 in steps of 0.25.

We thank the reviewer for this careful observation. During our latest simulation experiments, we used a step size of 0.5, extending it to 2.5. However, the plot shows ticks at multiples of 0.25. We have now updated the figure and the manuscript's description to accurately reflect the performed experiments.

In SI-7 under larger dataset analysis and bootstrapping test subsections, the figure that the text refers to is Extended Data Figure 6, but it should be Figure SI-16.

In SI-8 under the results section, the text refers to Figure SI-20, but it should be Figure SI-19.

In the figure captions where the LHA is described, the area is referred to as a dotted circle, but shown as a dashed circle.

We thank the reviewers for carefully noticing these issues; we have now corrected them in the article.

Referee #2 (Remarks on code availability):

Code Quality

The updated code repository has sufficiently addressed all the concerns I raised previously. All instructions are given for code installation and execution of each project within the code repository. To properly replicate the visual homing simulation experiments, instructions for a docker environment are given, which is easy to follow.

The README files have provided all details about the system input and outputs and hardware requirements, as well as describing the expected results and where the results are stored. The README file of the provided data folder clearly explains the details of each data and how they can be used and/or visualised.

The revised code repository has removed all hard-coded local paths, and has replaced them with relative file paths, which now work without requiring any modification.

There were only a few minor issues that I came across, which would be worth addressing when releasing the code:

We are glad to read that the reviewers consider their comments on the code repository sufficiently addressed. We will delve into the remaining minor issues below.

In code_theoretical_simulation_analysis/insect_navigation.py, the configuration property cancel_checkered_neurons is used in one of the conditions. However, none of the .json files have that property.

In robot_network_training/home_learning_laptop/evaluate.py, the plot_vectors function call for the condition where the ground truth is mocap has missed the 'heading_mocap' variable to be passed as one of the required parameters. We

For ease of the user, it would be great to provide the option to save the generated plots across all modules with meaningful names, as opposed to the current approach of showing them using the matplotlib's popup window. This is because when too many figures are generated from the same script, it would be hard to keep track of what they represent.

We agree with the reviewers. Thus, for the theoretical analysis code and path integration noise simulation code, we have now changed the code such the outputs are saved as csv and figures in an output folder.

I could test all possible modules except the robot onboard due to specific hardware requirements:

Theoretical simulation analysis (generated supplemental results, figures 1-12)

*Provides the visualisations for the learning of the home vectors within the learned homing area.
Path integration noise simulation (generated figures 2a and 2b)*

The model for a drone flying outwards and then attempting to return home using path integration with noisy sensors and uses different noise simulations.

Robot network training (generated extended data figure 5a)

The code for training the home learning network using the 3 different collected data that was provided: Cyberzoo, outdoor goodday, outdoor orangepole.

Robot onboard

The code for the onboard robot system requires Raspberry Pi 4 with PX4 flight controller, and a PiCamera. As it needs these specific hardware to run the system, I could not test it. The README files describe all components of the system and provide information on the prerequisites and installation.

We fully understand that it was not possible to run the onboard robot system, given that it requires the robotic hardware, which is described in the methodology.

The provided code seems to contain the required files.

Visual simulator docker (generates figures 2c, 2d, 2e and 2f)

The simulation environment. Each script runs and generates random forest environments, saves the generated maps, onboard images and training data (forests) and saves the results and trained models.

Visualised obtained results from the onboard robot testing:

Cyberzoo: using the provided python script (generated figure 3f – homing trajectories) Followed the instructions to view the PX4 data from the flights for robot learning and full flight trajectories from 7 different collected data. The visualisations include the performance of the controller and the estimated trajectory. The extended data figure 7 shows the flight trajectories of these data.

We thank both reviewer #2 and #3 again for the thorough and constructive reviews, which have helped to substantially improve the manuscript. Moreover, we want to commend them for their careful evaluation of the code, which will be of great help to other scientists that want to reproduce the results.